# Test-Time Anchoring for Discrete Diffusion Posterior Sampling

**Litu Rout** [1 2]   **Andreas Lugmayr** [2]   **Yasamin Jafarian** [2]   **Srivatsan Varadharajan** [2]   **Constantine Caramanis** [1]
**Sanjay Shakkottai** [1]   **Ira Kemelmacher-Shlizerman** [2]

## Abstract

While continuous diffusion models have achieved remarkable success, discrete diffusion offers a unified framework for jointly modeling text and images. Beyond unification, discrete diffusion provides faster inference, finer control, and principled training-free guidance, making it well-suited for posterior sampling. Existing approaches to posterior sampling using discrete diffusion face severe challenges: derivative-free guidance yields sparse signals, continuous relaxations limit applicability, and split Gibbs samplers suffer from the curse of dimensionality. To overcome these limitations, we introduce Anchored Posterior Sampling (APS), built on two key innovations: *quantized expectation* for gradient-like guidance in discrete embedding space, and *anchored remasking* for adaptive decoding. APS achieves state-of-the-art performance among discrete diffusion samplers on both linear and nonlinear inverse problems across the standard image benchmarks. We demonstrate the generality of APS through training-free stylization and text-guided editing. We further apply APS to a large-scale diffusion language model, showing consistent improvement in question answering.

## 1. Introduction

Diffusion models have become the state-of-the-art across a wide range of generative tasks, including images (Ramesh et al., 2021; Rombach et al., 2022; Baldridge et al., 2024; Esser et al., 2024; Black Forest Labs, 2024), audio (Huang et al., 2023; Veo, 2025), and video (Singer et al., 2023; OpenAI, 2024; Veo, 2025). Most of this progress has been driven by *continuous* diffusion models, where Gaussian noise is gradually added in pixel or latent space and then

reversed by a learned denoiser (Sohl-Dickstein et al., 2015; Ho et al., 2020). Recently, however, *discrete* diffusion has emerged as a powerful alternative, showing superior performance in modeling categorical distributions such as text (Lou et al., 2024; Sahoo et al., 2024; Shi et al., 2024; Nie et al., 2025b; Rout et al., 2025a) and images (Shi et al., 2024; Yang et al., 2025b). Discrete diffusion further enables a unified framework for both image and text generation, supporting multimodal generation and editing.

Beyond unification, discrete diffusion offers several advantages over continuous diffusion that are particularly relevant for posterior sampling. First, it achieves *faster inference*, often generating high-quality samples in significantly fewer reverse steps (Shi et al., 2024; Schiff et al., 2025; Ma et al., 2025). Second, it provides *finer control*: the model predicts a normalized categorical distribution per token (e.g., a pixel or a patch), which decouples different parts of the image, unlike Gaussian diffusion where the entire image is coupled. Third, it enables *training-free posterior sampling*: since the model outputs full conditional distributions at each step, these can be reweighted by the likelihood to yield a better posterior estimate (Murata et al., 2024; Chu et al., 2025). This property unlocks precise image editing and inverse problem solving without additional training (§4), motivating the use of discrete diffusion as a prior.

State-of-the-art posterior samplers (Rout et al., 2024; Chung et al., 2024; Zhang et al., 2025) use continuous diffusion as a prior. These approaches rely on guiding the reverse diffusion process using likelihood gradients in continuous latent spaces (Rout et al., 2023; 2024; Chung et al., 2024; Zhang et al., 2025). This is infeasible for *discrete* diffusion due to non-differentiability in token space. Derivative-free discrete methods (Li et al., 2024) inspired by reinforcement learning provide weak guidance. G2D2 (Murata et al., 2024) uses Gumbel-Softmax relaxation but it is limited to discrete tokens with continuous embeddings. SGDD (Chu et al., 2025) introduces a split Gibbs sampler, but suffers from exponential complexity in sequence length (Chewi, 2023). These methods unmask tokens in *random order*, which is suboptimal compared to adaptive decoding strategies in language modeling (Yang et al., 2025b; Rout et al., 2025a). These limitations underscore the need for a discrete diffusion posterior sampler with adaptive decoding, leveraging

---

[1] The University of Texas at Austin [2] Google. Correspondence to: Litu Rout <litu.rout@utexas.edu>.

*Proceedings of the $43^{rd}$ International Conference on Machine Learning*, Seoul, South Korea. PMLR 306, 2026. Copyright 2026 by the author(s).

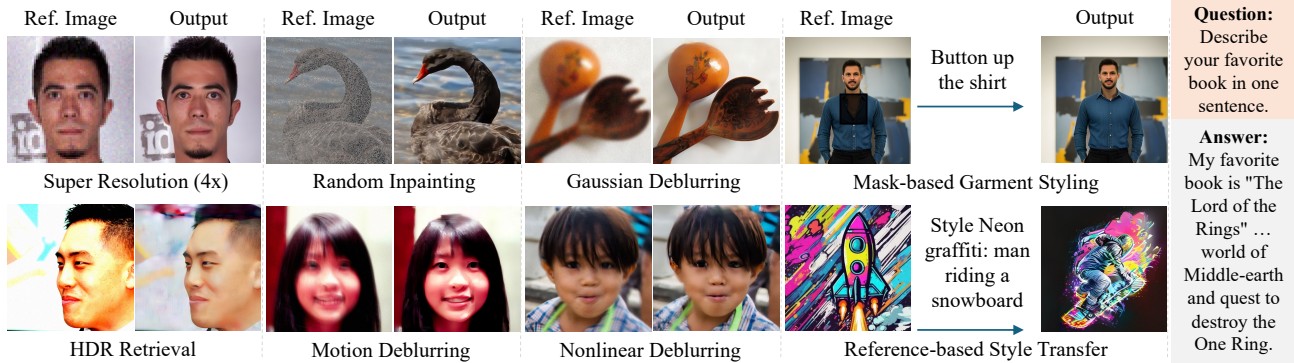

*Figure 1.* We introduce Anchored Posterior Sampling (APS), built on two key innovations: (i) *quantized expectation*, which provides gradient-like guidance in discrete embedding space, and (ii) *anchored remasking*, which enables adaptive decoding. APS supports a variety of linear and nonlinear restoration tasks (left three columns), as well as garment styling and style transfer (fourth column). Furthermore, APS improves question answering by guiding a 7B-parameter diffusion language model without additional training (last column).

next-generation multimodal models (Yang et al., 2025b; Gemini Team, 2024).

In this work, we take the first step towards leveraging multimodal discrete diffusion models (Yang et al., 2025b; Ye et al., 2025) for posterior sampling. We introduce two key algorithmic innovations: (i) *quantized expectation*, which provides gradient-like guidance in purely discrete embedding space by updating the full conditional probability table (§3.1), and (ii) *anchored remasking*, an adaptive decoding strategy that unmasks important "anchor" tokens early during inference (§3.2). Together, these techniques enable discrete diffusion posterior sampling. Like classifier-free guidance (Ho & Salimans, 2022) and Tweedie-based posterior sampling (Chung et al., 2023; Rout et al., 2023) in continuous diffusion, our approach sacrifices asymptotic correctness in favor of scalable posterior inference. As illustrated in Figure 1, we show up to 35.82% LPIPS and 10.94% PSNR improvements on linear and nonlinear inverse problems including training-free stylization (§4) . We further demonstrate generalization of our algorithm to diffusion language models achieving up to 21.99% gain.

**Our contributions are summarized below.**

- **Theoretical results:** we derive (i) a *training* upper bound, $\mathcal{L}_{\text{DDPS}}$ (**Theorem 3.1**), that integrates measurements into the reverse diffusion process, and (ii) a *test-time* bound, $\mathcal{L}_{\text{APS}}$ (**Theorem 3.3**), that reuses a pretrained denoiser without expensive retraining per downstream task (§3).
- **Quantized expectation:** a novel strategy to update *all* entries of the conditional probability table, enabling strong *gradient-like* guidance in discrete diffusion (§3.1).
- **Anchored remasking:** an adaptive decoding strategy to unmask "anchor" tokens early that better utilizes the model's capacity to decode remaining tokens (§3.2).
- **Extensive evaluation** on *linear* (super resolution, Gaussian deblurring, inpainting, motion deblur) and *nonlinear* (HDR, nonlinear deblurring) inverse problems on FFHQ

and ImageNet. We further demonstrate *training-free stylization* and *diffusion language model guidance*, highlighting flexibility beyond typical inverse problems (§4).

## 2. Preliminaries

**Visual Tokenizer.** A cornerstone of modern image tokenization is the VQ-VAE (Van Den Oord et al., 2017), which maps images into discrete codebook indices. It consists of an encoder $\mathcal{E} : \mathbb{R}^{H \times W \times 3} \to \mathbb{R}^{h \times w \times d}$ that projects an image into a latent embedding of size $h \times w \times d$. The embeddings are reshaped into a sequence $\mathbf{e} \in \mathbb{R}^{L \times d}$ of length $L = h \times w$. We denote images by $\hat{\mathbf{x}}$. The encoder produces $\mathbf{e} = \mathcal{E}(\hat{\mathbf{x}})$ where each embedding $\mathbf{e}^l$ for $l = 1, \dots, L$ is quantized to the nearest codebook entry $\mathbf{c}_j \in \mathcal{C}$, where $\mathcal{C} \in \mathbb{R}^{K \times d}$ is a learned codebook:

$$\mathcal{Q}_{\text{vq}}(\mathbf{e}^l) \coloneqq \mathbf{c}_j, \quad j = \underset{k \in \{1, \dots, K\}}{\arg\min} \left\| \mathbf{e}^l - \mathbf{c}_k \right\|_2 . \quad (1)$$

Codebook entries may be continuous vectors $\mathbf{c}_j \in \mathbb{R}^d$ or discrete binary embeddings $\mathbf{c}_j \in \{-1, +1\}^d$ used in this paper. Binary embeddings are particularly appealing because prior studies (Yu et al., 2024) show that masked diffusion models degrade in generation quality as continuous vocabulary size grows, yet lookup-free quantization (LFQ) with binary embeddings achieves both strong generation and reconstruction quality. In LFQ, the codebook is not learned but obtained by thresholding the encoder output: $\mathcal{Q}_{\text{lfq}}(\mathbf{e}^l) \coloneqq \text{sign}(\mathbf{e}^l)$,

$$[\mathcal{Q}_{\text{lfq}}(\mathbf{e}^l)]_i = \begin{cases} +1 & \text{if } \mathbf{e}^l[i] > 0, \\ -1 & \text{otherwise.} \end{cases} \quad (2)$$

Finally, each image is represented as a sequence of tokens $x = (x^1, \dots, x^L)$ corresponding to the selected indices from (1) as $x^l = j$, or equivalently for LFQ the token index is obtained by $x^l = j = \sum_{i=1}^d 2^{i-1} \mathbf{1}_{\{\mathbf{e}_l[i] > 0\}}$. Equivalently, we represent each sequence as a mixture of one-hot vectors $\mathbf{x} = (\mathbf{x}^1, \dots, \mathbf{x}^L)$, where $\sum_{k=1}^K \mathbf{x}^l[k] = 1, \mathbf{x}^l[k] \geq 0$ and

$\mathbf{x}^l[x^l] = 1$. This discrete token representation forms the basis of our masked diffusion posterior sampler.

*Notation:* We use '$:=$' to indicate architectural and parameterization choices, to distinguish from '$=$' which is an identity that follows from mathematical derivations.

**Masked diffusion** defines a generative model over the discrete state space $\mathcal{S} = \mathcal{V}^L$, where $\mathcal{V} = \{1, \ldots, K, K+1\}$ consisting of $K$ codebook indices and a special [MASK] token $\mathbf{m}$ corresponding to index $K+1$. Let $X = \{x^l\}_{l=1}^{L} \in \mathcal{S}$ denote a sequence of tokens (or equivalently its one-hot representation $\mathbf{x} = \{\mathbf{x}^l\}_{l=1}^{L}$). The data distribution is denoted by $q(\cdot)$ over $\mathcal{S}$. The goal of masked diffusion is to learn a generative model that samples from $q(\cdot)$. Masked diffusion models (MDMs) (Austin et al., 2021; Lou et al., 2024; Sahoo et al., 2024; Shi et al., 2024) construct a discrete-time Markov chain with $T$ steps, parameterized by $\alpha_t$ with $t \in [0, 1]$. The forward process gradually replaces each token with the [MASK] token: $q(\mathbf{z}_t|\mathbf{x}) = \prod_{l=1}^{L} q(\mathbf{z}_t^l|\mathbf{x})$,

$$q(\mathbf{z}_t^l|\mathbf{x}) = \text{Cat}\big(\mathbf{z}_t^l; \alpha_t \mathbf{x}^l + (1 - \alpha_t)\mathbf{m}\big), \quad (3)$$

where $\mathbf{z}_t^l$ is either preserved from $\mathbf{x}^l$ with probability $\alpha_t$ or replaced with $\mathbf{m}$. The corresponding reverse process is parameterized by a neural network $p_\theta$, which predicts categorical distributions over tokens. For each token position $l$, the transition probability is $p_\theta(\mathbf{z}_s^l|\mathbf{z}_t) := q(\mathbf{z}_s^l|\mathbf{z}_t^l, \mathbf{x}_\theta(\mathbf{z}_t)) =$

$$\begin{cases} \text{Cat}(\mathbf{z}_s^l; \mathbf{z}_t^l), & \mathbf{z}_t^l \neq \mathbf{m}, \\ \text{Cat}\big(\mathbf{z}_s^l; \frac{\alpha_s - \alpha_t}{1 - \alpha_t}\mathbf{x}_\theta(\mathbf{z}_t) + \frac{1 - \alpha_s}{1 - \alpha_t}\mathbf{m}\big), & \mathbf{z}_t^l = \mathbf{m}, \end{cases} \quad (4)$$

where $\mathbf{x}_\theta(\mathbf{z}_t)$ denotes the network prediction. Training minimizes the negative evidence lower bound (NELBO) by aligning the reverse transition $p_\theta(\mathbf{z}_s^l|\mathbf{z}_t)$ with the inference posterior $q(\mathbf{z}_s^l|\mathbf{z}_t^l, \mathbf{x})$ derived from (3). Concretely, the training objective $\mathcal{L}_{\text{NELBO}}(\mathbf{x}; \theta) := \mathbb{E}_{Z_0 \sim q(\cdot|\mathbf{x})}\big[-\log p_\theta(\mathbf{x}|Z_0)\big] +$

$$\sum_{i=1}^{T} \mathbb{E}_{Z_t \sim q(\cdot|\mathbf{x})}\Big[\frac{\alpha_t - \alpha_s}{1 - \alpha_t}\sum_{l=1}^{L}\log\langle\mathbf{x}_\theta^l(Z_t), \mathbf{x}^l\rangle\mathbf{1}_{\{Z_t^l = \mathbf{m}\}}\Big] \quad (5)$$

where, for brevity, we drop $i$ from $t(i) = i/T$ and $s(i) = (i-1)/T$. The objective (5) admits a score-based interpretation (Lou et al., 2024) and supports time-independent parameterization (Ou et al., 2025), which simplifies training and improves scalability (Nie et al., 2025a;b).

# 3. Test-Time Anchored Posterior Sampling

In posterior sampling, our goal is to construct a Markov chain to sample from the posterior: $q(\mathbf{x}|\mathbf{y}) \propto q(\mathbf{y}|\mathbf{x})\,q(\mathbf{x})$, where $\mathbf{y} = \mathcal{A}(\mathcal{D}(\mathbf{x})) + \sigma\varepsilon$ with measurement operator $\mathcal{A}(\cdot)$, image decoder $\mathcal{D}(\cdot)$, Gaussian noise $\varepsilon \sim \mathcal{N}(0, I)$, and standard deviation $\sigma$. When $\mathcal{A}$ is linear the task reduces to a *linear inverse problem*; otherwise a *nonlinear inverse*

*problem*. We approximate $q(\mathbf{x}|\mathbf{y})$ with a tractable sampler $p_\varphi(\mathbf{x}|\mathbf{y})$ using a masked diffusion model $p_\theta(\mathbf{x})$ previously trained to approximate $q(\mathbf{x})$.

To sample from the posterior $q(\cdot|\mathbf{y})$, we construct a Markov chain with the joint distribution defined as: $p_\varphi(\mathbf{x}, \mathbf{z}_{0:1}|\mathbf{y}) = p_\varphi(\mathbf{z}_1|\mathbf{y})\,p_\varphi(\mathbf{x}|\mathbf{z}_0, \mathbf{y})\prod_{i=1}^{T}p_\varphi\big(\mathbf{z}_{s(i)}|\mathbf{z}_{t(i)}, \mathbf{y}\big)$, where $\mathbf{z}_{0:1} = \mathbf{z}_0, \mathbf{z}_{1/T}, \ldots, \mathbf{z}_1$. We parameterize measurement conditional transitions by tilting the unconditional transition (4) with the likelihood of measurements given the current estimate: $p_\varphi(\mathbf{z}_s|\mathbf{z}_t, \mathbf{y}) := \prod_{l=1}^{L} p_\varphi(\mathbf{z}_s^l|\mathbf{z}_t, \mathbf{y})$, where

$$p_\varphi(\mathbf{z}_s^l|\mathbf{z}_t, \mathbf{y}) \propto q\big(\mathbf{z}_s^l|\mathbf{z}_t^l, \mathbf{x}_\varphi(\mathbf{z}_t)\big)\,q\big(\mathbf{y}|\mathbf{x}_\varphi(\mathbf{z}_t; \mathbf{z}_s^l)\big). \quad (6)$$

We can compute the measurement likelihood $q(\mathbf{y}|\mathbf{x})$ given a sequence of tokens $\mathbf{x} = (\mathbf{x}^1, \mathbf{x}^2, \ldots, \mathbf{x}^L)$, where each $\mathbf{x}^l$ is a one-hot vector. During inference, we have access to the model output $\mathbf{x}_\varphi(\mathbf{z}_t)$ rather than the ground-truth sequence $\mathbf{x}$. We define the log likelihood given the model's output as $\log q(\mathbf{y}|\mathbf{x}_\varphi(\mathbf{z}_t)) := \mathbb{E}_{\mathbf{x} \sim \mathbf{x}_\varphi(\mathbf{z}_t)}[\log q(\mathbf{y}|\mathbf{x})]$. Similarly, $\log q(\mathbf{y}|\mathbf{x}_\varphi(\mathbf{z}_t; \mathbf{z}_s^l))$ denotes the same log likelihood, but with $\mathbf{x}^l$ replaced by $\mathbf{z}_s^l$ while keeping all other positions drawn from $\mathbf{x}_\varphi(\mathbf{z}_t)$. We denote the normalizing factor in (6) by $\mathcal{Z}_\varphi^l(\mathbf{z}_t, \mathbf{y})$.

**Theorem 3.1** (Discrete Diffusion Posterior Sampling (DDPS)). *Given a sample $\mathbf{x} \sim q$, let $q(Z_{0:1}|\mathbf{x})$ denote the forward noising law of (3). Then, for any measurement $\mathbf{y} \sim q(\cdot|\mathbf{x})$, $-\log p_\varphi(\mathbf{x}|\mathbf{y}) \leq \mathcal{L}_{\text{DDPS}}(\mathbf{x}, \mathbf{y}; \varphi) := \mathbb{E}_{q(Z_0|\mathbf{x})}[-\log p_\varphi(\mathbf{x}|Z_0)] + \sum_{i=1}^{T}\mathbb{E}_{q(Z_t|\mathbf{x})}\big[\sum_{l=1}^{L}\log\mathcal{Z}_\varphi^l(Z_t, \mathbf{y})\big] +*

$$\sum_{i=1}^{T}\mathbb{E}_{q(Z_t|\mathbf{x})}\Big[\frac{\alpha_t - \alpha_s}{1 - \alpha_t}\sum_{l=1}^{L}\log\langle\mathbf{x}_\varphi^l(Z_t), \mathbf{x}^l\rangle\mathbf{1}_{\{Z_t^l = \mathbf{m}\}}$$

$$-\mathbb{E}_{q(Z_s|Z_t, \mathbf{x})}\Big[\sum_{l=1}^{L}\log q(\mathbf{y}|\mathbf{x}_\varphi(Z_t; Z_s^l))\Big]\Big].$$

*Implication* 3.2. **Theorem 3.1** shows that $\mathcal{L}_{\text{DDPS}}(\mathbf{x}, \mathbf{y}; \varphi)$ is a principled training criterion for discrete posterior samplers. The likelihood-based tilt $\log q(\mathbf{y}|\mathbf{x}_\varphi(Z_{t(i)}; Z_s))$ enforces measurement consistency. When $\mathbf{y}$ is absent, the tilting term vanishes and the objective reduces to the standard masked diffusion NELBO (5). We can treat $\mathcal{Z}_\varphi$ as a *stop-gradient term*—used to normalize the probability distribution (e.g., via Softmax) but excluded during gradient computation (Sohl-Dickstein et al., 2015; Dhariwal & Nichol, 2021; Ho & Salimans, 2022; Murata et al., 2024). Although expensive, a more formal treatment for the gradient of the log partition function can be obtained through importance sampling (Kahn, 1950), Hamiltonian Monte Carlo with leapfrog steps (MacKay, 2003), or noise contrastive estimation (Gutmann & Hyvärinen, 2010). The cross-entropy term $\log\langle\mathbf{x}_\varphi^l(Z_t), \mathbf{x}^l\rangle$ gets supervision through the masked tokens, with weights determined by the noise schedule.

For retraining, one can minimize $\mathcal{L}_{\text{DDPS}}(\mathbf{x}, \mathbf{y}; \varphi)$ with respect to $\varphi$ to obtain a discrete posterior sampler. In practice,

*Table 1.* **Quantitative results on Super Resolution (4×) and Gaussian Deblurring.** APS consistently outperforms prior discrete samplers (G2D2, SGDD, and SVDD-PM) and remains competitive with strong continuous diffusion baselines (shaded gray).

| | | (a) FFHQ | | | | (b) ImageNet | | | |
|---|---|---|---|---|---|---|---|---|---|
| | | SR (4×) | | Deblur | | SR (4×) | | Deblur | |
| Type | Method | LPIPS ↓ | PSNR ↑ | LPIPS ↓ | PSNR ↑ | LPIPS ↓ | PSNR ↑ | LPIPS ↓ | PSNR ↑ |
| Pixel | DPS | 0.269 | 25.86 | 0.219 | 25.87 | 0.367 | 22.61 | 0.443 | 19.04 |
| | DDRM | 0.282 | 26.58 | 0.239 | 24.93 | 0.352 | 24.00 | 0.246 | 27.30 |
| | DiffPIR | 0.260 | 26.64 | 0.236 | 27.36 | 0.371 | 23.18 | 0.355 | 22.80 |
| | DAPS | 0.177 | 29.07 | 0.165 | 29.19 | 0.276 | 25.89 | 0.253 | 26.15 |
| Latent | PSLD | 0.276 | 27.62 | 0.304 | 27.37 | 0.332 | 24.43 | 0.365 | 24.04 |
| | ReSample | 0.507 | 22.98 | 0.329 | 25.69 | 0.382 | 22.63 | 0.438 | 22.32 |
| | LatentDAPS | 0.182 | 27.48 | 0.234 | 27.93 | 0.276 | 25.06 | 0.345 | 25.05 |
| Uniform/Mask | SVDD-PM | 0.594 | 12.08 | – | – | – | – | – | – |
| | G2D2 | 0.271 | 26.93 | 0.287 | 26.35 | 0.349 | 23.20 | 0.375 | 22.71 |
| | SGDD | 0.288 | 25.85 | – | – | – | – | – | – |
| Mask | **APS** | **0.234** | **27.50** | **0.276** | **27.90** | **0.324** | **24.30** | **0.375** | **24.71** |
| | **APS-L** | **0.186** | **28.83** | **0.241** | **29.50** | **0.224** | **25.74** | **0.282** | **26.35** |

however, retraining a large-scale foundation model per task is often expensive due to excessive compute and lack of training data. We therefore focus on the training-free case.

**Theorem 3.3** (Test-time Anchored Posterior Sampling (APS)). *Given a sample* $\mathbf{x} \sim q$, *let* $q(Z_{0:1}|\mathbf{x})$ *denote the forward noising law of ($3$). Suppose the pretrained network* $p_\theta(\mathbf{x})$ *closely approximates the unconditional prior* $q(\mathbf{x})$. *Then, for any measurement* $\mathbf{y} \sim q(\cdot|\mathbf{x})$, *the negative log-posterior* $-\log p_\varphi(\mathbf{x}|\mathbf{y}) \leq \mathcal{L}_{\text{APS}}(\mathbf{x}, \mathbf{y}; \varphi) :=$
$\mathcal{L}_{\text{NELBO}}(\mathbf{x}; \theta) + \sum_{i=1}^{T} \mathbb{E}_{q(Z_t|\mathbf{x})} \Big[ \sum_{l=1}^{L} \log \mathcal{Z}_\varphi^l(Z_t, \mathbf{y}) \Big] +$

$\sum_{i=1}^{T} \mathbb{E}_{q(Z_t|\mathbf{x})} \Bigg[ \frac{\alpha_s - \alpha_t}{1 - \alpha_t} \sum_{l=1}^{L} \log \frac{\langle \mathbf{x}_\theta^l(Z_t), \mathbf{x}^l \rangle}{\langle \mathbf{x}_\varphi^l(Z_t), \mathbf{x}^l \rangle} \mathbf{1}_{\{Z_t^l = \mathbf{m}\}}$

$-\mathbb{E}_{q(Z_s|Z_t, \mathbf{x})} \Big[ \sum_{l=1}^{L} \log q(\mathbf{y}|\mathbf{x}_\varphi(Z_t; Z_s^l)) \Big] \Bigg]$.

*Implication* 3.4. **Theorem 3.3** shows posterior sampling *without additional training* by reusing a pretrained model.

- *Efficient test-time training.* The $\mathcal{L}_{\text{APS}}(\mathbf{x}, \mathbf{y}; \varphi)$ bound is expressed in terms of NELBO ($5$). Since this term is constant with respect to the new parameters $\varphi$, it can be ignored during optimization. The normalizing constant is treated as in Implication $3.2$. As a result, test-time training only needs to update the lightweight adaptation and measurement-consistency terms, while reusing the fixed pretrained network $\mathbf{x}_\theta(\cdot)$. This avoids backpropagation through the large denoiser (e.g., billions of parameters), making posterior sampling efficient at test time.
- *Training-free inference.* Although test-time training requires paired $(\mathbf{x}, \mathbf{y})$ data, in posterior sampling we only observe $\mathbf{y}$. Importantly, the bound $\mathcal{L}_{\text{APS}}$ motivates a *surrogate optimization objective* rather than an exact sampling procedure (see §3.1), placing APS in the class of variationally guided diffusion samplers rather than asymptotically unbiased MCMC methods.

- *Adaptation gap.* The log-ratio terms capture the mismatch between unconditional predictions $\mathbf{x}_\theta(Z_{t(i)})$ and adapted posterior $\mathbf{x}_\varphi(Z_{t(i)})$, active only at masked positions.
- *Measurement consistency.* The final summation ensures sample consistency with the observed measurements $\mathbf{y}$.

Next, we introduce two approximations: *Quantized Expectation* (§3.1) and *Anchored Remasking* (§3.2) that make our approach practically implementable. These two ideas together form our **Algorithm 1: Anchored Posterior Sampling (APS)**; please see Appendix C.2 for a detailed discussion.

### 3.1. Quantized Expectation

Following the standard practice in test-time optimization (Chung et al., 2023; Rout et al., 2023; 2024; Chung et al., 2024; Rout et al., 2025c; Zhang et al., 2025; Murata et al., 2024), the upper bound in **Theorem 3.3** is optimized at every time step $t$. We ignore the normalizing factor during optimization as discussed in Implication 3.2 and renormalize the optimized logits via Softmax to construct a valid distribution. Since $\mathcal{L}_{\text{NELBO}}(\mathbf{x}; \theta)$ does not depend upon $\varphi$, we obtain the following objective per time step:

$$\mathcal{L}_t(\varphi) := \frac{\alpha_s - \alpha_t}{1 - \alpha_t} \sum_{l=1}^{L} \Big[ \log \frac{\langle \mathbf{x}_\theta^l(\mathbf{z}_t), \mathbf{x}^l \rangle}{\langle \mathbf{x}_\varphi^l(\mathbf{z}_t), \mathbf{x}^l \rangle} \mathbf{1}_{\{\mathbf{z}_t^l = \mathbf{m}\}} \Big]$$
$$- \mathbb{E}_{q(Z_s|\mathbf{z}_t, \mathbf{x})} \Big[ \sum_{l=1}^{L} \log q(\mathbf{y}|\mathbf{x}_\varphi(\mathbf{z}_t; Z_s^l)) \Big]. \quad (7)$$

Equation ($7$) remains intractable because the first term depends on $\mathbf{x}^l$ which is not known in posterior sampling and the second term requires $\mathcal{O}(K^L)$ compute[1]. We address the first term by replacing $\mathbf{x}^l$ with $\mathbf{x}_\theta^l(\mathbf{z}_t)$ because the pre-

---

[1]Recall that $\log q(\mathbf{y}|\mathbf{x}_\varphi(\mathbf{z}_t; Z_s^l))$ within the outer expectation of ($7$) is defined by an inner expectation in ($6$). The marginaliza-

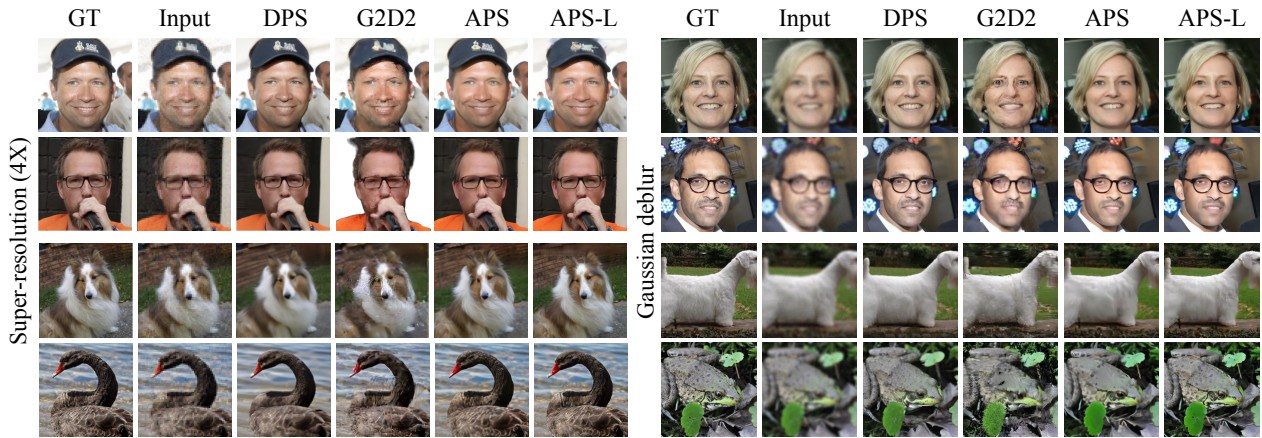

*Figure 2.* **Qualitative results on FFHQ and ImageNet for SR** $(4\times)$ **and Gaussian deblur.** Compared to DPS and G2D2, APS yields better results with sharper texture and refined facial features. For instance, in the third row, APS reconstructs fine strands of the dog's fur.

trained model provides a reasonable approximation. For the second, since $q(\mathbf{y}|\mathbf{x}_\varphi(\mathbf{z}_t; Z_s^l))$ involves an expectation over a parameterized distribution, one could in principle, optimize $\mathcal{L}_t(\varphi)$ using policy-gradient methods such as RE-INFORCE (Williams, 1992), PPO (Schulman et al., 2017), or GRPO (Shao et al., 2024), as adopted in SVDD (Li et al., 2024). This leads to high-variance gradients and sparse rewards, resulting in poor sample quality (see: Table 1).

We introduce *quantized expectation* to better approximate the second term in (7). Recall that the measurement likelihood is modeled as $q(\mathbf{y}|\mathbf{x}) \propto \exp(-\frac{\|\mathbf{y}-\mathcal{A}(\mathcal{D}(\mathbf{x}))\|_2^2}{2\sigma^2})$. Sampling $\mathbf{x} \sim \mathbf{x}_\varphi(\mathbf{z}_t)$ to compute likelihood makes $\mathcal{L}_t(\varphi)$ non-differentiable and noisy. Instead, we propose a differentiable and practically implementable surrogate[2]:

$$\hat{\mathcal{L}}_t(\varphi) \coloneqq \frac{\alpha_t - \alpha_s}{1 - \alpha_t} \sum_{l=1}^{L} \Big[ \log\langle \mathbf{x}_\varphi^l(\mathbf{z}_t), \mathbf{x}_\theta^l(\mathbf{z}_t)\rangle \, \mathbf{1}_{\{\mathbf{z}_t^l = \mathbf{m}\}} \Big]$$
$$- L \cdot \log q(\mathbf{y}|\tilde{\mathbf{x}}_\varphi(\mathbf{z}_t))\Big]. \quad (8)$$

Appendix C.4.1 includes the derivation of (8) from (7). Differentiability has propelled posterior sampling to achieve state-of-the-art results using continuous diffusion (Chung et al., 2023; Rout et al., 2023; 2024; Zhang et al., 2025). To restore differentiability in discrete diffusion, we parameterize $\mathbf{x}_\varphi(\mathbf{z}_{t(i)}) = \text{Softmax}(\varphi_{t(i)})$ with $\varphi_{t(i)} = \{\varphi_{t(i)}^l\}_{l=1}^L \in \mathbb{R}^{K \times L}$ containing logits $\varphi_{t(i)}^l$ over the codebook $\{\mathbf{c}_k \in \mathcal{C}\}$ for each position $l$. We compute the expected embedding $\bar{\mathbf{x}}^l = \sum_{k=1}^{K} \mathbf{c}_k \cdot \mathbf{x}_\varphi^l(\mathbf{z}_{t(i)})[k] \in \mathbb{R}^d$, and then *quantize* it using LFQ (2) (Yu et al., 2024): $\mathbf{x}^l = \mathcal{Q}_{\text{lfq}}(\bar{\mathbf{x}}^l) \in \{-1, +1\}^d$. We then apply the straight-through estimator (Van Den Oord

---

tion over $Z_s$ in the outer expectation costs $\mathcal{O}(2^L)$ (given $\mathbf{z}_t$ and $\mathbf{x}$, $Z_s \in \{\mathbf{x}, \mathbf{m}\}$). For each $Z_s$, the computation of the inner expectation (over random variables except $Z_s^l$) inside the summation costs $\mathcal{O}(LK^{L-1})$. In the worst case, the total cost becomes $\mathcal{O}(K^L)$.

[2]Note that $\alpha_t - \alpha_s < 0$; minimizing (8) is equivalent to maximizing the correlation between the prior and the tilted posterior.

et al., 2017) to obtain an image $\hat{\mathbf{x}} = \mathcal{D}(\tilde{\mathbf{x}}_\varphi(\mathbf{z}_t))$, $\tilde{\mathbf{x}}_\varphi(\mathbf{z}_t) = \bar{\mathbf{x}} + [\mathbf{x} - \bar{\mathbf{x}}]_{\text{sg}}$, where sg denotes the stop-gradient operator. Finally, we compute the differentiable likelihood as $q(\mathbf{y}|\tilde{\mathbf{x}}_\varphi(\mathbf{z}_t)) \propto \exp(-\frac{\|\mathbf{y}-\mathcal{A}(\hat{\mathbf{x}})\|_2^2}{2\sigma^2})$.

**Discussion.** We minimize (8) at each time step $t$ to obtain the optimal logits $\varphi_t^*$. This ensures that the optimized probabilities $\mathbf{x}_{\varphi^*}(\mathbf{z}_t)$ remain close to the prior predictions $\mathbf{x}_\theta(\mathbf{z}_t)$ while aligning with measurements $\mathbf{y}$. To our knowledge, this is the first use of *quantized expectation of codebook embeddings* for posterior sampling in discrete diffusion, allowing measurement gradients to backpropagate through $\bar{\mathbf{x}}$ and update all entries of $\mathbf{x}_\varphi(\mathbf{z}_t)$. This reduces the cost of the measurement term from $\mathcal{O}(K^L)$ to $\mathcal{O}(KL)$ because (8) requires only *one* forward pass through $\mathcal{A} \circ \mathcal{D}$ using $\tilde{\mathbf{x}}_\varphi(\mathbf{z}_t)$. We defer complexity analysis to Appendix C.4.

### 3.2. Anchored Remasking

In masked diffusion, the reverse process progressively unmasks tokens. Typical samplers (Austin et al., 2021; Chang et al., 2022; Lou et al., 2024; Sahoo et al., 2024; Shi et al., 2024) choose to unmask based on confidence or random remasking. ADLM (Rout et al., 2025a) shows that prioritizing "anchor" tokens (e.g., nouns or verbs in language, rather than articles or conjunctions) reduces conditional entropy of the remaining tokens in the sequence that subsequently improves generation. To decode anchor tokens, ADLM jointly trains an anchor network in addition to the standard denoising network using an anchored NELBO objective. Here, we propose a *training-free* variant of anchored denoising, enabling posterior sampling with discrete diffusion.

Let $\varphi$ be a minimizer of (8), resulting in $\mathbf{x}_\varphi^*(\mathbf{z}_t) = \{(\mathbf{x}_\varphi^*(\mathbf{z}_t))^l\}_{l=1}^L$ that represents categorical distributions over tokens at each position $l \in \{1, \ldots, L\}$ at time step $t$. Anchored remasking selects a subset of positions $\mathcal{P}_t \subseteq \{1, \ldots, L\}$ to decode early. The selection is based on the confidence of quantized tokens $\mathbf{x}$ (as defined in §3.1) under

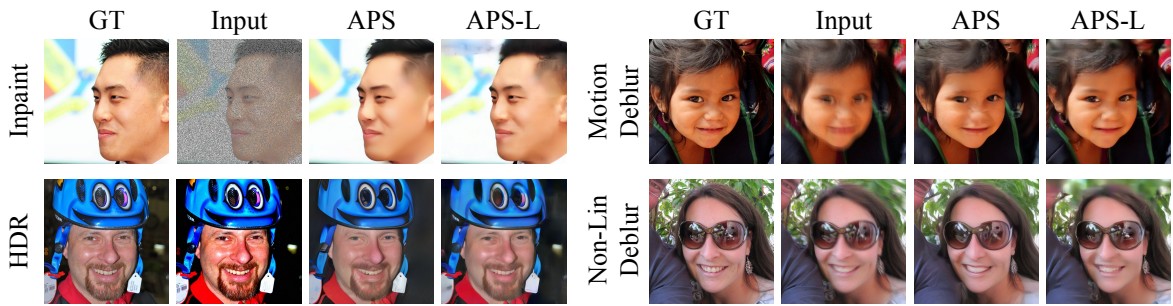

*Figure 3.* **Qualitative results on FFHQ** for linear (top row) and nonlinear (bottom row) inverse problems. APS and APS-L recover high-fidelity images from severely degraded inputs.

the posterior estimate $\mathbf{x}_\varphi^*(\mathbf{z}_t)$. Importantly, the posterior estimate is a function of the $L$-length sequence $\mathbf{z}_t$, and hence encodes the *joint* relation across all tokens; thus, anchored remasking is a function of *all* tokens, unlike standard remasking (Chang et al., 2022; Yang et al., 2025b) that depend only on per-token confidence. Formally, we compute the confidence of $\mathbf{x}^l$ as $\kappa_t^l = \langle(\mathbf{x}_\varphi^*(\mathbf{z}_t))^l, \mathbf{x}^l\rangle$, and choose anchor positions as $\mathcal{P}_t = \{l : \kappa_t^l \geq \tau_t\}$, where $\tau_t$ is an adaptive threshold that follows from the cosine schedule of MMaDA. We then unmask anchor tokens as: $\mathbf{z}_s^l = \mathbf{x}^l$ with probability $\frac{\alpha_s - \alpha_t}{1 - \alpha_t}$ if $l \in \mathcal{P}_t$ else $\mathbf{m}$. Once a token is unmasked, it stays unmasked during inference.

**Discussion.** Diffusion language models tend to be highly confident on low-informative tokens such as articles ("a", "an" or "the") or conjunctions (Rout et al., 2025a); similarly, discrete image samplers using *independent per-token* confidence often unmask background pixels first. In contrast, our method leverages the *joint posterior* $\mathbf{x}_\varphi(\mathbf{z}_t)$ to identify anchor tokens consistent with the measurements. This leads to earlier decoding of informative tokens (e.g., a bird against a flat background), while aligning with the *joint distribution*; refer to §C.3 for details. While multi-token sampling from the joint is typically intractable for frameworks like MDLM (Sahoo et al., 2024) or MD4 (Shi et al., 2024), anchored diffusion (Rout et al., 2025a) circumvents this limitation by reducing the conditional entropy of the remaining tokens. We follow a similar anchoring strategy for adaptive decoding of discrete image tokens.

## 4. Experiments

**Baselines.** Since our focus is on discrete diffusion, we first compare with existing discrete methods: G2D2 (Murata et al., 2024) and SGDD (Chu et al., 2025). To provide a comprehensive evaluation, we also include established continuous baselines, both in pixel space (DPS (Chung et al., 2023), DDRM (Kawar et al., 2022), and DiffPIR (Zhu et al., 2023)) and in latent space (PSLD (Rout et al., 2023) and ReSample (Song et al., 2024)). Thus, our evaluation spans both discrete and continuous paradigms. For language modeling, we compare with `LLaDA-8B-Instruct` (Nie et al., 2025b) and `Dream-7B-Instruct` (Ye et al., 2025).

**Benchmarks.** We evaluate on the standard inverse problem benchmarks used in prior works. For high-resolution faces we use FFHQ at $256 \times 256$ (Karras et al., 2019), and for diverse natural images we use ImageNet at $256 \times 256$ (Deng et al., 2009). Performance is measured using three standard metrics: Learned Perceptual Image Patch Similarity (Zhang et al., 2018) (LPIPS), Peak Signal-to-Noise Ratio (PSNR), and Structural Similarity Index (SSIM) (Wang et al., 2004). All methods are evaluated at the same resolution and on the same images following G2D2 (Murata et al., 2024). For question answering, we use `Qwen2.5-3B` model (Yang et al., 2025a) to create a benchmark of 20 questions (§ C.7), where evaluation is done using `Qwen3-0.6B` as a reward model and `Qwen3-8B` as an independent judge.

**Tasks.** We consider both linear and nonlinear inverse problems. Following prior works (e.g., G2D2 (Murata et al., 2024) and SGDD (Chu et al., 2025)), we evaluate on Super Resolution (SR) ($4\times$) and Gaussian deblurring on both FFHQ and ImageNet. Additionally, we evaluate on more linear (random inpainting and motion blur) and nonlinear (high dynamic range (HDR) recovery and nonlinear deblurring) tasks. Beyond inverse problems, we conduct experiments on training-free stylization, an emerging area largely dominated by continuous diffusion (Hertz et al., 2023; Wang et al., 2024; Rout et al., 2025c). To demonstrate scalability, we upsample the benchmark images to $1024 \times 1024$ and apply our sampler to these scaled images (termed **APS-L** in discussion below). We show generalization of our algorithm to guide diffusion language models. We defer implementation details to §C.1 and complexity analysis to §C.4.

### 4.1. Results on Linear Inverse Problems

**Evaluation on FFHQ:** Table 1 (a) shows that APS outperforms prior discrete diffusion samplers across both super resolution and Gaussian deblurring tasks. For *super resolution*, APS reduces LPIPS by 13.65% and improves PSNR by 2.11%. On *Gaussian deblurring*, APS lowers LPIPS by 3.83% and raises PSNR by 5.88%. The large variant, APS-L, pushes performance even further, achieving up to a 31.36% gain over G2D2 in terms of LPIPS. These results

*Table 2.* **Quantitative results on general inverse problems.** We report results on two additional linear (random inpainting, motion deblurring) and nonlinear (HDR, nonlinear blur) tasks. Since G2D2 and SGDD do not evaluate on these tasks, we compare our *discrete* sampler against representative *continuous* baselines: DPS (pixel-space) (Chung et al., 2023) and PSLD (latent-space) (Rout et al., 2023).

| Type | Method | Random Inpainting | | Motion Deblur | | HDR | | Nonlinear Blur | |
|------|--------|---------|---------|---------|---------|---------|---------|---------|---------|
| | | LPIPS ↓ | PSNR ↑ | LPIPS ↓ | PSNR ↑ | LPIPS ↓ | PSNR ↑ | LPIPS ↓ | PSNR ↑ |
| FFHQ | | | | | | | | | |
| Pixel | DPS | **0.203** | 25.46 | **0.246** | 24.52 | **0.264** | 22.73 | 0.278 | 23.39 |
| Latent | PSLD | 0.221 | **30.31** | 0.336 | 22.31 | – | – | – | – |
| Discrete | **APS (ours)** | 0.304 | 27.38 | 0.317 | 26.58 | 0.282 | **23.89** | 0.263 | 27.19 |
| | **APS-L (ours)** | 0.291 | 28.11 | 0.298 | **27.98** | 0.323 | 23.56 | **0.262** | **28.46** |
| ImageNet | | | | | | | | | |
| Pixel | DPS | **0.297** | 23.52 | 0.423 | 18.96 | 0.503 | 19.23 | **0.306** | 22.49 |
| Latent | PSLD | 0.337 | **31.30** | 0.511 | 20.85 | – | – | – | – |
| Discrete | **APS (ours)** | 0.378 | 24.59 | 0.410 | 23.37 | 0.345 | 21.92 | 0.330 | 24.18 |
| | **APS-L (ours)** | 0.338 | 25.39 | **0.318** | **25.19** | **0.346** | **22.68** | 0.309 | **25.35** |

**Reference Style**   **Stylized Output**   **Reference Style**   **Stylized Output**

Celestial Artwork   Carousel   Astronaut   Bowl of Fruits   Street Art Graffiti   Truck   Watch   Playmobil

*Figure 4.* **Qualitative results on stylization.** We present four style–content combinations. For each case, our APS algorithm conditions on a single reference style image together with a text prompt to generate the stylized output images.

show that our quantized expectation (§3.1) and anchored remasking (§3.2) strategies not only outperform discrete diffusion baselines but often surpass strong continuous diffusion methods, such as DiffPIR (which uses pixel-space diffusion) and PSLD (which uses latent-space diffusion).

Figure 2 (top two rows) presents qualitative comparisons for super resolution (4×) and Gaussian deblurring on FFHQ. DPS produces over-smoothed results with blurry facial details, while G2D2 often introduces artifacts and fails to restore a natural facial structure. Our APS sampler yields sharper textures and more faithful reconstructions, recovering details such as hair strands, facial contours, and eyeglass edges with higher perceptual quality. The large variant, APS-L, further enhances structure and realism, generating photo-realistic outputs with finer details and fewer artifacts. These results confirm that APS and APS-L deliver superior qualitative performance, where perceptual quality is crucial.

**Evaluation on ImageNet:** Table 1 (b) quantifies that our method achieves consistent improvements across both SR and Gaussian deblurring tasks on the ImageNet benchmark. For *super resolution*, APS reduces LPIPS by 7.16% compared to G2D2 and improves PSNR by 4.74%. On *Gaussian deblurring*, APS improves PSNR by 8.81% while maintaining comparable LPIPS. APS-L improves LPIPS by up to 35.82% compared to G2D2. These results confirm that APS

outperforms prior discrete posterior samplers and often surpasses continuous baselines such as DiffPIR and PSLD with superior perceptual quality and image fidelity.

Figure 2 (bottom two rows) shows SR (4×) and Gaussian deblurring results on ImageNet. Notably, DPS produces overly smooth outputs with a loss of fine details, while G2D2 introduces struggles to recover sharp edges. In contrast, APS reconstructs sharper textures (e.g., the fur of the dog and the feathers of the swan) and yields more natural color. The large variant, APS-L, further enhances structural fidelity, recovering finer details in challenging regions such as a frog's skin texture and a goat's fur. These examples highlight that our approach achieves superior perceptual quality and faithful structure reconstruction compared to both continuous and discrete diffusion baselines.

### 4.2. Results on General Inverse Problems

Table 2 shows that APS generalizes effectively to more challenging linear (random inpainting, motion deblurring) and nonlinear (HDR, nonlinear blur) inverse problems on FFHQ and ImageNet. Unlike existing methods such as G2D2 and SGDD, which were demonstrated on limited tasks, APS achieves strong perceptual quality (lower LPIPS) and reconstruction fidelity (higher PSNR). For instance, on ImageNet motion deblurring APS-L attains 0.318 LPIPS and 25.19

*Table 3.* **Quantitative results on stylization.** APS enables new capabilities such as reference-based stylization in MMaDA.

| | ImageReward ↑ | CLIP-T ↑ | DINO ↑ |
|---|---|---|---|
| IP-Adapter | -1.51 | 0.26 | 0.89 |
| StyleAligned | 0.01 | 0.31 | 0.85 |
| InstantStyle | 0.72 | 0.33 | 0.72 |
| RB-Modulation | 1.18 | 0.34 | 0.73 |
| MMaDA | 0.48 | 0.33 | 0.32 |
| APS (ours) | 0.63 | 0.34 | 0.41 |

PSNR, substantially outperforming continuous baselines DPS and PSLD. Figure 3 shows the qualitative results. In nonlinear tasks such as HDR and nonlinear blur, APS delivers sharper, more consistent reconstructions, closing the gap with continuous diffusion while operating within a purely discrete framework. These results highlight the broader applicability and robustness of our approach compared to existing discrete diffusion samplers.

### 4.3. Results on Reference-based Stylization

We compare APS with the discrete diffusion baseline MMaDA (Yang et al., 2025b) and existing continuous diffusion baselines (shaded gray). Table 3 reports quantitative results, while Figure 4 shows qualitative examples. APS consistently improves over the base MMaDA model across all metrics: ImageReward (Xu et al., 2023), CLIP-T (Radford et al., 2021), and DINO (Caron et al., 2021) demonstrating that our posterior sampler successfully unlocks zero-shot stylization capabilities that are otherwise absent in the base model. By formulating style alignment as a reward maximization problem, APS guides the discrete generation trajectory to capture style attributes from the reference image without requiring auxiliary control networks or fine-tuning. Notably, APS surpasses established continuous baselines such as IP-Adapter (Ye et al., 2023) and StyleAligned (Hertz et al., 2023) on ImageReward and CLIP-T metrics, despite relying on a weaker generative prior.

### 4.4. Results on Question Answering

Although this paper primarily focuses on discrete diffusion for vision tasks, we show that our approach generalizes to language by evaluating it on a question answering task.

Figure 5 qualitatively demonstrates the efficacy of Anchored Posterior Sampling (APS) in diffusion-based question answering. While `Dream-7B-Instruct` suffers from incoherence due to a known artifact of independent sampling from the product of marginals, APS employs a lightweight reward model (`Qwen3-0.6B`) to enable joint sampling. Thus, APS yields structurally consistent and semantically richer outputs, such as better *rhyming* in the poem.

These qualitative gains are further confirmed by our

*Figure 5.* **Qualitative comparison on a creative writing prompt.** APS produces better rhyming and more structurally coherent creative writing compared to the generic outputs of the baseline.

*Table 4.* **Quantitative results on question answering.**

| Method | Reward Model (`Qwen3-0.6B`) | Oracle Judge (`Qwen3-8B`) |
|---|---|---|
| LLaDA | -0.0970 | 2.1699 |
| Dream | 0.9970 | 3.7523 |
| **Dream+APS (Ours)** | **1.3786** | **4.5775** |

*Table 5.* **Extended evaluations on complex constraints and human alignment.** APS consistently outperforms the baseline across larger-scale benchmarks.

| Method | RewardBench 2 | RM-Bench |
|---|---|---|
| Baseline | 1.7145 | 1.8281 |
| **APS (Ours)** | **1.7937** | **1.8690** |

quantitative evaluation using a held-out oracle judge (`Qwen3-8B`) as given in Table 4. APS achieves a **21.99%** relative improvement over the state-of-the-art `Dream-7B-Instruct` without additional training.

To evaluate the scalability and robustness of our method, we expand our evaluation to two additional benchmarks (100 examples each): RewardBench 2, which tests the model's ability to adhere to complex constraints, and RM-Bench, which measures alignment with human preference scores. As detailed in Table 5, APS consistently outperforms the baseline across these broader settings. Implementation details are provided in Appendix C.1. Implementation details are provided in Appendix C.1.

### 4.5. Effect of Optimization Steps

Table 6 presents an ablation study on the number of inner optimization steps used during test-time anchoring in **Algorithm 1**. We observe that performance improves rapidly up to 50 steps and saturates beyond 100 steps, confirming that our method converges efficiently without requiring

*Table 6.* **Effect of Optimization Steps (Super-Resolution 4×) on FFHQ).** Performance saturates beyond 100 optimization steps, confirming robustness and efficiency.

| # Opt. Steps ($M$) | PSNR ↑ | SSIM ↑ | LPIPS ↓ |
|---|---|---|---|
| 0 | 9.49 | 0.1855 | 0.6747 |
| 10 | 21.47 | 0.5541 | 0.4298 |
| 50 | 24.80 | 0.7130 | 0.2730 |
| 100 | 25.71 | 0.7422 | 0.2468 |
| 200 | 25.72 | 0.7389 | 0.2474 |

*Table 7.* **Quantitative results for super resolution (4×) on FFHQ.** APS outperforms G2D2 across standard evaluation metrics: PSNR, SSIM and LPIPS. Both methods have a reconstruction loss and use the same discrete diffusion prior VQ-Diffusion (Gu et al., 2022), which has a VQ-VAE (Van Den Oord et al., 2017) tokenizer as opposed to the LFQ (Yu et al., 2024) tokenizer used in MMaDA (Yang et al., 2025b). The runtime of G2D2 and APS is nearly identical (see Table 9). We tune perceptual coefficient of the baseline to compare against better quality.

| Method | PSNR ↑ | SSIM ↑ | LPIPS ↓ |
|---|---|---|---|
| G2D2 (original) | 25.33 | 0.706 | 0.353 |
| G2D2 (w/ percep. 1e-3) | 25.46 | 0.720 | 0.349 |
| G2D2 (w/ percep. 1e-2) | 25.34 | 0.707 | 0.357 |
| APS (Ours) | **26.80** | **0.759** | **0.310** |

additional iterations. Importantly, all optimization steps backpropagate only through the lightweight VQ-VAE decoder rather than the large 8B-parameter MMaDA backbone, keeping computational overhead minimal.

**4.6. Evaluation Under Identical Discrete Diffusion Prior**

We compare APS to G2D2 using the *official* G2D2 codebase (§C.6) on 100 images from the FFHQ validation set. Both methods use an identical compute budget: 100 reverse diffusion steps, each with 30 inner optimization steps. As shown in Table 7, APS improves over G2D2 across all metrics: PSNR improves from 25.46 to 26.80, SSIM from 0.717 to 0.759, and LPIPS decreases from 0.350 to 0.310. Since the generative prior and compute budget are identical, these performance gains arise purely from our algorithmic innovations: quantized expectation (§3.1), anchored remasking (§3.2), and the use of a perceptual loss. This experiment demonstrates that APS is not only effective but also compatible with both *mask-based* and *uniform* discrete diffusion frameworks.

The appendix provides extensive details on our Anchored Posterior Sampling method. We present the theoretical derivation of variational bounds (§B), all implementation specifics and hyperparameters (Algorithm 1, Table 8), and a detailed ablation study (§C.3, Figure 6) demonstrating the impact of our innovations. We also showcase the superior

computational efficiency (Table 9) and numerous additional qualitative results (§C) across complex inverse problems, high-resolution text-guided block inpainting (Figure 12), and extended stylization results (Figure 13 and 14).

# 5. Conclusion

We introduce **Anchored Posterior Sampling (APS)**, a theoretically grounded, training-free sampler that enables the reuse of pretrained discrete diffusion models without task-specific retraining. APS leverages *Quantized Expectation* to provide gradient-like guidance in discrete spaces and *Anchored Remasking* to adaptively decode informative tokens early in the denoising process. In our primary vision tasks on linear and nonlinear inverse problems, APS achieves state-of-the-art results among discrete samplers and remains competitive with continuous baselines while using significantly reduced inference cost. Furthermore, APS demonstrates versatility by enabling training-free stylization and steering language models toward better question answering. This work establishes discrete diffusion as a scalable posterior sampling alternative, with promising extensions to multimodal generation and editing.

# Acknowledgments

The authors thank the **Google's ARML Commerce team** for their support and for providing a stimulating environment for this research, which was conducted while the first author was an intern at Google. We are also grateful to **Akash Sengupta** and **Yingwei Li** for their insightful discussions during the early stages of this project. This research has been partially supported by NSF Grants 2112471, 2505865 and the UT Austin Machine Learning Lab.

# Impact Statement

Our method enables training-free, controlled image editing using pretrained generative models, which can broaden access to advanced creative tools. As with all generative models, it has dual-use potential and may be misapplied if its limitations are misunderstood.

In inverse problems such as super-resolution, inpainting, and deblurring, APS produces a plausible sample from a posterior distribution rather than a unique or guaranteed reconstruction of the original signal. Misinterpreting such outputs as factual reconstructions could be harmful in sensitive settings, including forensic analysis or identity verification. Accordingly, our method is intended for perceptual enhancement tasks, and is not suitable for applications requiring evidentiary reliability.

For stylization, there is a risk that information from the source content image may leak in the generated output, po-

tentially leading to unintended disclosure of sensitive details. Users should therefore exercise caution when applying the method to private or confidential imagery.

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

# A. Additional Related Works

**Gaussian pixel-space diffusion** methods (Chung et al., 2023; Kawar et al., 2022; Zhu et al., 2023; Song et al., 2023) study inverse problems using diffusion priors trained in the pixel space. These priors are domain-specific, e.g., a model trained on ImageNet must be used for ImageNet tasks, and one trained on FFHQ for FFHQ tasks, resulting in informative but narrow priors. Mixing domains during training convolutes these priors, and at the extreme of internet-scale training, the priors remain valid for generation (Ramesh et al., 2021; Baldridge et al., 2024; Esser et al., 2024; Black Forest Labs, 2024) but become less informative for domain-specific problems. This motivates the challenge of extracting domain-specific priors from general-purpose priors.

**Gaussian latent diffusion.** PSLD (Rout et al., 2023) introduced posterior sampling with latent diffusion, showing how domain-specific priors can be extracted from general-purpose priors, e.g., large-scale foundation models. This line of work (Chung et al., 2024; Song et al., 2024; Rout et al., 2024; 2025b; Noroozi et al., 2024; Zhang et al., 2025; Chung et al., 2025) uses latent diffusion priors: a single pretrained model can handle multiple domains, enabling inverse problems and semantic edits without retraining, while also being faster and more scalable to high-resolution synthesis. A drawback, however, is that posterior sampling often requires backpropagation through large denoisers (e.g., Flux (Black Forest Labs, 2024), SD3.5 (Esser et al., 2024)), which is prohibitively slow. RB-Modulation (Rout et al., 2025c) alleviates this by framing the problem as stochastic optimal control, directly optimizing the terminal latent state and reducing runtime from minutes (PSLD: $\sim$12 min, P2L: $\sim$30 min, STSL: $\sim$3 min) to under 40 seconds. This efficiency relies on continuous, differentiable latent embeddings, a property that does not extend to discrete diffusion. Addressing this gap motivates the need for new posterior sampling approaches in discrete settings.

**Uniform discrete diffusion.** Recent works have explored posterior sampling with discrete diffusion. G2D2 (Murata et al., 2024) extends the proximal sampler of RB-Modulation to VQ-diffusion (Gu et al., 2022) using a star-shaped noising process and Gumbel-Softmax dequantization (Gumbel, 1954; Jang et al., 2017; Maddison et al., 2017). While it enables gradient guidance, G2D2 depends on continuous relaxations, requires storing log-probabilities from previous step, and struggles to generalize to purely discrete token embeddings (§4). SGDD (Chu et al., 2025) instead proposes a split Gibbs sampler with Hamming-distance reweighting and rejection sampling via Metropolis–Hastings, but its exponential rejection rate restricts their results to low-resolution tasks.

**Masked (absorbing) discrete diffusion.** While G2D2 and SGDD can, in principle, be adapted to masked diffusion, they perform poorly with purely discrete token embeddings. In contrast, our method leverages a unified masked discrete diffusion model and introduces two key components: *quantized expectation* (§3.1) and *anchored remasking* (§3.2). Together, these yield an efficient and scalable posterior sampler for high-resolution inverse problems. To our knowledge, this is the first inverse problem solver tailored for masked discrete diffusion with purely discrete embeddings, outperforming prior discrete samplers and remaining competitive—often superior—to continuous diffusion methods at substantially lower inference cost (§4).

**Gradient propagation in discrete domains.** Propagating gradients through discrete variables has been studied extensively, particularly for representation learning and image tokenization. Foundational works such as VQ-VAE (Van Den Oord et al., 2017) and recent advancements in scalable image tokenization (Shi et al., 2025) utilize gradient estimation to build highly optimized codebooks. A cornerstone of these approaches is the straight-through estimator (STE) (Bengio et al., 2013), which we similarly employ to mitigate the computational expense of our exact variational upper bounds (Theorems 3.1 and 3.3), thereby drastically reducing the complexity over the vocabulary size. However, our objectives differ fundamentally from this line of work: rather than focusing on training-time representation learning, our work is strictly dedicated to inference-time guidance. APS is a training-free method that enables posterior sampling using pre-trained discrete backbones and existing tokenizers. We note that exploring how advanced tokenizers (Shi et al., 2025) interact with multi-modal priors under our framework remains an exciting direction for future work.

# B. Additional Theoretical Results

This appendix develops complementary theory for masked discrete diffusion posterior sampling. We first derive a pathwise variational bound for *training* a likelihood-tilted reverse process (**Theorem B.1**), showing that $-\log p_\varphi(\mathbf{x}|\mathbf{y})$ is upper bounded by a reconstruction term, a sum of token-wise KL-divergence terms, and a sequence of measurement likelihood-based tilting terms. We then specialize this analysis to the *training-free* setting where the token-to-image decoder is shared between unconditional generation and posterior sampling, yielding a bound for test-time anchored posterior sampling

(**Theorem B.2**). We provide theoretical insights drawn from each theorem in **Implication** subsections after the corresponding proofs.

**Theorem B.1** (Discrete Diffusion Posterior Sampling(DDPS)). *Let $Z_{0:1} = \{Z_{t(i)}\}_{i=0}^T$ with $t(i) = i/T$ and $s(i) = (i-1)/T$ be the latent path of a masked discrete diffusion model, and let $q(Z_{0:1}|\mathbf{x})$ be the forward noising law from (3). Consider reverse kernels and a terminal decoder that factorize as*

$$p_\varphi(\mathbf{x}, Z_{0:1}|\mathbf{y}) = p_\varphi(Z_1|\mathbf{y})\, p_\varphi(\mathbf{x}|Z_0, \mathbf{y}) \prod_{i=1}^T p_\varphi(Z_{s(i)}|Z_{t(i)}, \mathbf{y}),$$

*with token-wise reverse transitions given by the inference posterior in (4) tilted by the likelihood,*

$$p_\varphi(Z_s^l|Z_t, \mathbf{y}) \propto q(Z_s^l|Z_t^l, \mathbf{x}_\varphi(Z_t))\, q(\mathbf{y}|\mathbf{x}_\varphi(Z_t; Z_s^l)).$$

*Then, for any $(\mathbf{x}, \mathbf{y})$,*

$$
\begin{aligned}
-\log p_\varphi(\mathbf{x}|\mathbf{y}) \leq \mathcal{L}_{\text{DDPS}}(\mathbf{x}, \mathbf{y}; \varphi) &:= \mathbb{E}_{q(Z_0|\mathbf{x})}[-\log p_\varphi(\mathbf{x}|Z_0)] \\
&+ \sum_{i=1}^T \mathbb{E}_{q(Z_{t(i)}|\mathbf{x})}\left[ \frac{\alpha_{s(i)} - \alpha_{t(i)}}{1 - \alpha_{t(i)}} \sum_{l=1}^L \text{CE}(\mathbf{x}^l, \mathbf{x}_\varphi^l(Z_{t(i)}))\mathbf{1}_{\{Z_{t(i)}^l = \mathbf{m}\}} \right. \\
&\qquad - \mathbb{E}_{q(Z_{s(i)}|Z_{t(i)}, \mathbf{x})}\left[ \sum_{l=1}^L \log q(\mathbf{y}|\mathbf{x}_\varphi(Z_{t(i)}; Z_{s(i)}^l)) \right] \\
&\qquad \left. + \sum_{l=1}^L \log \mathcal{Z}_\varphi^l(Z_{t(i)}, \mathbf{y}) \right].
\end{aligned}
\tag{9}
$$

*Proof.* We present the derivation for sequence length $L = 1$; the extension to $L > 1$ follows by token-wise factorization as in (Sohl-Dickstein et al., 2015). Recall our parameterized reverse kernel (identical in form to (4) up to a likelihood tilt):

$$
q(Z_s^l|Z_t^l, \mathbf{x}_\varphi(Z_t)) = \begin{cases} \text{Cat}(Z_s^l; Z_t^l), & Z_t^l \neq \mathbf{m}, \\ \text{Cat}\left(Z_s^l; \frac{\alpha_s - \alpha_t}{1 - \alpha_t}\mathbf{x}_\varphi(Z_t) + \frac{1 - \alpha_s}{1 - \alpha_t}\mathbf{m}\right), & Z_t^l = \mathbf{m}. \end{cases}
\tag{10}
$$

Starting from the conditional likelihood,

$$
\begin{aligned}
-\log p_\varphi(\mathbf{x}|\mathbf{y}) &= -\log \int p_\varphi(\mathbf{x}, Z_{0:1}|\mathbf{y})\, dZ_{0:1} \\
&= -\log \int p_\varphi(\mathbf{x}, Z_{0:1}|\mathbf{y}) \frac{q(Z_{0:1}|\mathbf{x}, \mathbf{y})}{q(Z_{0:1}|\mathbf{x}, \mathbf{y})}\, dZ_{0:1}.
\end{aligned}
$$

By the conditional independence of the forward process (the noising path is conditionally independent of $\mathbf{y}$ given $\mathbf{x}$), we have $q(Z_{0:1}|\mathbf{x}, \mathbf{y}) = q(Z_{0:1}|\mathbf{x})$, hence

$$-\log p_\varphi(\mathbf{x}|\mathbf{y}) = -\log \mathbb{E}_{q(Z_{0:1}|\mathbf{x})}\left[ \frac{p_\varphi(\mathbf{x}, Z_{0:1}|\mathbf{y})}{q(Z_{0:1}|\mathbf{x})} \right].$$

Applying Jensen's inequality,

$$-\log p_\varphi(\mathbf{x}|\mathbf{y}) \leq \mathbb{E}_{q(Z_{0:1}|\mathbf{x})}\left[ -\log \frac{p_\varphi(\mathbf{x}, Z_{0:1}|\mathbf{y})}{q(Z_{0:1}|\mathbf{x})} \right].$$

Using the factorization $p_\varphi(\mathbf{x}, Z_{0:1}|\mathbf{y}) = p_\varphi(Z_1|\mathbf{y}) \, p_\varphi(\mathbf{x}|Z_0, \mathbf{y}) \prod_{i=1}^T p_\varphi(Z_{s(i)}|Z_{t(i)}, \mathbf{y})$ and the forward process factorization $q(Z_{0:1}|\mathbf{x}) = q(Z_1|\mathbf{x}) \prod_{i=1}^T q(Z_{s(i)}|Z_{t(i)}, \mathbf{x})$, we obtain

$$- \log \frac{p_\varphi(\mathbf{x}, Z_{0:1}|\mathbf{y})}{q(Z_{0:1}|\mathbf{x})}$$

$$= - \log p_\varphi(\mathbf{x}|Z_0, \mathbf{y}) + \log \frac{q(Z_1|\mathbf{x})}{p_\varphi(Z_1|\mathbf{y})} + \sum_{i=1}^T \log \frac{q(Z_{s(i)}|Z_{t(i)}, \mathbf{x})}{p_\varphi(Z_{s(i)}|Z_{t(i)}, \mathbf{y})}.$$

Taking expectation under $q(Z_{0:1}|\mathbf{x})$ yields

$$- \log p_\varphi(\mathbf{x}|\mathbf{y}) \le \mathbb{E}_{q(Z_{0:1}|\mathbf{x})}\big[ - \log p_\varphi(\mathbf{x}|Z_0, \mathbf{y})\big] + \mathbb{E}_{q(Z_1|\mathbf{x})}\Big[ \log \frac{q(Z_1|\mathbf{x})}{p_\varphi(Z_1|\mathbf{y})} \Big]$$

$$+ \sum_{i=1}^T \mathbb{E}_{q(Z_{t(i)}|\mathbf{x})} \mathbb{E}_{q(Z_{s(i)}|Z_{t(i)}, \mathbf{x})} \Big[ \log \frac{q(Z_{s(i)}|Z_{t(i)}, \mathbf{x})}{p_\varphi(Z_{s(i)}|Z_{t(i)}, \mathbf{y})} \Big]. \tag{11}$$

Expanding the tilted reverse kernel using the parameterization:

$$p_\varphi(Z_{s(i)}|Z_{t(i)}, \mathbf{y}) = \frac{q(Z_{s(i)}|Z_{t(i)}, \mathbf{x}_\varphi(Z_{t(i)})) \, q(\mathbf{y}|\mathbf{x}_\varphi(Z_{t(i)}; Z_{s(i)}))}{\mathcal{Z}_\varphi(Z_{t(i)}, \mathbf{y})}, \quad \text{where}$$

$$\mathcal{Z}_\varphi(Z_{t(i)}, \mathbf{y}) = \sum_{\mathbf{z}_s \in \mathcal{V}} q\big(\mathbf{z}_s|Z_{t(i)}, \mathbf{x}_\varphi(Z_{t(i)})\big) \, q\big(\mathbf{y}|\mathbf{x}_\varphi(Z_{t(i)}; \mathbf{z}_s)\big),$$

we can rewrite the last term as

$$\sum_{i=1}^T \mathbb{E}_{q(Z_{t(i)}|\mathbf{x})} \mathbb{E}_{q(Z_{s(i)}|Z_{t(i)}, \mathbf{x})} \Big[ \log \frac{q(Z_{s(i)}|Z_{t(i)}, \mathbf{x})}{q(Z_{s(i)}|Z_{t(i)}, \mathbf{x}_\varphi(Z_{t(i)}))} - \log q(\mathbf{y}|\mathbf{x}_\varphi(Z_{t(i)}; Z_{s(i)})) + \log \mathcal{Z}_\varphi(Z_{t(i)}, \mathbf{y}) \Big].$$

This yields the final decomposition:

$$- \log p_\varphi(\mathbf{x}|\mathbf{y}) \le \underbrace{\mathbb{E}_{q(Z_0|\mathbf{x})}[- \log p_\varphi(\mathbf{x}|Z_0, \mathbf{y})] + \mathrm{KL}\big(q(Z_1|\mathbf{x}) \,\|\, p_\varphi(Z_1|\mathbf{y})\big)}_{\text{reconstruction + boundary KL}}$$

$$+ \underbrace{\sum_{i=1}^T \mathbb{E}_{q(Z_{t(i)}|\mathbf{x})} \Big[ \mathrm{KL}\big(q(Z_{s(i)}|Z_{t(i)}, \mathbf{x}) \,\|\, q(Z_{s(i)}|Z_{t(i)}, \mathbf{x}_\varphi(Z_{t(i)}))\big) \Big]}_{\text{per-step KLs}}$$

$$- \underbrace{\sum_{i=1}^T \mathbb{E}_{q(Z_{t(i)}, Z_{s(i)}|\mathbf{x})} \big[ \log q(\mathbf{y}|\mathbf{x}_\varphi(Z_{t(i)}; Z_{s(i)})) \big]}_{\text{likelihood tilt terms}}$$

$$+ \underbrace{\sum_{i=1}^T \mathbb{E}_{q(Z_{t(i)}|\mathbf{x})} \big[ \log \mathcal{Z}_\varphi(Z_{t(i)}, \mathbf{y}) \big]}_{\text{normalizing factor}} \coloneqq \mathcal{L}_{\mathrm{DDPS}}(\mathbf{x}, \mathbf{y}; \varphi).$$

The first two groups of terms correspond to the standard masked diffusion posterior NELBO objective (Sohl-Dickstein et al., 2015; Austin et al., 2021) denoted by:

$$\mathcal{L}_{\mathrm{PNELBO}}(\mathbf{x}, \mathbf{y}; \varphi) \coloneqq \mathbb{E}_{q(Z_0|\mathbf{x})}\big[ - \log p_\varphi(\mathbf{x}|Z_0, \mathbf{y})\big]$$

$$+ \mathrm{KL}\big(q(Z_1|\mathbf{x}) \,\|\, p_\varphi(Z_1|\mathbf{y})\big)$$

$$+ \sum_{i=1}^T \mathbb{E}_{q(Z_{t(i)}|\mathbf{x})} \Big[ \mathrm{KL}\big(q(Z_{s(i)}|Z_{t(i)}, \mathbf{x}) \,\|\, q(Z_{s(i)}|Z_{t(i)}, \mathbf{x}_\varphi(Z_{t(i)}))\big) \Big].$$

The likelihood term,

$$-\sum_{i=1}^{T}\mathbb{E}_{q(Z_{t(i)},Z_{s(i)}|\mathbf{x})}\big[\log q(\mathbf{y}|\mathbf{x}_\varphi(Z_{t(i)};Z_{s(i)}))\big],$$

capture the effect of incorporating observations $\mathbf{y}$ into posterior sampling. Finally, we have an additive normalizing term that ensures the validity of the reverse transition (6).

We now bound the per-step KL terms. Having connected the decomposition to the NELBO, it suffices to compute, for each step $i$, the divergence

$$\mathrm{KL}\big(q(Z_{s(i)}|Z_{t(i)},\mathbf{x}) \,\|\, q(Z_{s(i)}|Z_{t(i)},\mathbf{x}_\varphi(Z_{t(i)}))\big).$$

Since masked diffusion induces a two–state posterior at each step (either remain masked or reveal the data token), we treat the two cases for $Z_{t(i)}$ separately.

**Case I:** $Z_{t(i)} = \mathbf{m}$. From the masked diffusion forward process (3), when $Z_{t(i)} = \mathbf{m}$ the true posterior over $Z_{s(i)}$ has mass

$$q(Z_{s(i)} = \mathbf{m}|Z_{t(i)} = \mathbf{m}, \mathbf{x}) = \frac{1 - \alpha_{s(i)}}{1 - \alpha_{t(i)}}, \qquad q(Z_{s(i)} = \mathbf{x}|Z_{t(i)} = \mathbf{m}, \mathbf{x}) = \frac{\alpha_{s(i)} - \alpha_{t(i)}}{1 - \alpha_{t(i)}}.$$

Under the model with prediction $\mathbf{x}_\varphi(Z_{t(i)})$ (a categorical distribution) as in (4), the "unmask" branch is weighted by the model's probability of the true token, $\langle\mathbf{x}_\varphi(Z_{t(i)}), \mathbf{x}\rangle$, while the mask probability is unchanged. Hence,

$$\mathrm{KL}\big(q(Z_{s(i)}|Z_{t(i)}\!=\!\mathbf{m}, \mathbf{x}) \,\|\, q(Z_{s(i)}|Z_{t(i)}\!=\!\mathbf{m}, \mathbf{x}_\varphi(Z_{t(i)}))\big)$$

$$= \frac{1 - \alpha_{s(i)}}{1 - \alpha_{t(i)}} \log \frac{\frac{1-\alpha_{s(i)}}{1-\alpha_{t(i)}}}{\frac{1-\alpha_{s(i)}}{1-\alpha_{t(i)}}} + \frac{\alpha_{s(i)} - \alpha_{t(i)}}{1 - \alpha_{t(i)}} \log \frac{\frac{\alpha_{s(i)}-\alpha_{t(i)}}{1-\alpha_{t(i)}}}{\frac{\alpha_{s(i)}-\alpha_{t(i)}}{1-\alpha_{t(i)}} \langle\mathbf{x}_\varphi(Z_{t(i)}), \mathbf{x}\rangle}$$

$$= \frac{\alpha_{s(i)} - \alpha_{t(i)}}{1 - \alpha_{t(i)}} \log \frac{1}{\langle\mathbf{x}_\varphi(Z_{t(i)}), \mathbf{x}\rangle} \;\; = \;\; \frac{\alpha_{t(i)} - \alpha_{s(i)}}{1 - \alpha_{t(i)}} \log\langle\mathbf{x}_\varphi(Z_{t(i)}), \mathbf{x}\rangle.$$

Equivalently, writing the cross-entropy with the one-hot target $\mathbf{x}$ as $\mathrm{CE}(\mathbf{x}, \mathbf{x}_\varphi(Z_{t(i)})) = -\log\langle\mathbf{x}_\varphi(Z_{t(i)}), \mathbf{x}\rangle$,

$$\mathrm{KL}\big(q(Z_{s(i)}|Z_{t(i)}\!=\!\mathbf{m}, \mathbf{x}) \,\|\, q(Z_{s(i)}|Z_{t(i)}\!=\!\mathbf{m}, \mathbf{x}_\varphi(Z_{t(i)}))\big) = \frac{\alpha_{s(i)} - \alpha_{t(i)}}{1 - \alpha_{t(i)}} \mathrm{CE}(\mathbf{x}, \mathbf{x}_\varphi(Z_{t(i)})).$$

**Case II:** $Z_{t(i)} \neq \mathbf{m}$. When the current token is already unmasked, the posterior is deterministic: $q(Z_{s(i)}\!=\!Z_{t(i)}|Z_{t(i)} \neq \mathbf{m}, \mathbf{x}) = 1$. Thus,

$$\mathrm{KL}\big(q(Z_{s(i)}|Z_{t(i)} \neq \mathbf{m}, \mathbf{x}) \,\|\, q(Z_{s(i)}|Z_{t(i)} \neq \mathbf{m}, \mathbf{x}_\varphi(Z_{t(i)}))\big) = 0.$$

Only masked coordinates contribute to the per-step KL, yielding for each step $i$,

$$\mathrm{KL}\big(q(Z_{s(i)}|Z_{t(i)}, \mathbf{x}) \,\|\, q(Z_{s(i)}|Z_{t(i)}, \mathbf{x}_\varphi(Z_{t(i)}))\big) = \frac{\alpha_{s(i)} - \alpha_{t(i)}}{1 - \alpha_{t(i)}} \mathrm{CE}\big(\mathbf{x}, \mathbf{x}_\varphi(Z_{t(i)})\big)\mathbf{1}_{\{Z_{t(i)}=\mathbf{m}\}}.$$

This recovers the standard masked-diffusion NELBO weighting (cf. (5)): per-step contributions are cross-entropies at masked positions, scaled by $(\alpha_{s(i)} - \alpha_{t(i)})/(1 - \alpha_{t(i)})$. Generalizing this to sequences with length $L > 1$ yields:

$$\mathrm{KL}\big(q(Z_{s(i)}|Z_{t(i)}, \mathbf{x}) \,\|\, q(Z_{s(i)}|Z_{t(i)}, \mathbf{x}_\varphi(Z_{t(i)}))\big) = \frac{\alpha_{s(i)} - \alpha_{t(i)}}{1 - \alpha_{t(i)}} \sum_{l=1}^{L} \mathrm{CE}\big(\mathbf{x}^l, \mathbf{x}_\varphi^l(Z_{t(i)})\big)\mathbf{1}_{\{Z_{t(i)}^l=\mathbf{m}\}}.$$

Factorizing (11) for $L > 1$ and following the results from above, we get

$$\mathcal{L}_{\mathrm{DDPS}}(\mathbf{x}, \mathbf{y}; \varphi) = \mathbb{E}_{q(Z_0|\mathbf{x})}[-\log p_\varphi(\mathbf{x}|Z_0, \mathbf{y})] + \mathrm{KL}\big(q(Z_1|\mathbf{x}) \,\|\, p_\varphi(Z_1|\mathbf{y})\big)$$

$$+ \sum_{i=1}^{T}\mathbb{E}_{q(Z_{t(i)}|\mathbf{x})}\left[\frac{\alpha_{s(i)} - \alpha_{t(i)}}{1 - \alpha_{t(i)}} \sum_{l=1}^{L} \mathrm{CE}\big(\mathbf{x}^l, \mathbf{x}_\varphi^l(Z_{t(i)})\big)\mathbf{1}_{\{Z_{t(i)}^l=\mathbf{m}\}}\right]$$

$$- \sum_{i=1}^{T}\mathbb{E}_{q(Z_{t(i)},Z_{s(i)}|\mathbf{x})}\left[\sum_{l=1}^{L}\log q(\mathbf{y}|\mathbf{x}_\varphi(Z_{t(i)}; Z_{s(i)}^l))\right]$$

$$+ \sum_{i=1}^{T}\mathbb{E}_{q(Z_{t(i)}|\mathbf{x})}\left[\sum_{l=1}^{L}\log \mathcal{Z}_\varphi^l(Z_{t(i)}, \mathbf{y})\right].$$

In masked diffusion $Z_1$ is typically the fully masked state or absorbing state, so $q(Z_1|\mathbf{x})$ is degenerate and, with $p_\varphi(Z_1|\mathbf{y}) = q(Z_1|\mathbf{x})$, the boundary KL vanishes. Thus,

$$
\begin{aligned}
\mathcal{L}_{\text{DDPS}}(\mathbf{x}, \mathbf{y}; \varphi) = \; & \mathbb{E}_{q(Z_0|\mathbf{x})}[-\log p_\varphi(\mathbf{x}|Z_0, \mathbf{y})] \\
& + \sum_{i=1}^{T} \mathbb{E}_{q(Z_{t(i)}|\mathbf{x})} \left[ \frac{\alpha_{s(i)} - \alpha_{t(i)}}{1 - \alpha_{t(i)}} \sum_{l=1}^{L} \text{CE}\big(\mathbf{x}^l, \mathbf{x}_\varphi^l(Z_{t(i)})\big) \mathbf{1}_{\{Z_{t(i)}^l = \mathbf{m}\}} \right] \\
& - \sum_{i=1}^{T} \mathbb{E}_{q(Z_{t(i)}, Z_{s(i)}|\mathbf{x})} \left[ \sum_{l=1}^{L} \log q(\mathbf{y}|\mathbf{x}_\varphi(Z_{t(i)}; Z_{s(i)}^l)) \right] \\
& + \sum_{i=1}^{T} \mathbb{E}_{q(Z_{t(i)}|\mathbf{x})} \left[ \sum_{l=1}^{L} \log \mathcal{Z}_\varphi^l(Z_{t(i)}, \mathbf{y}) \right].
\end{aligned}
$$

Furthermore, $\mathbf{y}$ imposes a distribution over $\mathbf{x}$ from which we wish to sample. However, when $Z_0$ is given then $\mathbf{x}$ is uniquely determined. Therefore,

$$
\begin{aligned}
\mathcal{L}_{\text{DDPS}}(\mathbf{x}, \mathbf{y}; \varphi) = \; & \mathbb{E}_{q(Z_0|\mathbf{x})}[-\log p_\varphi(\mathbf{x}|Z_0)] \\
& + \sum_{i=1}^{T} \mathbb{E}_{q(Z_{t(i)}|\mathbf{x})} \left[ \frac{\alpha_{s(i)} - \alpha_{t(i)}}{1 - \alpha_{t(i)}} \sum_{l=1}^{L} \text{CE}\big(\mathbf{x}^l, \mathbf{x}_\varphi^l(Z_{t(i)})\big) \mathbf{1}_{\{Z_{t(i)}^l = \mathbf{m}\}} \right] \\
& - \sum_{i=1}^{T} \mathbb{E}_{q(Z_{t(i)}, Z_{s(i)}|\mathbf{x})} \left[ \sum_{l=1}^{L} \log q(\mathbf{y}|\mathbf{x}_\varphi(Z_{t(i)}; Z_{s(i)}^l)) \right] \\
& + \sum_{i=1}^{T} \mathbb{E}_{q(Z_{t(i)}|\mathbf{x})} \left[ \sum_{l=1}^{L} \log \mathcal{Z}_\varphi(Z_{t(i)}, \mathbf{y}) \right],
\end{aligned}
$$

which upon regrouping completes the proof of the statement. $\qquad\square$

**Implications.** **Theorem B.1** provides a principled upper bound on the negative log-posterior likelihood in discrete masked diffusion models.

- *Trainable upper bound.* The pathwise upper bound $\mathcal{L}_{\text{DDPS}}(\mathbf{x}, \mathbf{y}; \varphi)$ in (9) provides a principled training criterion for discrete diffusion posterior sampling.

- *Reduction to standard training when $\mathbf{y}$ is absent.* Setting the likelihood to a constant (no measurements) removes the tilt terms and recovers the masked-diffusion training objective: only the reconstruction term and per-step KL-divergence terms remain.

- *Masked-token supervision.* The per-step KL-divergence terms vanish on already-revealed tokens and reduce to weighted cross-entropies on masked tokens, focusing learning signal exactly where denoising must occur. The weights $(\alpha_{s(i)} - \alpha_{t(i)})/(1 - \alpha_{t(i)})$ expose how the noise schedule shapes gradient magnitude.

- *Data-consistency via tilt.* The additive terms $-\sum_i \mathbb{E}_{q(Z_{t(i)}, Z_{s(i)}|\mathbf{x})}[\log q(\mathbf{y}|\mathbf{x}_\varphi(Z_{t(i)}; Z_{s(i)}))]$ encourage reverse transitions that produce intermediate predictions consistent with the measurement model, integrating task specific information at every denoising step.

- *Boundary conditions.* With an absorbing mask state, the boundary KL-divergence is constant (often zero), so optimization concentrates on reconstruction, token-level KL-divergence terms, and measurement consistency.

- *Compatibility with efficient parameterizations.* Because (9) is written in terms of token-wise categoricals, it directly supports time-independent or lightweight parameterizations (e.g., shared denoisers), helping scalability to long sequences and high resolution.

**Theorem B.2** (Test-time Anchored Posterior Sampling)**.** *Let $Z_{0:1} = \{Z_{t(i)}\}_{i=0}^{T}$ with $t(i) = i/T$ and $s(i) = (i-1)/T$ denote the latent path of a masked discrete diffusion model, and let $q(Z_{0:1}|\mathbf{x})$ be the forward noising law from (3). Assume the*

*decoder is shared between unconditional generation and posterior sampling ($p_\varphi(\mathbf{x}|Z_0) = p_\theta(\mathbf{x}|Z_0)$), and the unconditional reverse transitions are parameterized as in (4). Define $\mathcal{L}_{\mathrm{NELBO}}(\mathbf{x}; \theta)$ as in (5). Then, for any $(\mathbf{x}, \mathbf{y})$,*

$$-\log p_\varphi(\mathbf{x}|\mathbf{y}) \ \le \ \mathcal{L}_{\mathrm{APS}}(\mathbf{x}, \mathbf{y}; \varphi),$$

*where*

$$\mathcal{L}_{\mathrm{APS}}(\mathbf{x}, \mathbf{y}; \varphi) \coloneqq \mathcal{L}_{\mathrm{NELBO}}(\mathbf{x}; \theta) + \sum_{i=1}^{T} \mathbb{E}_{q(Z_{t(i)}|\mathbf{x})} \Bigg[ \frac{\alpha_{s(i)} - \alpha_{t(i)}}{1 - \alpha_{t(i)}} \sum_{l=1}^{L} \log \frac{\langle \mathbf{x}_\theta^l(Z_{t(i)}), \mathbf{x}^l \rangle}{\langle \mathbf{x}_\varphi^l(Z_{t(i)}), \mathbf{x}^l \rangle} \mathbf{1}_{\{Z_{t(i)} = \mathbf{m}\}}$$

$$- \mathbb{E}_{q(Z_{s(i)}|Z_{t(i)}, \mathbf{x})} \Big[ \sum_{l=1}^{L} \log q(\mathbf{y}|\mathbf{x}_\varphi(Z_{t(i)}; Z_{s(i)}^l)) \Big]$$

$$+ \sum_{l=1}^{L} \log \mathcal{Z}_\varphi^l(Z_{t(i)}, \mathbf{y}) \Bigg].$$

*Proof.* From the proof of **Theorem B.1**, the conditional likelihood satisfies

$$-\log p_\varphi(\mathbf{x}|\mathbf{y}) \le \mathbb{E}_{q(Z_0|\mathbf{x})} \big[ -\log p_\varphi(\mathbf{x}|Z_0, \mathbf{y}) \big] + \mathbb{E}_{q(Z_1|\mathbf{x})} \Big[ \log \frac{q(Z_1|\mathbf{x})}{p_\varphi(Z_1|\mathbf{y})} \Big]$$

$$+ \sum_{i=1}^{T} \mathbb{E}_{q(Z_{t(i)}|\mathbf{x})} \mathbb{E}_{q(Z_{s(i)}|Z_{t(i)}, \mathbf{x})} \Big[ \log \frac{q(Z_{s(i)}|Z_{t(i)}, \mathbf{x})}{p_\varphi(Z_{s(i)}|Z_{t(i)}, \mathbf{y})} \Big]. \tag{12}$$

**Compute each term inside summation.** We now focus on the $i^{th}$ term inside the summation. Using the pretrained network $p_\theta(Z_{s(i)}|Z_{t(i)})$ as given in (4), we decompose the term as:

$$\mathbb{E}_{q(Z_{s(i)}|Z_{t(i)}, \mathbf{x})} \Bigg[ \log \frac{q(Z_{s(i)}|Z_{t(i)}, \mathbf{x})}{p_\varphi(Z_{s(i)}|Z_{t(i)}, \mathbf{y})} \Bigg]$$

$$= \mathbb{E}_{q(Z_{s(i)}|Z_{t(i)}, \mathbf{x})} \Bigg[ \log \frac{q(Z_{s(i)}|Z_{t(i)}, \mathbf{x})}{p_\theta(Z_{s(i)}|Z_{t(i)})} + \log \frac{p_\theta(Z_{s(i)}|Z_{t(i)})}{p_\varphi(Z_{s(i)}|Z_{t(i)}, \mathbf{y})} \Bigg].$$

Isolating the first-term as a KL-divergence term, this yields

$$\mathbb{E}_{q(Z_{s(i)}|Z_{t(i)}, \mathbf{x})} \Bigg[ \log \frac{q(Z_{s(i)}|Z_{t(i)}, \mathbf{x})}{p_\varphi(Z_{s(i)}|Z_{t(i)}, \mathbf{y})} \Bigg]$$

$$= \underbrace{\mathrm{KL}\Big( q(Z_{s(i)}|Z_{t(i)}, \mathbf{x}) \,\|\, p_\theta(Z_{s(i)}|Z_{t(i)}) \Big)}_{\text{divergence w.r.t. pretrained network}} + \mathbb{E}_{q(Z_{s(i)}|Z_{t(i)}, \mathbf{x})} \Bigg[ \log \frac{p_\theta(Z_{s(i)}|Z_{t(i)})}{p_\varphi(Z_{s(i)}|Z_{t(i)}, \mathbf{y})} \Bigg].$$

Since $p_\theta(Z_{s(i)}|Z_{t(i)})$ is parameterized via the pretrained network prediction $\mathbf{x}_\theta(Z_{t(i)})$, we get

$$\mathrm{KL}\Big( q(Z_{s(i)}|Z_{t(i)}, \mathbf{x}) \,\|\, p_\theta(Z_{s(i)}|Z_{t(i)}) \Big) = \mathrm{KL}\Big( q(Z_{s(i)}|Z_{t(i)}, \mathbf{x}) \,\|\, q(Z_{s(i)}|Z_{t(i)}, \mathbf{x}_\theta(Z_{t(i)})) \Big).$$

Following the argument of **Theorem B.1**, we distinguish two cases for $Z_{t(i)}$. For $Z_{t(i)} = \mathbf{m}$, the KL-divergence evaluates to

$$\mathrm{KL}\big( q(Z_{s(i)}|Z_{t(i)}, \mathbf{x}) \,\|\, q(Z_{s(i)}|Z_{t(i)}, \mathbf{x}_\theta(Z_{t(i)})) \big) = \frac{\alpha_{s(i)} - \alpha_{t(i)}}{1 - \alpha_{t(i)}} \mathrm{CE}\big( \mathbf{x}, \mathbf{x}_\theta(Z_{t(i)}) \big),$$

while for $Z_{t(i)} \ne \mathbf{m}$ the KL-divergence vanishes. Thus we obtain

$$\mathbb{E}_{q(Z_{s(i)}|Z_{t(i)}, \mathbf{x})} \Bigg[ \log \frac{q(Z_{s(i)}|Z_{t(i)}, \mathbf{x})}{p_\varphi(Z_{s(i)}|Z_{t(i)}, \mathbf{y})} \Bigg] \tag{13}$$

$$= \frac{\alpha_{s(i)} - \alpha_{t(i)}}{1 - \alpha_{t(i)}} \mathrm{CE}\big( \mathbf{x}, \mathbf{x}_\theta(Z_{t(i)}) \big) \mathbf{1}_{\{Z_{t(i)} = \mathbf{m}\}} + \mathbb{E}_{q(Z_{s(i)}|Z_{t(i)}, \mathbf{x})} \Bigg[ \log \frac{p_\theta(Z_{s(i)}|Z_{t(i)})}{p_\varphi(Z_{s(i)}|Z_{t(i)}, \mathbf{y})} \Bigg].$$

Substituting the above expression in the conditional likelihood (12), we get

$$
\begin{aligned}
-\log p_\varphi(\mathbf{x}|\mathbf{y}) \leq\ & \mathbb{E}_{q(Z_0|\mathbf{x})}\big[-\log p_\varphi(\mathbf{x}|Z_0,\mathbf{y})\big] + \mathbb{E}_{q(Z_1|\mathbf{x})}\Big[\log \frac{q(Z_1|\mathbf{x})}{p_\varphi(Z_1|\mathbf{y})}\Big] \\
& + \sum_{i=1}^{T} \mathbb{E}_{q(Z_{t(i)}|\mathbf{x})}\left[\frac{\alpha_{s(i)}-\alpha_{t(i)}}{1-\alpha_{t(i)}}\,\mathrm{CE}\big(\mathbf{x},\mathbf{x}_\theta(Z_{t(i)})\big)\,\mathbf{1}_{\{Z_{t(i)}=\mathbf{m}\}}\right] \\
& + \sum_{i=1}^{T} \mathbb{E}_{q(Z_{t(i)}|\mathbf{x})}\mathbb{E}_{q(Z_{s(i)}|Z_{t(i)},\mathbf{x})}\left[\log \frac{p_\theta(Z_{s(i)}|Z_{t(i)})}{p_\varphi(Z_{s(i)}|Z_{t(i)},\mathbf{y})}\right].
\end{aligned}
\tag{14}
$$

**Expected log-likelihood ratio under** $q$**.** We now examine the expected difference in log-likelihoods between the prior and posterior transition distributions under the conditional law of the forward process. Recall that the reverse transition under our parameterization is

$$
p_\varphi(Z_{s(i)}|Z_{t(i)},\mathbf{y}) = \frac{q(Z_{s(i)}|Z_{t(i)},\mathbf{x}_\varphi(Z_{t(i)}))\,q(\mathbf{y}|\mathbf{x}_\varphi(Z_{t(i)};Z_{s(i)}))}{\mathcal{Z}_\varphi(Z_{t(i)},\mathbf{y})}, \quad \text{where}
$$
$$
\mathcal{Z}_\varphi(Z_{t(i)},\mathbf{y}) = \sum_{\mathbf{z}_s\in\mathcal{V}} q\big(\mathbf{z}_s|Z_{t(i)},\mathbf{x}_\varphi(Z_{t(i)})\big)\,q\big(\mathbf{y}|\mathbf{x}_\varphi(Z_{t(i)};\mathbf{z}_s)\big).
$$

Consider the expectation of the log-likelihood ratio under the forward posterior $q(Z_{s(i)}|Z_{t(i)},\mathbf{x})$:

$$
\mathbb{E}_{q(Z_{s(i)}|Z_{t(i)},\mathbf{x})}\left[\log \frac{p_\theta(Z_{s(i)}|Z_{t(i)})}{p_\varphi(Z_{s(i)}|Z_{t(i)},\mathbf{y})}\right].
$$

Substituting the definition of $p_\varphi$, we obtain

$$
\begin{aligned}
& \mathbb{E}_{q(Z_{s(i)}|Z_{t(i)},\mathbf{x})}\left[\log \frac{p_\theta(Z_{s(i)}|Z_{t(i)})\,\mathcal{Z}_\varphi(Z_{t(i)},\mathbf{y})}{q(Z_{s(i)}|Z_{t(i)},\mathbf{x}_\varphi(Z_{t(i)}))\,q(\mathbf{y}|\mathbf{x}_\varphi(Z_{t(i)};Z_{s(i)}))}\right] \\
&= \mathbb{E}_{q(Z_{s(i)}|Z_{t(i)},\mathbf{x})}\left[\log \frac{p_\theta(Z_{s(i)}|Z_{t(i)})}{q(Z_{s(i)}|Z_{t(i)},\mathbf{x}_\varphi(Z_{t(i)}))} - \log q(\mathbf{y}|\mathbf{x}_\varphi(Z_{t(i)};Z_{s(i)})) + \log \mathcal{Z}_\varphi(Z_{t(i)},\mathbf{y})\right] \\
&= \mathbb{E}_{q(Z_{s(i)}|Z_{t(i)},\mathbf{x})}\left[\log \frac{p_\theta(Z_{s(i)}|Z_{t(i)})}{q(Z_{s(i)}|Z_{t(i)},\mathbf{x}_\varphi(Z_{t(i)}))}\right] - \mathbb{E}_{q(Z_{s(i)}|Z_{t(i)},\mathbf{x})}\left[\log q(\mathbf{y}|\mathbf{x}_\varphi(Z_{t(i)};Z_{s(i)}))\right] \\
& \hspace{9cm} + \log \mathcal{Z}_\varphi(Z_{t(i)},\mathbf{y}).
\end{aligned}
$$

Since $p_\theta(Z_{s(i)}|Z_{t(i)})$ is represented by the network prediction $\mathbf{x}_\theta(Z_{t(i)})$, we can rewrite the first expectation as:

$$
\mathbb{E}_{q(Z_{s(i)}|Z_{t(i)},\mathbf{x})}\left[\log \frac{p_\theta(Z_{s(i)}|Z_{t(i)})}{q(Z_{s(i)}|Z_{t(i)},\mathbf{x}_\varphi(Z_{t(i)}))}\right] = \mathbb{E}_{q(Z_{s(i)}|Z_{t(i)},\mathbf{x})}\left[\log \frac{q(Z_{s(i)}|Z_{t(i)},\mathbf{x}_\theta(Z_{t(i)}))}{q(Z_{s(i)}|Z_{t(i)},\mathbf{x}_\varphi(Z_{t(i)}))}\right].
$$

Putting everything together, the expected log-likelihood ratio under $q$ becomes

$$
\begin{aligned}
& \mathbb{E}_{q(Z_{s(i)}|Z_{t(i)},\mathbf{x})}\left[\log \frac{p_\theta(Z_{s(i)}|Z_{t(i)})}{p_\varphi(Z_{s(i)}|Z_{t(i)},\mathbf{y})}\right] \\
&= \mathbb{E}_{q(Z_{s(i)}|Z_{t(i)},\mathbf{x})}\left[\log \frac{q(Z_{s(i)}|Z_{t(i)},\mathbf{x}_\theta(Z_{t(i)}))}{q(Z_{s(i)}|Z_{t(i)},\mathbf{x}_\varphi(Z_{t(i)}))}\right] - \mathbb{E}_{q(Z_{s(i)}|Z_{t(i)},\mathbf{x})}\left[\log q(\mathbf{y}|\mathbf{x}_\varphi(Z_{t(i)};Z_{s(i)}))\right] \\
& \hspace{9cm} + \log \mathcal{Z}_\varphi(Z_{t(i)},\mathbf{y}).
\end{aligned}
\tag{15}
$$

Similarly to the proof of **Theorem B.1**, we consider two cases to compute the first expectation.

**Case I:** $Z_{t(i)} = \mathbf{m}$. Using the masked diffusion inference posterior derived from (3), when $Z_{t(i)} = \mathbf{m}$ we have

$$q(Z_{s(i)} = \mathbf{m} \mid Z_{t(i)} = \mathbf{m}, \mathbf{x}) = \frac{1 - \alpha_{s(i)}}{1 - \alpha_{t(i)}}, \quad q(Z_{s(i)} = \mathbf{x} \mid Z_{t(i)} = \mathbf{m}, \mathbf{x}) = \frac{\alpha_{s(i)} - \alpha_{t(i)}}{1 - \alpha_{t(i)}}.$$

Under the pretrained model parameterization (4), the corresponding terms are

$$q(Z_{s(i)} = \mathbf{m} \mid Z_{t(i)} = \mathbf{m}, \mathbf{x}_\theta(Z_{t(i)})) = \frac{1 - \alpha_{s(i)}}{1 - \alpha_{t(i)}},$$

$$q(Z_{s(i)} = \mathbf{x} \mid Z_{t(i)} = \mathbf{m}, \mathbf{x}_\theta(Z_{t(i)})) = \frac{\alpha_{s(i)} - \alpha_{t(i)}}{1 - \alpha_{t(i)}} \langle \mathbf{x}_\theta(Z_{t(i)}), \mathbf{x} \rangle,$$

and likewise with $\mathbf{x}_\varphi(Z_{t(i)})$ replacing $\mathbf{x}_\theta(Z_{t(i)})$ for our parameterization. Hence,

$$\mathbb{E}_{q(Z_{s(i)} \mid Z_{t(i)} = \mathbf{m}, \mathbf{x})} \left[ \log \frac{q(Z_{s(i)} \mid Z_{t(i)} = \mathbf{m}, \mathbf{x}_\theta(Z_{t(i)}))}{q(Z_{s(i)} \mid Z_{t(i)} = \mathbf{m}, \mathbf{x}_\varphi(Z_{t(i)}))} \right]$$

$$= \frac{1 - \alpha_{s(i)}}{1 - \alpha_{t(i)}} \log \frac{\frac{1 - \alpha_{s(i)}}{1 - \alpha_{t(i)}}}{\frac{1 - \alpha_{s(i)}}{1 - \alpha_{t(i)}}} + \frac{\alpha_{s(i)} - \alpha_{t(i)}}{1 - \alpha_{t(i)}} \log \frac{\frac{\alpha_{s(i)} - \alpha_{t(i)}}{1 - \alpha_{t(i)}} \langle \mathbf{x}_\theta(Z_{t(i)}), \mathbf{x} \rangle}{\frac{\alpha_{s(i)} - \alpha_{t(i)}}{1 - \alpha_{t(i)}} \langle \mathbf{x}_\varphi(Z_{t(i)}), \mathbf{x} \rangle}$$

$$= \frac{\alpha_{s(i)} - \alpha_{t(i)}}{1 - \alpha_{t(i)}} \log \frac{\langle \mathbf{x}_\theta(Z_{t(i)}), \mathbf{x} \rangle}{\langle \mathbf{x}_\varphi(Z_{t(i)}), \mathbf{x} \rangle}.$$

**Case II:** $Z_{t(i)} \neq \mathbf{m}$. When the token is already revealed, the posterior is deterministic:

$$q(Z_{s(i)} = Z_{t(i)} \mid Z_{t(i)} \neq \mathbf{m}, \mathbf{x}) = 1,$$

and this form is identical for the $\mathbf{x}_\theta$- and $\mathbf{x}_\varphi$-parameterized posteriors as well:

$$q(Z_{s(i)} = Z_{t(i)} \mid Z_{t(i)} \neq \mathbf{m}, \mathbf{x}_\theta(Z_{t(i)})) = 1, \quad q(Z_{s(i)} = Z_{t(i)} \mid Z_{t(i)} \neq \mathbf{m}, \mathbf{x}_\varphi(Z_{t(i)})) = 1.$$

Therefore,

$$\mathbb{E}_{q(Z_{s(i)} \mid Z_{t(i)} \neq \mathbf{m}, \mathbf{x})} \left[ \log \frac{q(Z_{s(i)} \mid Z_{t(i)}, \mathbf{x}_\theta(Z_{t(i)}))}{q(Z_{s(i)} \mid Z_{t(i)}, \mathbf{x}_\varphi(Z_{t(i)}))} \right] = 0$$

Combining both cases,

$$\mathbb{E}_{q(Z_{s(i)} \mid Z_{t(i)}, \mathbf{x})} \left[ \log \frac{q(Z_{s(i)} \mid Z_{t(i)}, \mathbf{x}_\theta(Z_{t(i)}))}{q(Z_{s(i)} \mid Z_{t(i)}, \mathbf{x}_\varphi(Z_{t(i)}))} \right] = \frac{\alpha_{s(i)} - \alpha_{t(i)}}{1 - \alpha_{t(i)}} \log \frac{\langle \mathbf{x}_\theta(Z_{t(i)}), \mathbf{x} \rangle}{\langle \mathbf{x}_\varphi(Z_{t(i)}), \mathbf{x} \rangle} \mathbf{1}_{\{Z_{t(i)} = \mathbf{m}\}},$$

which simplifies (15) as follows:

$$\mathbb{E}_{q(Z_{s(i)} \mid Z_{t(i)}, \mathbf{x})} \left[ \log \frac{p_\theta(Z_{s(i)} \mid Z_{t(i)})}{p_\varphi(Z_{s(i)} \mid Z_{t(i)}, \mathbf{y})} \right]$$

$$= \frac{\alpha_{s(i)} - \alpha_{t(i)}}{1 - \alpha_{t(i)}} \log \frac{\langle \mathbf{x}_\theta(Z_{t(i)}), \mathbf{x} \rangle}{\langle \mathbf{x}_\varphi(Z_{t(i)}), \mathbf{x} \rangle} \mathbf{1}_{\{Z_{t(i)} = \mathbf{m}\}} - \mathbb{E}_{q(Z_{s(i)} \mid Z_{t(i)}, \mathbf{x})} \left[ \log q(\mathbf{y} \mid \mathbf{x}_\varphi(Z_{t(i)}; Z_{s(i)})) \right]$$

$$+ \log \mathcal{Z}_\varphi(Z_{t(i)}, \mathbf{y}). \qquad (16)$$

Substituting (16) into (14), the conditional likelihood can be bounded as

$$
-\log p_\varphi(\mathbf{x}|\mathbf{y}) \le \mathbb{E}_{q(Z_{0:1}|\mathbf{x})}[-\log p_\varphi(\mathbf{x}|Z_0, \mathbf{y})] + \mathbb{E}_{q(Z_1|\mathbf{x})}\Big[\log \frac{q(Z_1|\mathbf{x})}{p_\varphi(Z_1|\mathbf{y})}\Big]
$$

$$
+ \sum_{i=1}^{T} \mathbb{E}_{q(Z_{t(i)}|\mathbf{x})}\left[\frac{\alpha_{s(i)} - \alpha_{t(i)}}{1 - \alpha_{t(i)}} \, \mathrm{CE}(\mathbf{x}, \mathbf{x}_\theta(Z_{t(i)})) \, \mathbf{1}_{\{Z_{t(i)}=\mathbf{m}\}}\right]
$$

$$
+ \sum_{i=1}^{T} \mathbb{E}_{q(Z_{t(i)}|\mathbf{x})}\mathbb{E}_{q(Z_{s(i)}|Z_{t(i)},\mathbf{x})}\left[\log \frac{p_\theta(Z_{s(i)}|Z_{t(i)})}{p_\varphi(Z_{s(i)}|Z_{t(i)}, \mathbf{y})}\right]
$$

$$
= \mathbb{E}_{q(Z_{0:1}|\mathbf{x})}[-\log p_\varphi(\mathbf{x}|Z_0, \mathbf{y})] + \mathbb{E}_{q(Z_1|\mathbf{x})}\Big[\log \frac{q(Z_1|\mathbf{x})}{p_\varphi(Z_1|\mathbf{y})}\Big]
$$

$$
+ \sum_{i=1}^{T} \mathbb{E}_{q(Z_{t(i)}|\mathbf{x})}\left[\frac{\alpha_{s(i)} - \alpha_{t(i)}}{1 - \alpha_{t(i)}} \, \mathrm{CE}(\mathbf{x}, \mathbf{x}_\theta(Z_{t(i)})) \, \mathbf{1}_{\{Z_{t(i)}=\mathbf{m}\}}\right]
$$

$$
+ \sum_{i=1}^{T} \mathbb{E}_{q(Z_{t(i)}|\mathbf{x})}\left[\frac{\alpha_{s(i)} - \alpha_{t(i)}}{1 - \alpha_{t(i)}} \, \log \frac{\langle \mathbf{x}_\theta(Z_{t(i)}), \mathbf{x}\rangle}{\langle \mathbf{x}_\varphi(Z_{t(i)}), \mathbf{x}\rangle} \, \mathbf{1}_{\{Z_{t(i)}=\mathbf{m}\}}\right]
$$

$$
- \sum_{i=1}^{T} \mathbb{E}_{q(Z_{t(i)}, Z_{s(i)}|\mathbf{x})}\Big[\log q(\mathbf{y}|\mathbf{x}_\varphi(Z_{t(i)}; Z_{s(i)}))\Big]
$$

$$
+ \sum_{i=1}^{T} \mathbb{E}_{q(Z_{t(i)}|\mathbf{x})}\Big[\log \mathcal{Z}_\varphi(Z_{t(i)}, \mathbf{y})\Big]. \tag{17}
$$

**Treatment of boundary conditions.** In masked diffusion the terminal state $Z_1$ is absorbing (all-mask), hence $q(Z_1|\mathbf{x}) = p_\varphi(Z_1|\mathbf{y})$ and the boundary KL-divergence vanishes. Thus (17) becomes

$$
-\log p_\varphi(\mathbf{x}|\mathbf{y}) \le \mathbb{E}_{q(Z_0|\mathbf{x})}\big[-\log p_\varphi(\mathbf{x}|Z_0, \mathbf{y})\big]
$$

$$
+ \sum_{i=1}^{T} \mathbb{E}_{q(Z_{t(i)}|\mathbf{x})}\left[\frac{\alpha_{s(i)} - \alpha_{t(i)}}{1 - \alpha_{t(i)}} \, \mathrm{CE}\big(\mathbf{x}, \mathbf{x}_\theta(Z_{t(i)})\big) \, \mathbf{1}_{\{Z_{t(i)}=\mathbf{m}\}}\right]
$$

$$
+ \sum_{i=1}^{T} \mathbb{E}_{q(Z_{t(i)}|\mathbf{x})}\left[\frac{\alpha_{s(i)} - \alpha_{t(i)}}{1 - \alpha_{t(i)}} \, \log \frac{\langle \mathbf{x}_\theta(Z_{t(i)}), \mathbf{x}\rangle}{\langle \mathbf{x}_\varphi(Z_{t(i)}), \mathbf{x}\rangle} \, \mathbf{1}_{\{Z_{t(i)}=\mathbf{m}\}}\right]
$$

$$
- \sum_{i=1}^{T} \mathbb{E}_{q(Z_{t(i)}, Z_{s(i)}|\mathbf{x})}\Big[\log q(\mathbf{y}|\mathbf{x}_\varphi(Z_{t(i)}; Z_{s(i)}))\Big]
$$

$$
+ \sum_{i=1}^{T} \mathbb{E}_{q(Z_{t(i)}|\mathbf{x})}\Big[\log \mathcal{Z}_\varphi(Z_{t(i)}, \mathbf{y})\Big]. \tag{18}
$$

Since $\mathbf{x}$ is a deterministic function of $Z_0$, the reconstruction term in (18) simplifies to:

$$
-\log p_\varphi(\mathbf{x}|\mathbf{y}) \le \mathbb{E}_{q(Z_0|\mathbf{x})}\big[-\log p_\varphi(\mathbf{x}|Z_0)\big]
$$

$$
+ \sum_{i=1}^{T} \mathbb{E}_{q(Z_{t(i)}|\mathbf{x})}\left[\frac{\alpha_{s(i)} - \alpha_{t(i)}}{1 - \alpha_{t(i)}} \, \mathrm{CE}\big(\mathbf{x}, \mathbf{x}_\theta(Z_{t(i)})\big) \, \mathbf{1}_{\{Z_{t(i)}=\mathbf{m}\}}\right]
$$

$$
+ \sum_{i=1}^{T} \mathbb{E}_{q(Z_{t(i)}|\mathbf{x})}\left[\frac{\alpha_{s(i)} - \alpha_{t(i)}}{1 - \alpha_{t(i)}} \, \log \frac{\langle \mathbf{x}_\theta(Z_{t(i)}), \mathbf{x}\rangle}{\langle \mathbf{x}_\varphi(Z_{t(i)}), \mathbf{x}\rangle} \, \mathbf{1}_{\{Z_{t(i)}=\mathbf{m}\}}\right]
$$

$$
- \sum_{i=1}^{T} \mathbb{E}_{q(Z_{t(i)}, Z_{s(i)}|\mathbf{x})}\Big[\log q(\mathbf{y}|\mathbf{x}_\varphi(Z_{t(i)}; Z_{s(i)}))\Big]
$$

$$
+ \sum_{i=1}^{T} \mathbb{E}_{q(Z_{t(i)}|\mathbf{x})}\Big[\log \mathcal{Z}_\varphi(Z_{t(i)}, \mathbf{y})\Big]. \tag{19}
$$

Finally, the decoder is same for unconditional generation and posterior sampling, i.e., $p_\varphi(\mathbf{x}|Z_0) = p_\theta(\mathbf{x}|Z_0)$. Substituting this property in (19) yields

$$
\begin{aligned}
-\log p_\varphi(\mathbf{x}|\mathbf{y}) \leq\ & \mathbb{E}_{q(Z_0|\mathbf{x})}\big[-\log p_\theta(\mathbf{x}|Z_0)\big] \\
& + \sum_{i=1}^{T} \mathbb{E}_{q(Z_{t(i)}|\mathbf{x})}\left[\frac{\alpha_{s(i)}-\alpha_{t(i)}}{1-\alpha_{t(i)}}\ \mathrm{CE}\big(\mathbf{x}, \mathbf{x}_\theta(Z_{t(i)})\big)\ \mathbf{1}_{\{Z_{t(i)}=\mathbf{m}\}}\right] \\
& + \sum_{i=1}^{T} \mathbb{E}_{q(Z_{t(i)}|\mathbf{x})}\left[\frac{\alpha_{s(i)}-\alpha_{t(i)}}{1-\alpha_{t(i)}}\ \log\frac{\langle \mathbf{x}_\theta(Z_{t(i)}),\mathbf{x}\rangle}{\langle \mathbf{x}_\varphi(Z_{t(i)}),\mathbf{x}\rangle}\ \mathbf{1}_{\{Z_{t(i)}=\mathbf{m}\}}\right] \\
& - \sum_{i=1}^{T} \mathbb{E}_{q(Z_{t(i)},Z_{s(i)}|\mathbf{x})}\Big[\log q(\mathbf{y}|\mathbf{x}_\varphi(Z_{t(i)};Z_{s(i)}))\Big] \\
& + \sum_{i=1}^{T} \mathbb{E}_{q(Z_{t(i)}|\mathbf{x})}\Big[\log \mathcal{Z}_\varphi(Z_{t(i)},\mathbf{y})\Big] := \mathcal{L}_{\mathrm{APS}}(\mathbf{x},\mathbf{y};\varphi)
\end{aligned}
\tag{20}
$$

Note that $\mathrm{CE}\big(\mathbf{x}, \mathbf{x}_\theta(Z_{t(i)})\big) = -\log\langle \mathbf{x}, \mathbf{x}_\theta(Z_{t(i)})\rangle$. Generalizing this to $L > 1$, the first two terms in (20) equals the standard NELBO (5) used to train the masked diffusion model. Therefore, we have

$$
\begin{aligned}
\mathcal{L}_{\mathrm{APS}}(\mathbf{x},\mathbf{y};\varphi) =\ & \mathcal{L}_{\mathrm{NELBO}}(\mathbf{x};\theta) + \sum_{i=1}^{T} \mathbb{E}_{q(Z_{t(i)}|\mathbf{x})}\left[\frac{\alpha_{s(i)}-\alpha_{t(i)}}{1-\alpha_{t(i)}}\ \sum_{l=1}^{L}\log\frac{\langle \mathbf{x}_\theta^l(Z_{t(i)}),\mathbf{x}^l\rangle}{\langle \mathbf{x}_\varphi^l(Z_{t(i)}),\mathbf{x}^l\rangle}\ \mathbf{1}_{\{Z_{t(i)}^l=\mathbf{m}\}}\right] \\
& - \sum_{i=1}^{T} \mathbb{E}_{q(Z_{t(i)},Z_{s(i)}|\mathbf{x})}\bigg[\sum_{l=1}^{L}\log q(\mathbf{y}|\mathbf{x}_\varphi(Z_{t(i)};Z_{s(i)}^l))\bigg] \\
& + \sum_{i=1}^{T} \mathbb{E}_{q(Z_{t(i)}|\mathbf{x})}\bigg[\sum_{l=1}^{L}\log \mathcal{Z}_\varphi^l(Z_{t(i)},\mathbf{y})\bigg],
\end{aligned}
$$

which upon regrouping completes the statement of the theorem. □

**Implications. Theorem B.2** establishes a principled upper bound on the negative log-posterior likelihood when posterior sampling is performed without additional training.

- *Reuse of pretrained objective.* The bound $\mathcal{L}_{\mathrm{APS}}(\mathbf{x},\mathbf{y};\varphi)$ is expressed in terms of the standard masked diffusion $\mathcal{L}_{\mathrm{NELBO}}(\mathbf{x};\theta)$, meaning that posterior sampling can be performed using pretrained masked diffusion models.

- *Adaptation gap.* The log-ratio correction term quantifies the mismatch between the pretrained transitions $\mathbf{x}_\theta$ and the proposed posterior transitions $\mathbf{x}_\varphi$, effective only at masked positions. This isolates the additional cost of posterior sampling.

- *Measurement consistency.* The final summation enforces alignment with the measurement likelihood $q(\mathbf{y}|\mathbf{x}_\varphi(Z_{t(i)};Z_{s(i)}))$, ensuring that the sampler accounts for observations at each diffusion step.

- *Boundary conditions.* As in the training bound, the absorbing mask state renders the boundary KL-divergence constant (often zero), so the effective objective simplifies to the pretrained NELBO plus adaptation and measurement terms.

- *Test-time posterior sampling.* Together, the decomposition clarifies how posterior sampling can be performed without retraining: start from the pretrained NELBO and add corrections for model adaptation at masked positions while incorporating measurements via tilting.

# C. Additional Experiments

This section provides supplementary details and evaluations of our APS method. We first describe implementation details for both inverse problems and stylization (§C.1), followed by algorithmic analysis including pseudocode and hyperparameter studies (§C.2). We then examine the impact of design choices in our ablations (§C.3), evaluate using the same prior (§4.6),

and discuss computational complexity (§C.4). Next, we outline the compared baselines (§C.6) and summarize benchmarks and metrics (§C.7). Finally, we present additional results (§C.8), limitations (§C.9), and conclude with reproducibility (§C.10).

## C.1. Implementation Details

### C.1.1. INVERSE PROBLEMS

For inverse problems, we implement our test-time optimization using two main configurations corresponding to the APS ($512 \times 512$) and APS-L ($1024 \times 1024$) results reported in §4. The full reverse sampling process is discretized into 15 time steps following a cosine mask schedule, using a classifier-free guidance scale of 3.5.

At each of the 15 reverse steps, we perform an inner optimization loop to ensure measurement consistency. This loop consists of 100 optimization steps using the Adam optimizer with a learning rate of 1.0. The total loss function is a weighted sum of two components: (1) a reconstruction loss, which could be L1 or L2 norm, but we choose L1 norm: $\|\mathbf{y} - \mathcal{A}(\hat{\mathbf{x}})\|_1$ as it is known to generate sharper quality), and (2) a VGG perceptual loss with a coefficient of $10^{-3}$.

The specific parameters for each degradation operator $\mathcal{A}(\cdot)$ vary by task, with all measurements simulated by adding Gaussian noise of $\sigma = 0.05$. For super resolution, we use a $4\times$ downsampling factor. For Gaussian Deblurring, the operator is a Gaussian kernel of size $61 \times 61$ with a standard deviation of 3.0. Motion Deblurring uses a kernel of the same size with an intensity parameter of 0.5 (on a scale of 0 for linear to 1 for highly nonlinear), corresponding to a moderately nonlinear motion path. For Inpainting, we randomly remove 70% of pixels. Nonlinear Deblurring kernels are generated using the `KernelWizard`[3] model from the `bkse`[4] library. Finally, High Dynamic Range (HDR) reconstruction is modeled by scaling the image data and clipping the result, following the operation clip(data $\times 2, -1, 1$).

### C.1.2. REFERENCE-BASED STYLIZATION

Our approach to training-free, reference-based stylization leverages the core APS framework by framing the task as a *highly nonlinear* inverse problem. Let $\hat{\mathbf{x}}_{\mathrm{ref}}$ denote the conditional reference image providing the style. The target measurement, a style vector $\mathbf{y}$, is obtained by applying a pretrained Contrastive Style Descriptor (CSD) (Somepalli et al., 2024) model as our measurement operator $\mathcal{A}(\cdot)$ to this reference image, i.e., $\mathbf{y} = \mathcal{A}(\hat{\mathbf{x}}_{\mathrm{ref}})$. At each reverse diffusion step $t$, APS aims to generate a sample whose style matches this target.

As discussed in §3, there are two main stages of APS. In the first stage (§3.1), we perform a differentiable forward pass by computing the expected codebook embedding $\bar{\mathbf{x}}^l = \sum_{k=1}^{K} \mathbf{c}_k \mathbf{x}_\varphi^l(\mathbf{z}_{t(i)})$ for $l = 1, \ldots, L$ using the model's output probabilities $\mathbf{x}_\theta(\mathbf{z}_t)^l$ as initial condition for $\varphi_{t(i)}^l$. The straight-through estimator is then used to obtain a differentiable image representation $\hat{\mathbf{x}} = \mathcal{D}(\tilde{\mathbf{x}})$. To guide the optimization, the measurement consistency loss is calculated as the cosine distance between the style vector of the generated image and the target style vector $\mathbf{y}$ as follows:

$$\mathcal{L}_{\mathrm{style}}(\varphi_{t(i)}) = 1 - \frac{\langle \mathcal{A}(\hat{\mathbf{x}}), \mathbf{y}\rangle}{\|\mathcal{A}(\hat{\mathbf{x}})\|\|\mathbf{y}\|}.$$

For each reverse step, we perform 100 optimization steps using an Adam optimizer with a learning rate of 0.1. Gradients from this style loss are backpropagated through the frozen VQ-VAE decoder $\mathcal{D}(\cdot)$ and the straight-through estimator $\tilde{\mathbf{x}}$ to update all the entries of the conditional probability table $\varphi_{t(i)} = \{\varphi_{t(i)}^l\}_{l=1}^{L}$.

In the second stage (§3.2), the posterior estimate $\mathbf{x}_\varphi(\mathbf{z}_{t(i)})$ obtained from the first stage is used adaptively unmask anchor tokens in the sequence. Both the processes continue over 15 total steps following a cosine mask schedule. The fully unmasked sequence satisfies the stylistic constraints without requiring any task-specific retraining of the foundation model.

### C.1.3. HIGH-RESOLUTION INFERENCE

To demonstrate the scalability of our method, we experiment with a higher-resolution setting. For a fair comparison on our $256 \times 256$ benchmark, we follow the upsampling protocol described in G2D2 (Murata et al., 2024). Specifically, for both our standard ($512 \times 512$) and large-scale (APS-L, $1024 \times 1024$) models, we first upsample the benchmark images to the model's native resolution before applying the forward operator. For the APS-L configuration, this increases the number of visual

---

[3] https://github.com/LeviBorodenko/motionblur
[4] https://github.com/VinAIResearch/blur-kernel-space-exploring

tokens by a factor of four (from 1024 to 4096)[5]. The Transformer-based MMaDA (Yang et al., 2025b) model accommodates this by processing a longer sequence without architectural changes. After the high-resolution reconstruction is complete, we downsample the output back to the benchmark's native $256 \times 256$ resolution for evaluation. As demonstrated in our experiments (§4.1, §4.2), this approach further improves performance, achieving substantial gains in both PSNR and LPIPS.

### C.1.4. QUESTION ANSWERING

In standard discrete diffusion frameworks, the model predicts logits for every position in the sequence, from which tokens are sampled independently. For images, these tokens are in the latent space and hence projected via a decoder into pixel space. In that setting, the combination of the decoder and a measurement error term effectively acts as a proxy for a reward model, allowing our approach APS to provide gradient-like guidance.

In the context of diffusion language modeling, we replace the combination of decoder and measurement error with a scalar reward model (e.g., `Qwen3-0.6B`) that evaluates the quality of the generated text. The input to this reward model is discrete. The non-differentiable nature of the sampling step breaks the computation graph, preventing the direct backpropagation of the reward signal to the diffusion logits.

Our proposed method, APS, addresses this intractability. APS derives gradient-like guidance through quantized expectation, allowing us to "tilt" the logits of the base model in the direction of higher reward. This enables effective guidance through the discrete bottleneck, analogous to how gradients guide the generation process in discrete image domains.

We integrate our APS sampler into the inference pipeline of `Dream-7B-Instruct`. For all question answering experiments, we fix the total diffusion steps at $T = 128$ and the maximum generation length at 128 tokens. We perform $M = 3$ gradient optimization steps per diffusion timestep with a learning rate of $\eta = 0.1$ to update the intermediate logits. During inference, we generate tokens using a sampling temperature of $0.2$ and nucleus sampling with $p = 0.95$. Finally, the remasking temperature is set to $0.0$, ensuring our underlying masking schedule remains consistent with the baseline.

### C.2. Algorithm Details

Algorithm 1 details our APS procedure. The process begins with a fully masked latent space, $\mathbf{z}_1$, and iteratively refines the image over $T$ reverse diffusion steps. Each step features an inner optimization phase designed to align the model's predictions with the given measurement $\mathbf{y}$. To enable gradient-based optimization through the discrete quantization step which assigns each dimension of an embedding to its nearest value in $\{-1, 1\}$ we employ the straight-through estimator (STE) via the stop-gradient operator, $\mathrm{sg}(\cdot)$.

This optimization is guided by a composite loss function. For reconstruction, we primarily use the Mean Absolute Error (MAE, or L1 loss), which we find produces perceptually superior results compared to the Mean Squared Error (MSE). While MSE corresponds to maximizing the Gaussian log-likelihood, MAE is typically more robust. This is supplemented with a VGG perceptual loss to enforce similarity in the feature domain, further improving visual quality. Finally, a small entropy term is included to regularize the posterior distribution. Once the optimization at a given step is complete, the Anchored Remasking strategy uses the model's confidence defined as the probability assigned to the chosen token for each position to selectively unmask tokens for the next iteration.

To determine the optimal weight for the perceptual loss, we conducted an ablation study on its coefficient, $\lambda_{\mathrm{p}}$. As shown in Table 8, a value of $10^{-3}$ provides the optimal trade-off between reconstruction fidelity (PSNR) and perceptual quality (LPIPS) across our benchmarks.

**Effect of perceptual loss Coefficient.** Table 8 reports the effect of varying the perceptual loss coefficient on ImageNet and FFHQ. Small weights ($10^{-5}$, $10^{-4}$) only marginally improve perceptual quality over the baseline, while larger weights ($10^{-2}$, $10^{-1}$) overly emphasize perceptual similarity at the cost of distortion and structure. A coefficient of $10^{-3}$ provides the best balance: on ImageNet, it reduces LPIPS from $0.416$ to $0.334$ while maintaining PSNR $23.61$ and SSIM $0.639$, and on FFHQ, it achieves the lowest LPIPS ($0.247$) with a slight drop in PSNR ($26.61$) and SSIM ($0.781$). We therefore adopt $10^{-3}$ as the default across datasets. Notably, our approach introduces only this single hyperparameter, whereas G2D2 relies on multiple carefully tuned schedules (e.g., four different coefficients) that must be re-optimized for each task. In contrast, we use the same setting for all inverse problems, underscoring the robustness and simplicity of our posterior sampler relative

---

[5]The visual tokenizer (Yu et al., 2024) used in MMaDA (Yang et al., 2025b) uses $16\times$ downscaling, generating $1024 = 32 \times 32$ tokens for an image of size $512 \times 512$.

---

**Algorithm 1** Test-Time Anchored Posterior Sampling (APS)

---

1: **Input:** Denoising steps $T$, measurement $\mathbf{y}$, denoiser $\mathbf{x}_\theta^{\text{logits}}(\cdot)$, operator $\mathcal{A}(\cdot)$, decoder $\mathcal{D}(\cdot)$.
2: **Tunable parameters:** Optimization steps $M$, learning rate $\eta$, loss coefficients $\lambda_{\text{p}}, \lambda_e$.
3: **Output:** Reconstructed image $\hat{\mathbf{x}}_{\text{final}}$.
4: Initialize latent state $\mathbf{z}_1 \leftarrow \{\mathbf{m}\}^L$ (all masked)
5: **for** $i = T$ **to** $1$ **do**
6: $\quad \varphi \leftarrow \mathbf{x}_\theta^{\text{logits}}(\mathbf{z}_{t(i)})$ $\hspace{4cm}$ ▷ Quantized Expectation §3.1
7: $\quad$ **for** $m = 1$ **to** $M$ **do**
8: $\quad\quad \mathbf{x}_\varphi(\mathbf{z}_{t(i)}) = \text{Softmax}(\varphi)$
9: $\quad\quad \bar{\mathbf{x}} \leftarrow \sum_{k=1}^{K} \mathbf{c}_k \cdot \mathbf{x}_\varphi(\mathbf{z}_{t(i)})[k]$
10: $\quad\quad \mathbf{x} = \mathcal{Q}_{\text{lfq}}(\bar{\mathbf{x}})$
11: $\quad\quad \tilde{\mathbf{x}} \leftarrow \bar{\mathbf{x}} + \left[\mathbf{x} - \bar{\mathbf{x}}\right]_{\text{sg}}$
12: $\quad\quad \hat{\mathbf{x}} \leftarrow \mathcal{D}(\tilde{\mathbf{x}})$
13: $\quad\quad \mathcal{L} \leftarrow \mathcal{L}_{\text{reconstruction}}(\mathcal{A}(\hat{\mathbf{x}}), \mathbf{y}) + \lambda_{\text{p}} \mathcal{L}_{\text{perceptual}}(\mathcal{A}(\hat{\mathbf{x}}), \mathbf{y})$
14: $\quad\quad \varphi \leftarrow \text{Adam}(\varphi, \nabla_\varphi \mathcal{L}, \eta)$
15: $\quad$ **end for**
16: $\quad \mathbf{x}_\varphi^*(\mathbf{z}_{t(i)}) = \text{Softmax}(\varphi)$ $\hspace{3.5cm}$ ▷ Anchored Remasking §3.2
17: $\quad \bar{\mathbf{x}}^* \leftarrow \sum_{k=1}^{K} \mathbf{c}_k \cdot \mathbf{x}_\varphi^*(\mathbf{z}_{t(i)})[k]$
18: $\quad \mathbf{x} = \mathcal{Q}_{\text{lfq}}(\bar{\mathbf{x}}^*)$
19: $\quad \kappa^l \leftarrow \langle (\mathbf{x}_\varphi^*(\mathbf{z}_{t(i)}))^l, \mathbf{x}^l \rangle$ for $l = 1, \ldots, L$
20: $\quad \mathcal{P}_{t(i)} \leftarrow \{l : \kappa^l \geq \tau_{t(i)}\}$
21: $\quad \mathbf{z}_{s(i)} \leftarrow \text{Update State}(\mathbf{z}_{t(i)}, \mathbf{x}, \mathcal{P}_{t(i)})$
22: **end for**
23: $\hat{\mathbf{x}}_{\text{final}} \leftarrow \mathcal{D}(\mathbf{z}_0)$
24: **return** $\hat{\mathbf{x}}_{\text{final}}$

---

to prior discrete diffusion methods.

## C.3. Ablation Study

To better understand the impact of our algorithmic innovations, we conduct a systematic ablation study on ImageNet SR ($4\times$), as shown in Figure 6 (top row). Our analysis focuses on three core design choices: (i) *standard remasking*, which highlights the limitations of confidence-based token selection under the prior; (ii) *quantized expectation*, which addresses sampling bias and improves measurement consistency; and (iii) *anchored remasking*, which preserves informative tokens identified by optimization while suppressing spurious high-confidence background tokens. Together, these ablations disentangle the contributions of each component, providing both theoretical insight and qualitative evidence (Figure 6) into how APS achieves stable and measurement-consistent posterior sampling.

### C.3.1. EFFECT OF STANDARD REMASKING

Standard remasking in MaskGIT (Chang et al., 2022) and MMaDA (Yang et al., 2025b) proceeds as follows. At each iteration, the base denoiser produces a distribution over all tokens $\mathbf{x}_\theta(\mathbf{z}_{t(i)}) = \{\mathbf{x}_\theta^l(\mathbf{z}_{t(i)})\}_{l=1}^L$ given the current partially unmasked sequence $\mathbf{z}_t$. Sampled tokens $\mathbf{x} = \{\mathbf{x}^l\}_{l=1}^L$ are drawn independently at each position, and previously unmasked tokens are carried over to the next state $\mathbf{z}_s$. For masked positions, the confidence of each sampled token $\mathbf{x}^l$ is defined as

$$\kappa_t^l = \langle \mathbf{x}_\theta^l(\mathbf{z}_{t(i)}), \mathbf{x}^l \rangle.$$

Tokens to unmask are then selected based on an adaptive threshold schedule (e.g., cosine),

$$\mathcal{P}_t = \{l : \kappa_t^l \geq \tau_t\},$$

which favors unmasking the most confident tokens under the prior distribution $\mathbf{x}_\theta(\mathbf{z}_t)$.

In language generation, this corresponds to unmasking frequent, low-information tokens (e.g., articles or conjunctions) (Rout et al., 2025a). Analogously in images, the model tends to unmask background regions first, since they dominate training

*Table 8.* **Hyperparameter sweeps on perceptual loss coefficient** ($\lambda_{\mathrm{p}}$) **on ImageNet and FFHQ.** Highlighted rows denote the chosen setting (1e-3), which offers a trade-off among perceptual quality (LPIPS), distortion (PSNR), and structure (SSIM).

<table>
<tr><td colspan="4" align="center">*(a)* ImageNet</td><td colspan="4" align="center">*(b)* FFHQ</td></tr>
<tr><td>$\lambda_{\mathrm{p}}$</td><td>LPIPS ↓</td><td>PSNR ↑</td><td>SSIM ↑</td><td>$\lambda_{\mathrm{p}}$</td><td>LPIPS ↓</td><td>PSNR ↑</td><td>SSIM ↑</td></tr>
<tr><td>0.0</td><td>0.416</td><td>23.98</td><td>0.652</td><td>0.0</td><td>0.311</td><td>27.32</td><td>0.801</td></tr>
<tr><td>1e-5</td><td>0.402</td><td>24.05</td><td>**0.658**</td><td>1e-5</td><td>0.301</td><td>27.40</td><td>0.803</td></tr>
<tr><td>1e-4</td><td>0.380</td><td>**24.06**</td><td>0.655</td><td>1e-4</td><td>0.268</td><td>**27.67**</td><td>**0.812**</td></tr>
<tr><td>1e-3</td><td>0.334</td><td>23.61</td><td>0.639</td><td>1e-3</td><td>**0.247**</td><td>26.61</td><td>0.781</td></tr>
<tr><td>1e-2</td><td>**0.325**</td><td>22.06</td><td>0.566</td><td>5e-3</td><td>0.252</td><td>24.90</td><td>0.729</td></tr>
<tr><td>5e-2</td><td>0.328</td><td>21.44</td><td>0.543</td><td>1e-2</td><td>0.256</td><td>24.49</td><td>0.715</td></tr>
<tr><td>1e-1</td><td>0.327</td><td>21.29</td><td>0.539</td><td>1e-1</td><td>0.260</td><td>23.44</td><td>0.692</td></tr>
</table>

statistics and are easier to predict. As a result, informative foreground tokens remain masked until late in the process, limiting semantic guidance and increasing conditional entropy.

Qualitatively, Figure 6 (second row) illustrates this phenomenon. While the model confidently unmasks background tokens early, the salient object is revealed only much later, highlighting the limitation of standard remasking for posterior sampling.

### C.3.2. EFFECT OF QUANTIZED EXPECTATION

Standard remasking suffers from two key issues: (i) sampling bias, and (ii) independent token-wise confidence. When sampling tokens directly from the unconditional distribution $\mathbf{x}_\theta(\mathbf{z}_t)$, the model may pick unrelated or spurious tokens, which—once unmasked—remain fixed in all future steps. This introduces inconsistency and often locks the model into poor generations.

To address this, we propose *quantized expectation* (§3.1). Instead of sampling, we tilt the unconditional distribution $\mathbf{x}_\theta(\mathbf{z}_t)$ towards the measurement likelihood, obtaining an approximate posterior $\mathbf{x}_\varphi(\mathbf{z}_t)$. We then optimize in the span of codebook embeddings by treating the tilted probabilities as coefficients in a linear combination and passing their expectation through the decoder using a straight-through estimator. The resulting embedding is then quantized back to the nearest valid token. This procedure implicitly maximizes the measurement likelihood, avoids sampling noise, and enables the discovery of tokens with zero prior probability mass but strong measurement consistency–leading to a better posterior sample.

We treat such tokens as "anchor tokens," since they minimize reconstruction error and provide critical guidance under the measurement operator. As shown in Figure 6 (third row, APS-I), quantized expectation corrects sampling bias and yields reconstructions that remain consistent with observations throughout the denoising trajectory.

### C.3.3. EFFECT OF ANCHORED REMASKING

A limitation of confidence-based remasking under the prior is that anchor tokens, obtained through our optimization procedure, may receive near-zero probability mass under $\mathbf{x}_\theta(\mathbf{z}_t)$. As a result, the model would discard these informative tokens in favor of background tokens, which the prior predicts with high confidence. This reintroduces the very bias we aim to avoid.

To address this, we compute token confidence using the posterior estimate $\mathbf{x}_\varphi(\mathbf{z}_t)$ rather than the unconditional prior. In this way, anchor tokens identified via quantized expectation are preserved, while low-likelihood background tokens are remasked. Qualitatively, this effect is evident at $t = 6/15$ in Figure 6 (fourth row, APS-II): unlike the standard prior-based strategy (second row), which prematurely unmasks background pixels, our approach commits to anchor tokens aligned with the bird's body. This ensures that the background (blue sky in the measurement) is correctly down-weighted, as it is inconsistent with the prior white background generated by $\mathbf{x}_\theta$. Subsequent steps therefore refine the image conditioned on these anchor tokens, reducing conditional entropy and producing reconstructions that remain faithful to the measurements.

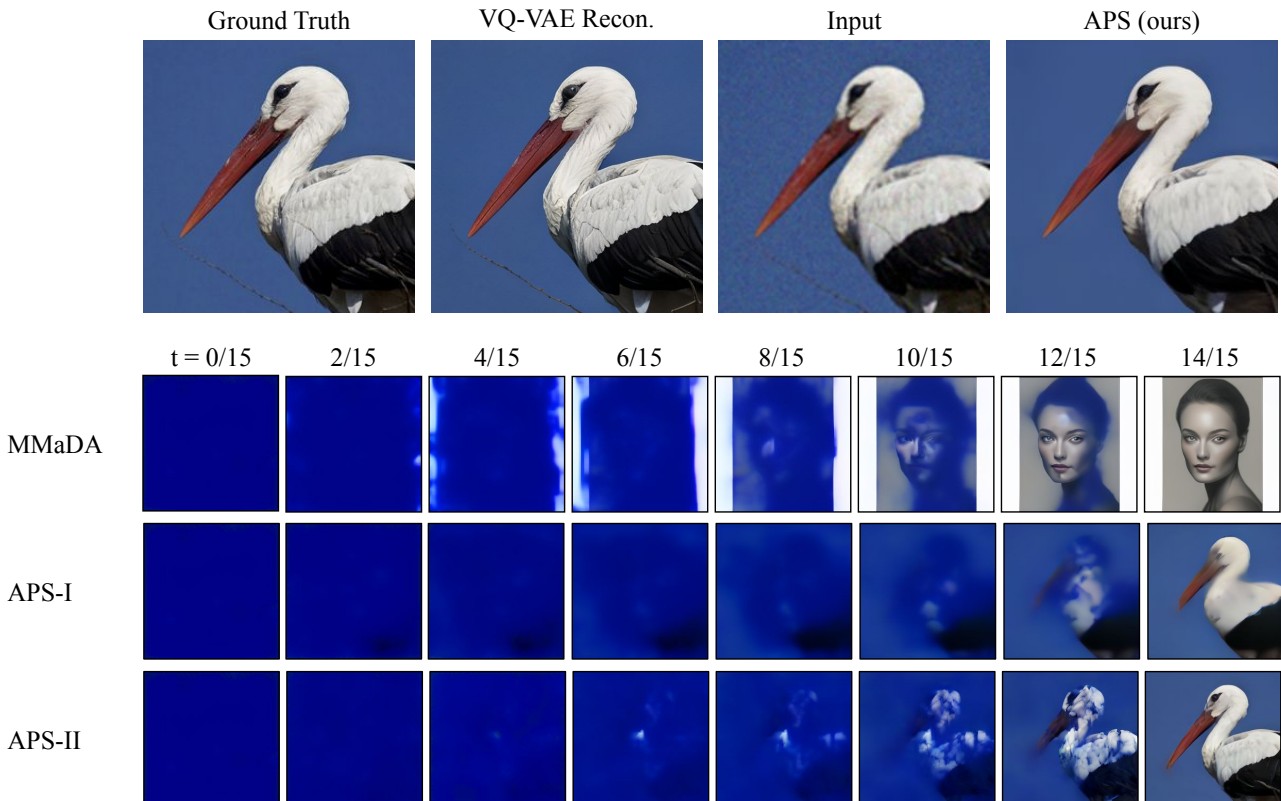

Prompt: "A very high quality image, natural looking"

*Figure 6.* **Ablation study of design choices in APS.** We study the posterior sampling problem with the input of the noisy image of a bird (top row, third image from left). Since MMaDA (second row) uses an unconditional prior (no measurement) as the backbone, the final image is inconsistent with the noisy input image (image of person instead of bird). Thus, this shows the effect of *standard remasking*, where high-confidence background tokens are unmasked early. The third row (APS-I) adds *quantized expectation*, which mitigates sampling bias and improves consistency with the measurement, but still relies on prior-based confidence remasking. This results in blurry reconstructions. Finally, the fourth row (APS-II) combines *quantized expectation* with *anchored remasking*, preserving optimized anchor tokens while remasking uninformative background tokens. This combination yields the most stable and measurement-consistent generations, as discussed in Appendix C.3. Indeed, we note that best case one could hope for is the direct VQ-VAE reconstruction (top row, second from left) of the Ground Truth image (top row, leftmost). We observe that APS (top row, rightmost) is comparable in quality to the VQ-VAE reconstruction.

## C.4. Computational Complexity

### C.4.1. TOKENWISE EXPECTATION TO DIFFERENTIABLE SURROGATE

In this section, we provide a detailed derivation of our optimization objective (8) from Equation (7), and analyze the associated computational complexity.

Define the measurement term in Equation (7) as

$$\mathcal{L}_t^{\text{meas}} := \mathbb{E}_{q(Z_s|\mathbf{z}_t,\mathbf{x})}\left[\sum_{l=1}^{L} -\log q\big(\mathbf{y} \mid \mathbf{x}_\varphi(\mathbf{z}_t; Z_s^l)\big)\right] = \sum_{l=1}^{L}\mathbb{E}_{q(Z_s^l|\mathbf{z}_t^l,\mathbf{x})}\Big[-\log q\big(\mathbf{y} \mid \mathbf{x}_\varphi(\mathbf{z}_t; Z_s^l)\big)\Big], \tag{21}$$

where the last equality is due to $q(Z_s|\mathbf{z}_t, \mathbf{x})$ factorizing across sites given $\mathbf{z}_t$ (the forward process is conditionally independent tokenwise). Each inner expectation depends on the *local* categorical $q(Z_s^l|\mathbf{z}_t^l,\mathbf{x})$, which (under the mask process) has support on only two outcomes (e.g., $\mathbf{x}$ or $\mathbf{m}$). Evaluating $\mathcal{L}_t^{\text{meas}}$ exactly would require propagating each choice through $\mathcal{D}$ and $\mathcal{A}$, which is expensive in practice.

**Complexity of the exact marginalization.** We emphasize that evaluating the measurement term exactly is exponential

in the sequence length $L$. Concretely, the outer marginalization over the tokenwise mask choices $Z_s$ involves at most $2^L$ joint assignments (each site is either $\mathbf{m}$ or set to $\mathbf{x}$ under the mask process). For each fixed assignment $Z_s$, computing the inner expectation $\mathbb{E}_{X \sim \mathbf{x}_\varphi(\mathbf{z}_t),\, X^l \leftarrow Z_s^l}[-\log q(\mathbf{y}|X)]$ requires aggregating the likelihood over all possible completions drawn accordingly. If the decoder has a discrete codebook of size $K$ per position, the number of such completions is upper-bounded by $K^L$ (or more tightly $K^{L-1}$ since $l$-the position is fixed with $Z_s^l$), so a worst-case cost for the inner expectation is $O(K^L)$. Combining these two sources of combinatorial explosion gives a conservative worst-case cost of $O(2^L \cdot K^L) = O\big((2K)^L\big)$, i.e., exponential in $L$. Thus, while one can write down the exact marginalization formally, it is computationally intractable for realistic sequence lengths and vocabulary sizes.

**Complexity of the proposed approximation.** We adopt a plug-in approximation that replaces the tokenwise expectation by a single differentiable surrogate. Let $h(\mathbf{x}) := -\log q(\mathbf{y}|\mathbf{x})$ and $h(\mathbf{x}_\varphi(\mathbf{z}_t; \mathbf{z}_s^l))$ be defined as in §3, which we recall here for clarity: $h(\mathbf{x}_\varphi(\mathbf{z}_t; \mathbf{z}_s^l)) = -\log q(\mathbf{y}|\mathbf{x}_\varphi(\mathbf{z}_t; \mathbf{z}_s^l)) = -\mathbb{E}_{X \sim \mathbf{x}_\varphi(\mathbf{z}_t),\, X^l \leftarrow \mathbf{z}_s^l}[\log q(\mathbf{y}|X)]$. Then, a first-order Taylor's expansion of each term inside the summation of (21) around $\mathbf{u} := \mathbb{E}_{Z_s^l \sim q(\cdot|\mathbf{z}_t^l, \mathbf{x})}\mathbb{E}_{X \sim \mathbf{x}_\varphi(\mathbf{z}_t),\, X^l \leftarrow Z_s^l}\big[X\big]$ yields:

$$
\begin{aligned}
\mathbb{E}_{Z_s^l \sim q(\cdot|\mathbf{z}_t^l, \mathbf{x})}\Big[h\big(\mathbf{x}_\varphi(\mathbf{z}_t; Z_s^l)\big)\Big] &= \mathbb{E}_{Z_s^l \sim q(\cdot|\mathbf{z}_t^l, \mathbf{x})}\Big[-\log q\big(\mathbf{y}|\mathbf{x}_\varphi(\mathbf{z}_t; Z_s^l)\big)\Big] = \mathbb{E}_{Z_s^l \sim q(\cdot|\mathbf{z}_t^l, \mathbf{x})}\mathbb{E}_{X \sim \mathbf{x}_\varphi(\mathbf{z}_t),\, X^l \leftarrow Z_s^l}\Big[-\log q\big(\mathbf{y}|X\big)\Big] \\
&\approx \mathbb{E}_{Z_s^l \sim q(\cdot|\mathbf{z}_t^l, \mathbf{x})}\mathbb{E}_{X \sim \mathbf{x}_\varphi(\mathbf{z}_t),\, X^l \leftarrow Z_s^l}\Big[-\log q\big(\mathbf{y}|\mathbf{u}\big) - \langle\nabla\log q(\mathbf{y}|\mathbf{u}), X - \mathbf{u}\rangle\Big] \\
&\qquad \text{(Ignore higher order terms in the Taylor's series expansion.)} \\
&= -\log q\big(\mathbf{y}|\mathbf{u}\big) - \langle\nabla\log q(\mathbf{y}|\mathbf{u}),\ \mathbb{E}_{Z_s^l \sim q(\cdot|\mathbf{z}_t^l, \mathbf{x})}\mathbb{E}_{X \sim \mathbf{x}_\varphi(\mathbf{z}_t),\, X^l \leftarrow Z_s^l}\big[X\big] - \mathbf{u}\rangle \\
&= -\log q\big(\mathbf{y}|\mathbf{u}\big), \quad \text{(The second term cancels by the definition of } \mathbf{u}.) \\
&= h\Big(\mathbb{E}_{Z_s^l \sim q(\cdot|\mathbf{z}_t^l, \mathbf{x})}\mathbb{E}_{X \sim \mathbf{x}_\varphi(\mathbf{z}_t),\, X^l \leftarrow Z_s^l}\big[X\big]\Big).
\end{aligned}
$$

With this approximation, we avoid the computation of $\log q\big(\mathbf{y}|X\big)$ for every sample $X$. This would otherwise require a function call through $\mathcal{A} \circ \mathcal{D}$ per sample $X$, which is expensive. In contrast, the double expectation inside $h(\cdot)$ scales linearly in sequence length ($L$) and vocabulary size ($K$) because for each position of the sequence $X = (X^1, \ldots, X^L)$, the expectation can be independently computed using $\mathbf{x}_\varphi^l(\mathbf{z}_t)$ that costs $\mathcal{O}(K)$, totaling up to $\mathcal{O}(KL)$. With the outer sum over $L$ tokens in (21), the total complexity of $\mathcal{L}_t^{\text{meas}}$ becomes $\mathcal{O}(KL^2)$.

Next, we discuss how to reduce the complexity from $\mathcal{O}(KL^2)$ to $\mathcal{O}(KL)$. Note that $\mathbf{x}_\varphi(\mathbf{z}_t; Z_s^l)$ differs from $\mathbf{x}_\varphi(\mathbf{z}_t)$ only at position $l$. Assuming local smoothness of $\mathcal{A} \circ \mathcal{D}$, we approximate $q(\cdot|\mathbf{z}_t^l, \mathbf{x})$ by $\mathbf{x}_\varphi^l(\mathbf{z}_t)$ resulting in the *quantized expectation*: for each site, form the expected codebook embedding $\bar{\mathbf{x}}^l = \sum_k \mathbf{c}_k\, \mathbf{x}_\varphi^l(\mathbf{z}_t)[k]$, quantize via LFQ as $\mathbf{x}^l = \mathcal{Q}_{\text{lfq}}(\bar{\mathbf{x}}^l)$, and apply the straight-through estimator to obtain a single surrogate sequence $\tilde{\mathbf{x}}_\varphi(\mathbf{z}_t) = \bar{\mathbf{x}} + [\mathbf{x} - \bar{\mathbf{x}}]_{\text{sg}}$. Substituting the same surrogate for all $l$ gives the following[6]:

$$
\mathcal{L}_t^{\text{meas}} \approx \sum_{l=1}^{L} h\big(\tilde{\mathbf{x}}_\varphi(\mathbf{z}_t)\big) = L \cdot \big(-\log q(\mathbf{y}|\tilde{\mathbf{x}}_\varphi(\mathbf{z}_t))\big).
$$

### C.4.2. RUNTIME ANALYSIS OF ANCHORED POSTERIOR SAMPLING

Table 9 highlights the *sampling efficiency* of our anchored posterior sampler (APS) relative to both continuous-diffusion baselines and the prior discrete diffusion sampler G2D2.

**Against continuous (pixel/latent) diffusion.** Pixel-space samplers (DPS, DDRM) either require very long Markov chains (e.g., DPS: 1000 steps, 277 s) or sacrifice quality when shortened; latent-space samplers (PSLD, ReSample) still need 500–1000 steps and hundreds of seconds per image at $256 \times 256$–$512 \times 512$ resolutions (e.g., PSLD: 738 s at $512 \times 512$). In contrast, APS runs with only 15 reverse steps on the MMaDA backbone: 55 s at $256 \times 256$ and 121 s at $512 \times 512$ on a single A100, which is *66× shorter chain* for comparable or better perceptual quality. At $512 \times 512$, APS matches the quality of PSLD as shown in Table 1 while being $\sim$6× faster (PSLD: $\sim$720–740 s vs. APS: 121 s). When scaling the sequence length to $L$=4096 tokens (corresponding resolution $1024 \times 1024$), APS-L completes in 484 s and becomes more accurate: ImageNet Gaussian deblur improves by 22.7% LPIPS and 9.6% PSNR over PSLD with $\sim$1.5× less time.

---

[6]In practice, the factor $L$ can be absorbed into the likelihood weight or learning rate. If only positions with $\mathbf{z}_t^l = \mathbf{m}$ contribute, one may replace $L$ by $|\{l : \mathbf{z}_t^l = \mathbf{m}\}|$. We explicitly show the absorbed constant here for clarity.

*Table 9.* **Quantitative results on sampling efficiency of continuous and discrete samplers.** PDM/LDM denote pixel-/latent-space continuous diffusion models, respectively; VQ-Diffusion and MMaDA are discrete diffusion models. Rows shaded gray report runtimes on a single NVIDIA A6000 GPU copied from G2D2 (Murata et al., 2024); rows shaded orange are measured by us on a single NVIDIA A100 GPU. APS matches or improves the efficiency of prior discrete samplers while achieving significantly better results (§4). With only 15 denoising steps, APS achieves comparable or better reconstruction quality at up to 6× lower runtime than latent diffusion baselines that require 1000 steps.

| Method | Model | Resolution | GPU (GiB) | Time (s) | #Steps |
|---|---|---|---|---|---|
| DPS (Chung et al., 2023) | PDM | 256 | 10.7 | 277 | 1000 |
| DDRM (Kawar et al., 2022) | PDM | 256 | 5.8 | 4 | 20 |
| PSLD (Rout et al., 2023) | LDM | 512 | 20.9 | 738 | 1000 |
| ReSample (Song et al., 2024) | LDM | 256 | 7.1 | 555 | 500 |
| G2D2 (Murata et al., 2024) | VQ-Diffusion | 256 | 4.7 | 194 | 100 |
| DPS (Chung et al., 2023) | PDM | 256 | 10.7 | 180 | 1000 |
| LDPS (Rout et al., 2023) | LDM | 256 | 15.4 | 190 | 1000 |
| PSLD (Rout et al., 2023) | LDM | 256 | 15.5 | 194 | 1000 |
| G2D2 (Murata et al., 2024) | VQ-Diffusion | 256 | 4.7 | 107 | 100 |
| **APS (ours)** | VQ-Diffusion | 256 | 4.6 | **106** | 100 |
| **APS (ours)** | MMaDA | 256 | 19.2 | **55** | 15 |
| PSLD (Rout et al., 2023) | LDM | 512 | 20.9 | 720 | 1000 |
| **APS (ours)** | MMaDA | 512 | 26.2 | **121** | 15 |
| **APS-L (ours)** | MMaDA | 1024 | 51.8 | **484** | 15 |

*Table 10.* **Noise robustness analysis for 256×256 Super-Resolution (4x) on FFHQ.** APS consistently outperforms G2D2 across varying noise levels ($\sigma \in \{0.01, 0.02, 0.03, 0.04, 0.05\}$) in PSNR, SSIM, and LPIPS. The performance gain of APS remains consistent across different noise levels.

| Measurement Noise ($\sigma$) | Method | PSNR ↑ | SSIM ↑ | LPIPS ↓ |
|---|---|---|---|---|
| 0.01 | G2D2 | 25.98 | 0.739 | 0.328 |
| | APS | **27.52** | **0.784** | **0.268** |
| 0.02 | G2D2 | 25.90 | 0.738 | 0.333 |
| | APS | **27.42** | **0.781** | **0.273** |
| 0.03 | G2D2 | 25.88 | 0.733 | 0.338 |
| | APS | **27.31** | **0.774** | **0.281** |
| 0.04 | G2D2 | 25.64 | 0.726 | 0.341 |
| | APS | **27.04** | **0.764** | **0.290** |
| 0.05 | G2D2 | 25.46 | 0.717 | 0.350 |
| | APS | **26.80** | **0.759** | **0.310** |

**Against prior discrete diffusion (G2D2).** Under the same budget, APS matches or improves G2D2's runtime while yielding better reconstructions: at $256 \times 256$ with VQ-Diffusion both methods use 100 steps, but APS yields higher quality as given in Table 7. Moreover, our 1024×1024 configuration (APS-L) demonstrates better test-time scaling behavior compared continuous diffusion: we keep 15 steps and still obtain substantial quality gains at reasonable cost (484 s).

Importantly, continuous methods struggle to match this performance without prohibitive runtimes. PSLD (Rout et al., 2023) already takes nearly 12 minutes to process a single $512 \times 512$ image and more complex methods such as P2L (Chung et al., 2024) take around 30 minutes for the same resolution. Therefore, training-free posterior sampling using continuous diffusion at very high resolutions such as $1024 \times 1024$ becomes computationally prohibitive.

## C.5. Analysis of Robustness to Measurement Noise

Table 10 evaluates the noise robustness of APS against G2D2 under Gaussian noise with varying standard deviations. APS consistently improves PSNR and SSIM while reducing LPIPS across all noise levels, indicating superior visual fidelity and structural consistency. The performance gap remains stable as $\sigma$ increases, showing that APS maintains high reconstruction quality even under strong measurement corruption. These results, along with experiments spanning different tokenizers (LFQ, VQ-VAE), datasets (FFHQ, ImageNet, StyleAligned), and tasks (inpainting, deblurring, super-resolution), demonstrate the robustness and versatility of our approach.

## C.6. Compared Baselines

We compare our method against state-of-the-art posterior samplers using pixel-/latent-space continuous and discrete diffusion models. Each baseline is evaluated under the same data as ours. We follow the experimental setup from G2D2 and reuse the baseline implementations to ensure a fair comparison. To address the resolution mismatch between our model and the benchmark datasets, we adopt the protocol from G2D2 (Murata et al., 2024). The benchmark images are first upsampled to match the input resolution of our base model MMaDA (Yang et al., 2025b). The forward corruption operator and our posterior sampling method are then applied in this high-resolution space. Finally, the resulting output is downsampled to the original $256 \times 256$ resolution for a fair evaluation. A brief description of each baseline and links to available source code are provided below:

- **DPS** (Chung et al., 2023): A *continuous* diffusion-based method operating in pixel space that solves noisy inverse problems by employing a one-step gradient update in the pixel domain. Source: `https://github.com/DPS2022/diffusion-posterior-sampling`

- **DDRM** (Kawar et al., 2022): A *continuous* diffusion-based method in pixel space, evaluated using the same base models as DPS. Source: `https://github.com/bahjat-kawar/ddrm`

- **DiffPIR** (Zhu et al., 2023): A pixel-space *continuous* diffusion-based method for plug-and-play image restoration. Source: `https://github.com/yuanzhi-zhu/DiffPIR`

- **DAPS** (Zhang et al., 2025): A *continuous* diffusion-based method that employs a decoupled noise annealing strategy to solve inverse problems. Source: `https://github.com/zhangbingliang2019/DAPS`

- **PSLD** (Rout et al., 2023): A latent-space *continuous* diffusion method that solves inverse problems by performing a one-step gradient update in the latent space and optimizing towards the fixed point of a VAE. Source: `https://github.com/LituRout/PSLD`

- **ReSample** (Song et al., 2024): A *continuous* latent-space diffusion method that enforces a hard data-consistency constraint during sampling. Source: `https://github.com/soominkwon/resample`

- **G2D2** (Murata et al., 2024): A *discrete* diffusion posterior sampler that uses a star-shaped noising process and Gumbel-Softmax continuous relaxation to enable gradient guidance in discrete space. We follow the exact implementation and hyperparameters provided in the original paper. Specifically, we use 100 reverse diffusion steps and 30 optimization steps per reverse step. The learning rate and the coefficient for the KL-divergence loss are scheduled logarithmically, as proposed. Source: `https://github.com/sony/g2d2`

- **SGDD** (Chu et al., 2025): A *discrete* diffusion posterior sampler that uses a split Gibbs sampler, reweights probabilities by Hamming distance, and employs rejection sampling via Metropolis–Hastings. Source: `https://github.com/chuwd19/Split-Gibbs-Discrete-Diffusion-Posterior-Sampling`

## C.7. Benchmarks & Metrics

The APS method is evaluated on standard inverse problem benchmarks and is also shown to generalize to more complex tasks, including non-linear inverse problems and training-free stylization. The evaluation uses two main datasets to cover diverse image types and resolutions, with performance measured using Learned Perceptual Image Patch Similarity (LPIPS) ($\downarrow$: lower the better), Peak Signal-to-Noise Ratio (PSNR) ($\uparrow$: higher the better), and Structural Similarity Index (SSIM) ($\uparrow$: higher the better).

**FFHQ (Flickr-Faces-HQ)** (Karras et al., 2019):

- **Dataset Focus**: High-resolution face images.

- **Evaluation Set**: To maintain a fair comparison with prior work, specifically SGDD (Chu et al., 2025), G2D2 (Murata et al., 2024) and DAPS (Zhang et al., 2025), our APS algorithm is evaluated on **100 images** (indices $0, 1, \ldots, 99$) from the FFHQ validation set.

- **Tasks**: The evaluation includes (1) linear inverse problems: SR ($4\times$), Gaussian Deblurring, random inpainting, and motion deblur and (2) nonlinear inverse problems: high dynamic range (HDR) and nonlinear blur.

**ImageNet (Deng et al., 2009):**

- **Dataset Focus**: Diverse natural images.

- **Evaluation Set**: Following the experimental setup by G2D2, a subset of **100 images** is selected from the validation set, ensuring diverse class representation by sampling from classes with indices $0, 10, \ldots, 990$. The specific image list is publicly available in the following text file: `imagenet_val_1k.txt` → https://github.com/XingangPan/deep-generative-prior/.

- **Tasks**: The evaluation includes the same linear and nonlinear inverse problems as in FFHQ.

**Licenses and Usage.** Both FFHQ and ImageNet datasets used in this work are publicly available and licensed for research use. FFHQ dataset, including its documentation and metadata, is distributed by NVIDIA Corporation under the Creative Commons Attribution-NonCommercial-ShareAlike 4.0 International (CC BY-NC-SA 4.0) license. ImageNet data is provided free of charge to researchers, for non-commercial research and educational purposes.

**Question Answering (`Qwen2.5-3B`) (Yang et al., 2025a):** For question answering, we use the `Qwen2.5-3B` model to conduct a targeted evaluation on open-ended generation without additional training. We curate a benchmark of 20 prompts (listed in Table 11) designed to test various capabilities, including creative writing, list creation, and constrained generation (e.g., specific sentence counts). Evaluation is performed using a dual strategy: `Qwen3-0.6B` serves as a reward model to guide diffusion inference, while `Qwen3-8B` acts as an independent oracle judge to assess instruction following capability and semantic coherence.

### C.8. Additional Results

#### C.8.1. GENERAL INVERSE PROBLEMS

Figure 7 illustrates super resolution ($4\times$) results on ImageNet (Deng et al., 2009). Competing methods—DPS (Chung et al., 2023), DDRM (Kawar et al., 2022), PSLD (Rout et al., 2023), and ReSample (Song et al., 2024)—recover coarse structures but often yield blurry textures or color shifts, while G2D2 (Murata et al., 2024) sharpens details at the cost of noticeable artifacts. In contrast, APS produces sharper and more natural reconstructions across both object and animal categories, closely adhering to the ground truth.

Similarly, Figure 8 shows results for Gaussian deblurring. Continuous methods again capture overall structure but leave residual blur or noise, and G2D2 (Murata et al., 2024) partially enhances details yet struggles with fine textures. APS delivers cleaner and more faithful reconstructions, effectively balancing sharpness and natural appearance across diverse scenes.

Figure 9 compares APS against continuous (DPS (Chung et al., 2023), DDRM (Kawar et al., 2022), PSLD (Rout et al., 2023), ReSample (Song et al., 2024)) and discrete (G2D2 (Murata et al., 2024)) approaches on FFHQ super resolution. Continuous methods capture overall facial structure but tend to oversmooth, leaving blurred or distorted skin textures, while ReSample introduces strong artifacts. G2D2 sharpens details but produces unnatural appearances. APS, by contrast, reconstructs sharper features with natural skin tones and clean edges, yielding perceptually faithful faces across diverse examples and demonstrating clear advantages for high-resolution face restoration.

Figure 10 presents Gaussian deblurring on FFHQ (Karras et al., 2019). DPS (Chung et al., 2023) and DDRM (Kawar et al., 2022) again oversmooth, suppressing fine facial detail; PSLD (Rout et al., 2023) and ReSample (Song et al., 2024) introduce ringing and plastic-like skin; and G2D2 (Murata et al., 2024) struggles to remove noise from the noisy measurements (Input), creating halos along edges. APS recovers crisp structures such as hair, eyeglass frames, and lip contours while preserving

*Table 11.* **Question Answering Benchmark Prompts.** The evaluation set consists of 20 diverse instructions, including creative writing, list generation, and constrained formatting tasks created using `Qwen2.5-3B`.

| ID | Prompt | ID | Prompt |
|---|---|---|---|
| 1 | Write a short paragraph describing your favorite food. | 11 | Enumerate three favorite hobbies you have incorporated into your weekend routine. |
| 2 | Create a list of three things you love most about your weekend. | 12 | Write one sentence describing what you did last Saturday. |
| 3 | Write a two-sentence story starting with 'Once upon a time, I found a mysterious letter in my mailbox.' | 13 | Create a sentence about your day, detailing what you did at home. |
| 4 | Provide reasons why you like singing. | 14 | Write about one day in your life, focusing on what you watched on TV. |
| 5 | Write about your pet's daily routine. | 15 | Create a list of three things you like about your city. |
| 6 | Explain in two sentences why spending time outdoors is important. | 16 | Describe your favorite book in one sentence. |
| 7 | Create a list of three things you did last Saturday. | 17 | Write a short summary (5 sentences) of a movie you recently watched. |
| 8 | Write down your dream for your dream job, describing what you see in it. | 18 | Describe one thing you did today that made you happy. |
| 9 | Create a simple diary entry about your best day at school. | 19 | Write (4 sentences) why home cooking is important to you. |
| 10 | Write a paragraph that explains why studying is beneficial. | 20 | Create a paragraph (5 sentences) about your favorite day in the park. |

natural highlights and avoiding artifacts, producing reconstructions that are perceptually closer to the ground truth and aligned with our quantitative improvements.

Figure 11 shows additional qualitative results on complex linear and nonlinear inverse problems on FHHQ dataset (Karras et al., 2019), showcasing the performance of both APS and APS-L.

### C.8.2. TEXT-GUIDED LARGE BLOCK INPAINTING

Large block inpainting is a particularly challenging setting for generative models, as it requires filling in large missing regions with semantically coherent and high-fidelity content guided by text descriptions. One interesting application of large block inpainting is virtual try-on (Han et al., 2018; Zhu et al., 2024), where models must realistically generate clothing or accessories consistent with both a reference garment and the overall body pose.

In a typical real-world fashion catalog, the full body images naturally have rectangular aspect ratios. Since most existing multimodal foundation models are trained on square images (e.g., 512×512 for MMaDA), we fine-tuned MMaDA on a collection of 1024×512 full-body images curated from a fashion dataset (Zhu et al., 2024), following preprocessing with segmentation-based cropping and padding to standardize framing. This adaptation enables our base model to better handle rectangular image structures. For training, we have randomly selected 100K images from this dataset, providing a diverse and challenging testbed for inpainting at scale.

**Comparison.** We compare APS against large-scale continuous diffusion baselines: Imagen3 (Baldridge et al., 2024), Flux (Black Forest Labs, 2024), and HDPainter (Manukyan et al., 2025). As illustrated in Figure 12, our method generates realistic clothing textures, with stronger alignment to the reference prompts and fewer artifacts (red boxes highlight failure regions of competing methods). In contrast, Imagen3 and Flux often introduce distorted or inconsistent garment regions, while HD-painter produces less faithful completions with mismatched styles. APS leverages discrete diffusion's ability to directly reweight categorical distributions under posterior guidance, yielding visually compelling and semantically accurate completions.

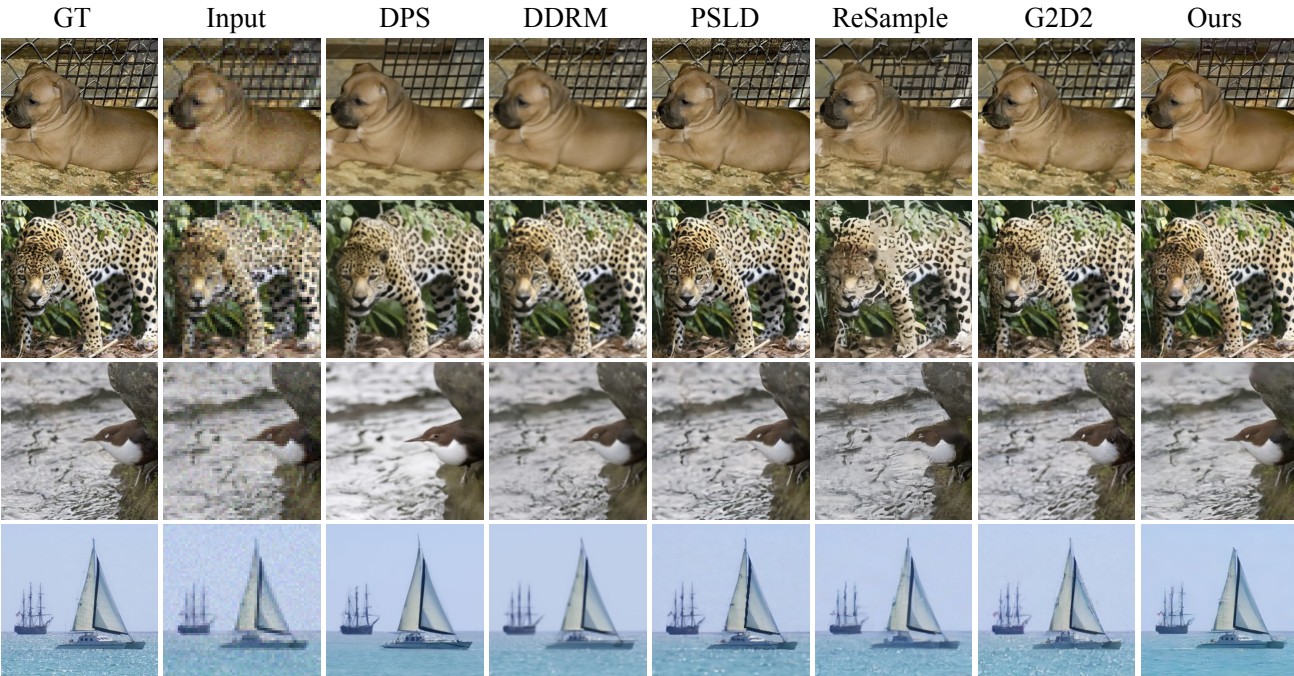

*Figure 7.* **Additional qualitative results for super resolution** ($4\times$) **on ImageNet.** Compared to continuous baselines (DPS, DDRM, PSLD, ReSample) and the discrete baseline G2D2, APS produces sharper details and more faithful reconstructions across diverse examples. For instance, in the second row, the leopard's eyes are reconstructed with fine detail, noticeably better than the baselines.

### C.8.3. STYLIZATION

Figure 13 presents additional qualitative results on reference-based stylization. In each case, our APS optimizer conditions on a single reference style image and a text prompt describing the desired content. The outputs demonstrate that APS effectively transfers diverse artistic styles—including tattoo art, steampunk mechanical, psychedelic art, and tribal tattoo—while preserving semantic fidelity to the target content. These results highlight the robustness and versatility of APS in handling a wide range of style-content combinations.

Figure 14 provides a qualitative comparison of our full method (MMaDA + APS) against the base model (MMaDA) and state-of-the-art continuous diffusion approaches. This experiment is designed to demonstrate the novel capability of discrete diffusion models for challenging nonlinear style transfer, not solely to outperform continuous alternatives. We observe that competing methods struggle to balance style fidelity with content alignment. Training-free methods like StyleAligned (Hertz et al., 2023) and InstantStyle (Wang et al., 2024) often drift towards generic textures. Conversely, the training-based StyleDrop (Sohn et al., 2023) tends to overfit to superficial color patterns, which compromises semantic coherence with the text prompt. Our base model, MMaDA (Yang et al., 2025b), maintains reasonable content fidelity but fails to transfer fine-grained style attributes, such as material textures or stroke-level details. In contrast, our full method (MMaDA + APS) consistently produces outputs that preserve the reference style while maintaining strong semantic alignment. For instance, our result for the "letter" prompt retains the intricate, flowing smoke design, while the "milkshake" example accurately captures the specified retro diner aesthetic. These results highlight the effectiveness of anchored posterior sampling in discrete diffusion models for complex style transfer tasks.

### C.8.4. ADDITIONAL QUANTITATIVE RESULTS

We perform a larger-scale evaluation for $4\times$ super resolution on FFHQ, extending our analysis to 1000 samples. Our results are compared against the numbers reported for the same task in G2D2 (Murata et al., 2024). Importantly, our observation in the main draft extends to the larger-scale setting and our APS algorithm consistently outperforms G2D2 in all metrics, as given in Table 12.

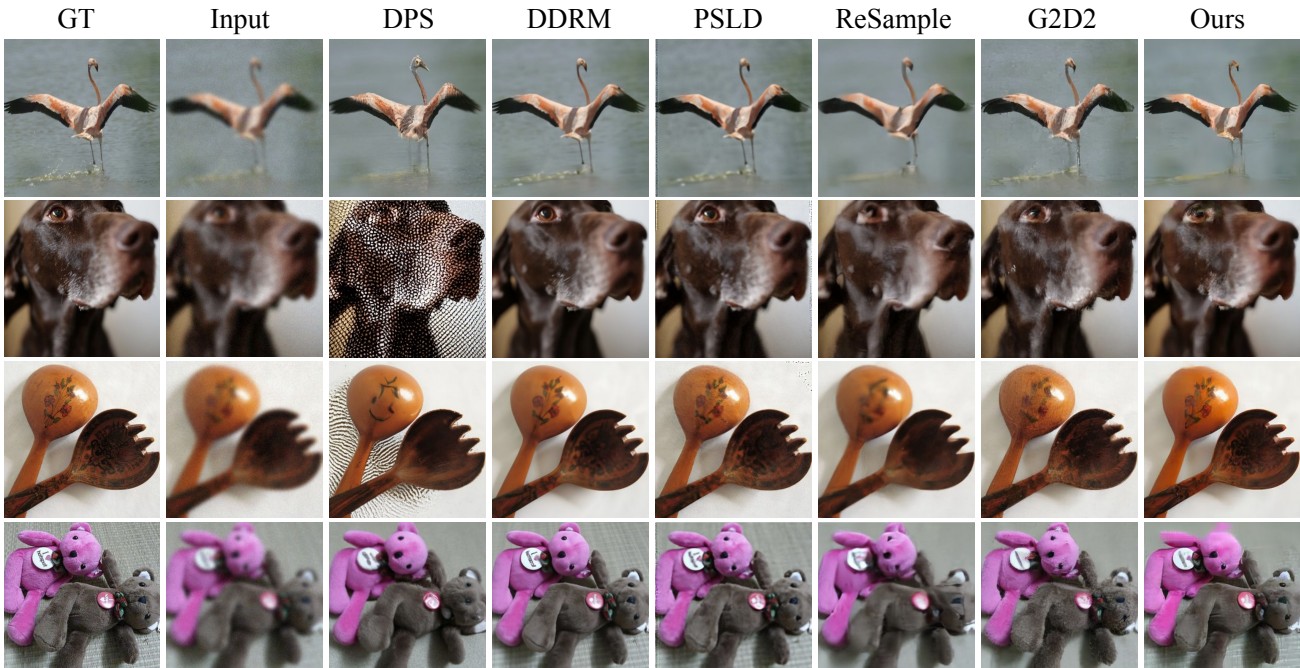

*Figure 8.* **Additional qualitative results for Gaussian deblurring on ImageNet.** Compared to continuous baselines (DPS, DDRM, PSLD, ReSample) and the discrete baseline G2D2, our APS sampler achieves sharper textures, less artifacts, and more faithful reconstructions across diverse examples. For instance, in the third row, both the floral pattern on the outside of the wooden spoon and the artistic pattern inside it are accurately preserved by our method, whereas most baselines either miss or misrepresent these details.

## C.9. Limitations

Despite achieving state-of-the-art performance among discrete diffusion samplers on (1) complex (linear and nonlinear) inverse problems and (2) reference-based stylization tasks, APS exhibits the following limitations:

1. **Tokenizer quality.** MMaDA uses MagVIT-v2 (Yu et al., 2024) tokenizer which has limited reconstruction quality compared to modern visual tokenizers (Black Forest Labs, 2024). Therefore, future improvements in discrete visual tokenizers could directly benefit APS.

2. **Base model performance.** Discrete diffusion backbones are still in an early stage of development and, at present, underperform large-scale continuous diffusion foundation models such as Flux (Black Forest Labs, 2024), SD3.5 (Esser et al., 2024), and Imagen (Baldridge et al., 2024) in unconditional generative quality. Nevertheless, our theoretical and empirical results indicate that discrete diffusion shows promising potential for posterior sampling and could, with further advances, become a viable alternative to the continuous models that dominate current practice.

3. **Stylization dependence.** The performance of APS in stylization tasks depends both on the pretrained discrete diffusion backbone and the quality of the style feature extractor (e.g., CSD). If the style extractor has not been trained on a particular style, our sampler struggles to transfer it faithfully, limiting its applicability to out-of-distribution styles.

4. **Approximate posterior sampling.** APS optimizes a surrogate variational objective and does not guarantee asymptotically exact posterior samples. While this approximation is essential for tractability in high-dimensional discrete spaces, it may introduce bias relative to exact discrete MCMC methods, which remain computationally infeasible at scale.

**Failure Cases.** Figure 15 illustrates failure cases of our approach in stylization. We observe that APS sometimes produces over-smoothed outputs when the reference style is out-of-distribution, or introduces artifacts when the measurement operator is poorly aligned with the pretrained backbone. These examples highlight opportunities for improving robustness and generalization.

| GT | Input | DPS | DDRM | PSLD | ReSample | G2D2 | Ours |

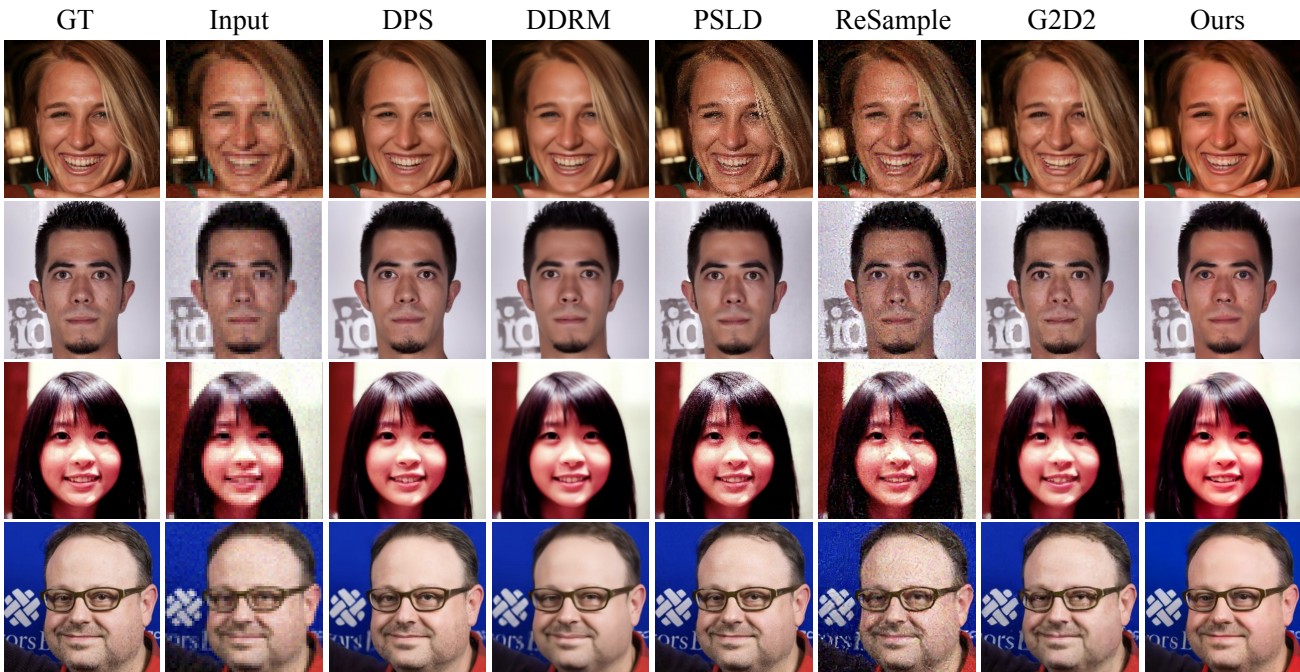

*Figure 9.* **Additional qualitative results for super resolution** (4×) **on FFHQ.** Continuous baselines (DPS, DDRM, PSLD, ReSample) generate plausible but oversmoothed faces, while the discrete baseline G2D2 often introduces artifacts. In contrast, our APS algorithm reconstructs sharper, more natural faces that closely align with the ground truth. For example, in the second row, our method successfully recovers the small mole on the person's left cheek, a detail overlooked by the baselines.

## C.10. Reproducibility Statement

Our experiments are built upon the publicly available MMaDA codebase (Yang et al., 2025b). All modifications and implementation details are described in Appendix C.1, which includes the pseudocode in Algorithm 1 and the specific parameters used for every experiment. Furthermore, Appendix C.2 provides comprehensive ablation studies and hyperparameter sweeps (Table 8). The experiments utilize the widely-used public datasets FFHQ (Karras et al., 2019) and ImageNet (Deng et al., 2009). The combination of a public codebase and datasets, along with our detailed Algorithm 1 and parameters (§C.2), should ensure that our results are readily reproducible.

| GT | Input | DPS | DDRM | PSLD | ReSample | G2D2 | Ours |
|----|-------|-----|------|------|----------|------|------|

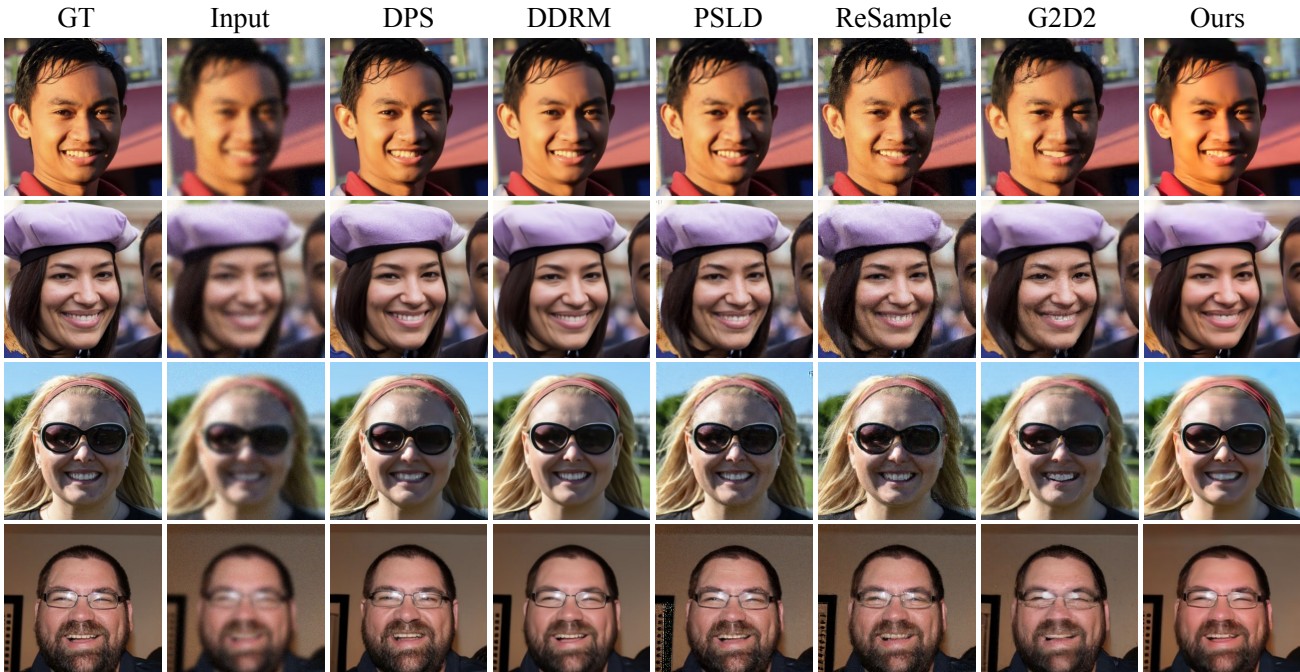

*Figure 10.* **Additional qualitative results for Gaussian deblurring on FFHQ.** Our proposed approach recovers sharper facial details and edges with fewer artifacts (e.g., reduced ringing and texture distortions), leading to more natural reconstructions. For instance, in the first row, our method accurately reconstructs the hair strands and their shadow on the face, whereas the prior baselines fail to capture these details as precisely.

*Table 12.* **Super Resolution (4×) on FFHQ.** Performance comparison of APS against prior works. Continuous methods are shaded gray.

| Type | Method | LPIPS ↓ | PSNR ↑ | SSIM ↑ |
|------|--------|---------|--------|--------|
| Pixel-domain | DPS | 0.238 | 26.07 | 0.756 |
| | DDRM | 0.252 | 28.09 | 0.804 |
| LDM | PSLD | 0.282 | 27.12 | 0.757 |
| | ReSample | 0.508 | 23.07 | 0.445 |
| Discrete | G2D2 | 0.265 | 27.29 | 0.763 |
| | G2D2 w/ Markov noise process | 0.369 | 25.15 | 0.699 |
| Mask | **APS** | **0.232** | **27.81** | **0.808** |

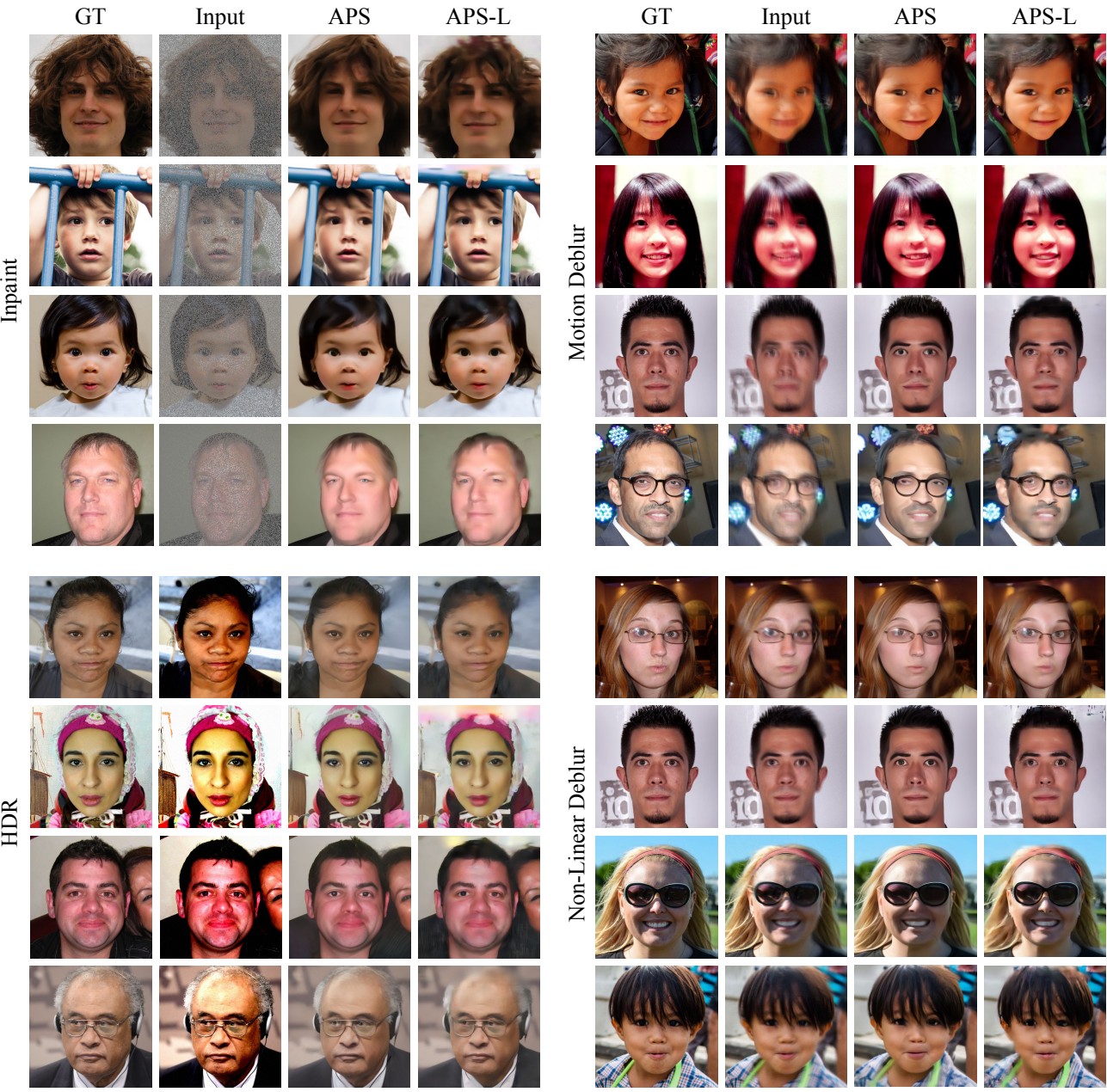

*Figure 11.* **Qualitative results on FFHQ** for linear (top 4 rows: inpaint and motion deblur) and nonlinear (bottom 4 rows: HDR and non-linear deblur) inverse problems .

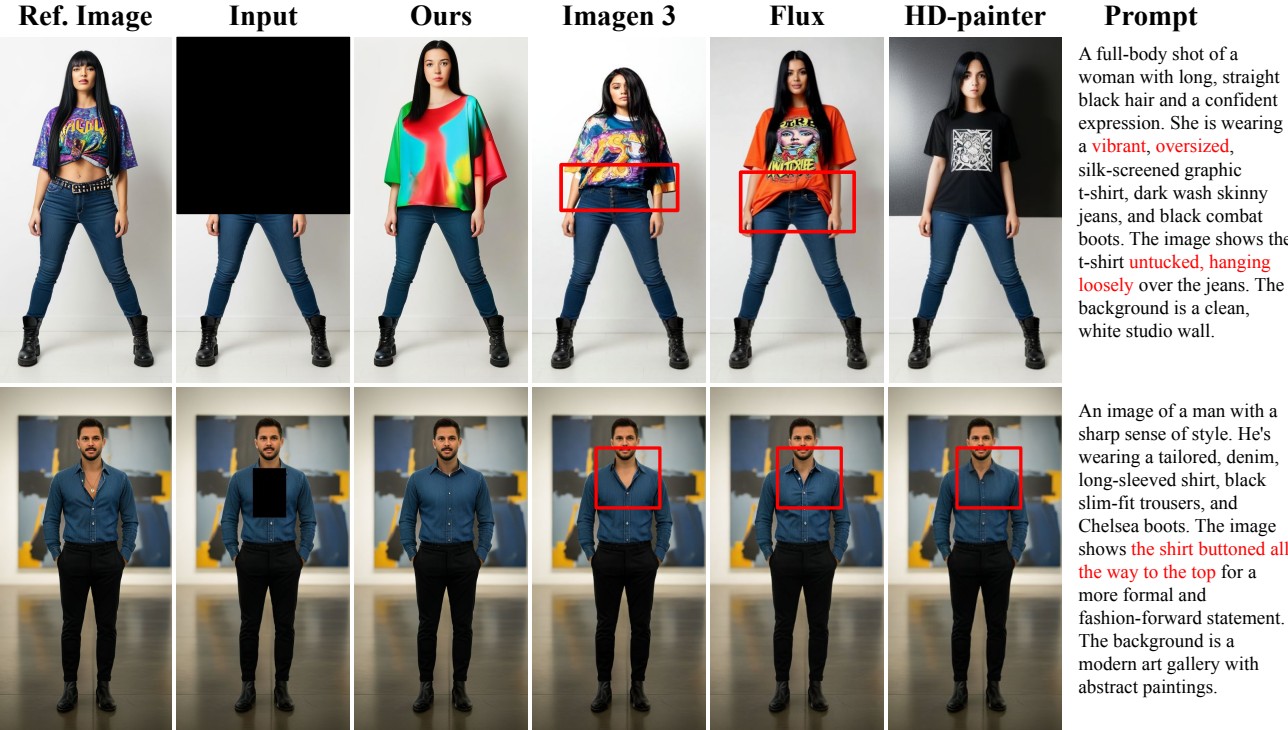

*Figure 12.* **Text-guided large-block inpainting on high-resolution (1024×512) fashion images.** APS generates coherent garment completions with stronger prompt alignment compared to prior methods (Imagen 3 (Baldridge et al., 2024), Flux (Black Forest Labs, 2024), and HD-Painter (Manukyan et al., 2025)). Red boxes highlight incorrect or prompt-inconsistent synthesis in competing approaches. In the first row, the prompt specifies an untucked, oversized T-shirt with vibrant colors—details missed by Imagen 3, Flux, and HD-Painter, respectively. In the second row, our approach correctly buttons the shirt all the way to the top.

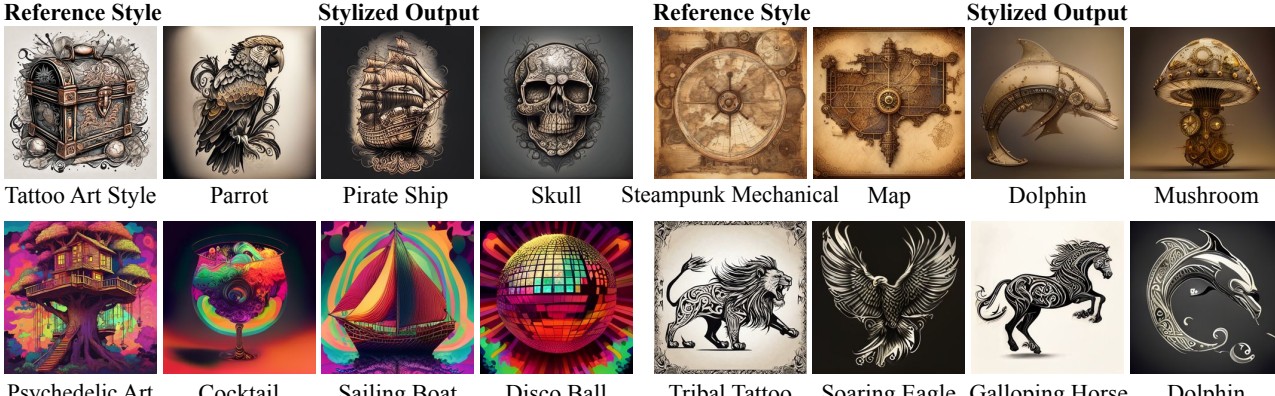

*Figure 13.* **Additional qualitative results on reference-based stylization.** We show four style-content combinations. For each, our APS optimizer conditions on a single Reference Style image and a text prompt describing the content to generate the Stylized Output images.

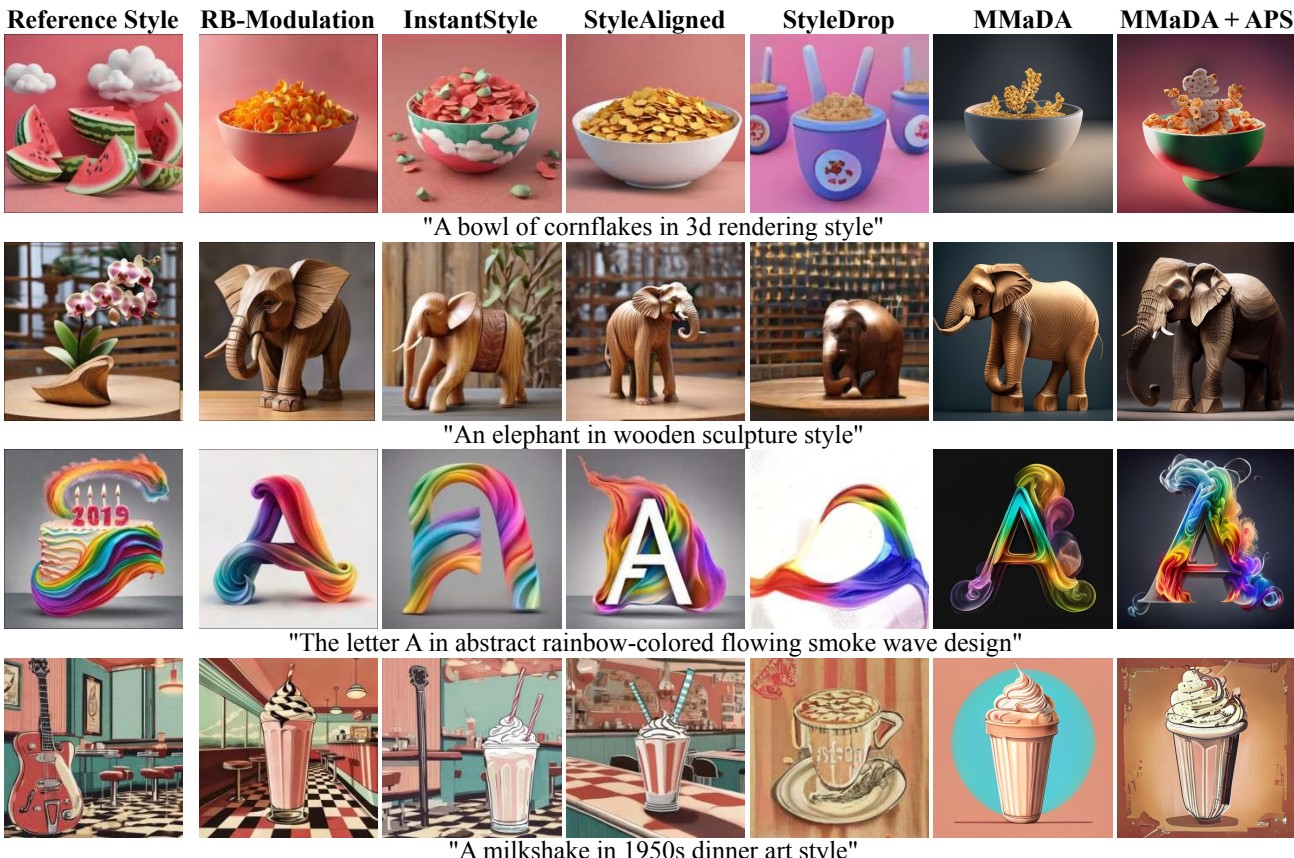

| Reference Style | RB-Modulation | InstantStyle | StyleAligned | StyleDrop | MMaDA | MMaDA + APS |

"A bowl of cornflakes in 3d rendering style"

"An elephant in wooden sculpture style"

"The letter A in abstract rainbow-colored flowing smoke wave design"

"A milkshake in 1950s dinner art style"

*Figure 14.* **Additional qualitative comparison on reference-based stylization.** We compare our full method (MMaDA + APS) with the base model (MMaDA) and several state-of-the-art continuous diffusion methods across four style-prompt pairs. For each row, all methods use the same Style Reference image (left) and text prompt (shown below the images).

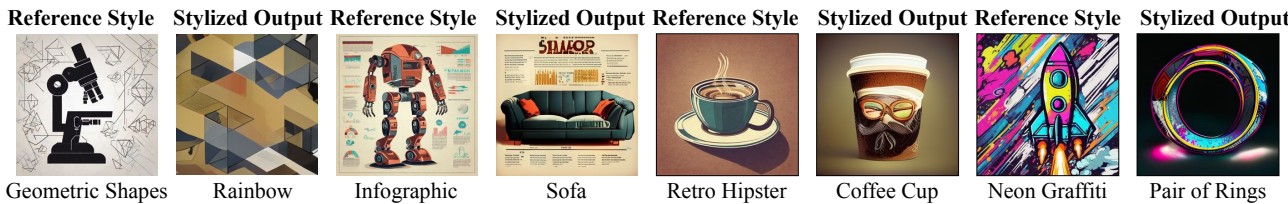

| Reference Style | Stylized Output | Reference Style | Stylized Output | Reference Style | Stylized Output | Reference Style | Stylized Output |

| Geometric Shapes | Rainbow | Infographic | Sofa | Retro Hipster | Coffee Cup | Neon Graffiti | Pair of Rings |

*Figure 15.* **Failure cases of APS.** Under extreme circumstances such as out-of-distribution styles or highly nonlinear measurement operators, our method can sometimes fail, producing over-smoothed reconstructions or noticeable artifacts.

