# OpenReview forum: "Test-Time Anchoring for Discrete Diffusion Posterior Sampling"
_ICML.cc/2026/Conference — ICML 2026 regular_

### Official Review · Reviewer_HKop · 2026-03-11

**Soundness:** 3
**Presentation:** 3
**Significance:** 3
**Originality:** 3
**Overall Recommendation:** 5
**Confidence:** 2

**Summary:**

This paper introduces anchored posterior sampling for solving inverse problems using pretrained discrete masked diffusion models. The approach proposes quantized expectation for gradient-like guidance in discrete diffusion and anchored remasking to adaptively decode important anchor tokens and unmasks them early. Experiments on FFHQ and ImageNet demonstrate good results among discrete diffusion samplers for super-resolution, deblurring, inpainting, HDR recovery, etc.

**Compliance With Llm Reviewing Policy:**

Affirmed.

**Final Justification:**

Rebuttal mostly addressed the issues, and I kept the positive rating.

**Key Questions For Authors:**

See weaknesses.

**Limitations:**

See weaknesses.

**Strengths And Weaknesses:**

Strengths:

The proposed method overall is motivated and sound. Theoretical analyses are provided for training and test-time bounds. Experiments on multiple inverse image tasks show good performance compared to other discrete diffusion samplers and are competitive with continuous diffusion samplers. The method also generalises to stylisation and language tasks. Excellent visual examples are provided. The supplementary material is long and comprehensive.

Weaknesses:

1. In Table 2, although compared to continuous baselines, the results are not as good as in Table 1.

2. Test-time optimisation will significantly reduce the efficiency of the sampling process. The sampling cost/speed should be added to all major tables (e.g., Tables 1 and 2) to compare them with other methods. This also helps to validate the claim, such as lower inference cost compared to continuous diffusion models.

3. The formula in Theorem 3.1 is too long and spans multiple lines, which is ugly.

---

> ### Author Rebuttal · Authors · 2026-03-31
>
> Dear Reviewer HKop,
>
> Thank you for your constructive feedback and for noting that our method is **"motivated and sound"** with **"excellent visual examples."** We appreciate your recognition of our theoretical analyses and the fact that our results are **"competitive with continuous diffusion samplers."** Below, we address your questions regarding the comparative results and efficiency.
>
> ---
>
> > **Q1. In Table 2, although compared to continuous baselines, the results are not as good as in Table 1.**
>
> **A1.** We would like to clarify the context of these two tables. In **Table 1**, we compare APS against both discrete and continuous baselines on standard tasks. Here, APS outperforms existing discrete samplers by a significant margin and remains competitive with continuous methods.
>
> In **Table 2**, we evaluate more complex non-linear inverse problems. Because existing discrete diffusion samplers have not been benchmarked on these tasks, we compare exclusively against established continuous alternatives. While continuous models sometimes hold a numerical edge in specific metrics on these complex tasks, APS remains within a comparable range while offering the unique benefits of multi-modal generation and editing using masked diffusion priors. Our goal was to demonstrate that discrete models, through APS, are now viable for high-complexity tasks previously reserved for continuous models.
>
> ---
>
> > **Q2. Test-time optimisation will significantly reduce the efficiency of the sampling process. The sampling cost/speed should be added to all major tables...**
>
> **A2.** We agree that transparency regarding inference speed is crucial for validating efficiency. While these metrics were previously located in the supplementary material (**Table 7, Appendix C.4** and **Table 8, Appendix C.5.2**), we will follow your suggestion and integrate these directly into Tables 1 and 2 in the main draft.
>
> As noted in the response to Reviewer mFey, although APS uses an inner optimization loop, it avoids backpropagating through the massive 8B parameter backbone, making it faster per iteration than continuous methods like DPS or PSLD.
>
> ---
>
> > **Q3. The formula in Theorem 3.1 is too long and spans multiple lines, which is ugly.**
>
> **A3.** We will restructure the formula in Theorem 3.1 in the final version, ensuring that the formula is both mathematically rigorous and visually concise.

---

> > ### Author Rebuttal · Reviewer_HKop · 2026-04-03
> >
> > Mostly resolved

---

### Official Review · Reviewer_TqMd · 2026-03-12

**Soundness:** 3
**Presentation:** 2
**Significance:** 3
**Originality:** 3
**Overall Recommendation:** 4
**Confidence:** 3

**Summary:**

This paper introduces Anchored Posterior Sampling for discrete diffusion models. They fix the unconditional model and learn the adaptation network $\varphi$ at test time. They propose a quantized expectation for practical computation and anchored remasking for adoptive decoding. The authors conduct experiments on various inverse problems to verify the effectiveness of the proposed method.

**Compliance With Llm Reviewing Policy:**

Affirmed.

**Final Justification:**

The strengths of this paper mainly lie in originality and clarity. During the rebuttal, the authors provide some necessary comparison with a series of works, which somehow improves the soundness.

**Key Questions For Authors:**

1. How to deal with $\sum_{i=1}^T \mathbb{E} _{q\left(Z_t \mid \mathbf{x}\right)}\left[\sum _{l=1}^L \log \mathcal{Z} _{\varphi}^l\left(Z_t, \mathbf{y}\right)\right]$ in practice?

2. Could you explain what is the meaning of "Uniform (Mask)" and "Mask" in Table 1 and "Mask" and "Discrete" in Table 2?

**Strengths And Weaknesses:**

Strengths:
1. Studying training-free posterior sampling for the discrete diffusion model is novel.
2. Quantized expectation and anchored remasking are novel.
3. Experimental results are comprehensive.

Weaknesses:
The paper lacks a comparison with some important continuous diffusion models for posterior sampling/ inverse problem [1, 2,...].

[1]: Kawar, B., Elad, M., Ermon, S., & Song, J. (2022). Denoising Diffusion Restoration Models. ArXiv, abs/2201.11793.

[2]: Song, J., Vahdat, A., Mardani, M., & Kautz, J. (2023). Pseudoinverse-Guided Diffusion Models for Inverse Problems. International Conference on Learning Representations.

---

> ### Author Rebuttal · Authors · 2026-03-31
>
> Dear Reviewer TqMd,
>
> Thank you for your constructive feedback and for recognizing the **novelty** of studying training-free posterior sampling in the discrete domain. We appreciate your positive comments on the **originality** of our Quantized Expectation and Anchored Remasking components, as well as the **comprehensiveness** of our experimental results. Below, we address your questions.
>
> ---
>
> > **Q1. The paper lacks a comparison with some important continuous diffusion models for posterior sampling/ inverse problem [1, 2,...].**
>
> **A1.** While our primary focus is advancing posterior sampling within the **discrete diffusion** setting, an area that has remained relatively underexplored, we agree that contextualizing these results against continuous baselines is valuable. In the paper, we included comparisons with representative continuous methods and the state-of-the-art DAPS (CVPR, 2025). We thank the reviewer for suggesting DDRM [1] and PiGDM [2]; we will include a discussion of these important works in our revised manuscript to provide a more complete picture of the landscape of inverse problem solvers.
>
> ---
>
> > **Q2. How to deal with the normalizing factor in practice?**
>
> **A2.** There are two key practical steps in dealing with the normalizing factor:
>
> 1. As described in **Implication 3.2**, we treat the normalization factor using a `stop-grad` operation during the inner-loop optimization. We then pass the optimized logits through a $softmax$ function to ensure the result remains a valid probability distribution in practice.
>
> 2. As discussed in **Section 3.1**, we follow the common approach used in practice by focusing the loss computation on the current time step $t$, effectively ignoring the computationally expensive outer summation while maintaining a tight surrogate for the posterior.
>
> ---
>
> > **Q3. Could you explain what is the meaning of "Uniform (Mask)" and "Mask" in Table 1 and "Mask" and "Discrete" in Table 2?**
>
> **A3.** We apologize for the terminology confusion and will clarify these labels in the final version:
>
> * **Table 1:** "Uniform (Mask)" refers to priors using a mixture of uniform transition noise and masking (e.g., VQ-Diffusion). "Mask" refers to priors that rely purely on the [MASK] token transition function.
>
> * **Table 2:** Both "Mask" and "Discrete" were intended to categorize our methods (APS and APS-L) as discrete-domain solvers to distinguish them from continuous-domain solvers like DPS (pixel) and PSLD (latent).
>
>
> To avoid confusion, we will standardize these labels to **"Mask"** and explicitly clarify this in the table caption.
>
> ### References
>
> [1] Kawar et al. "Denoising Diffusion Restoration Models." arXiv:2201.11793 (2022).
>
> [2] Song et al. "Pseudoinverse-Guided Diffusion Models for Inverse Problems." ICLR (2023).

---

> > ### Author Rebuttal · Reviewer_TqMd · 2026-04-04
> >
> > Thanks for the rebuttal. I maintain my positive rating.

---

### Official Review · Reviewer_mFey · 2026-03-13

**Soundness:** 3
**Presentation:** 3
**Significance:** 3
**Originality:** 3
**Overall Recommendation:** 4
**Confidence:** 3

**Summary:**

The paper presents a training-free posterior sampling method for masked discrete diffusion models, targeting inverse problems and controllable generation. The premise is well-motivated: existing posterior sampling approaches for discrete diffusion either rely on weak derivative-free rewards, continuous relaxations like Gumbel-Softmax, or split Gibbs procedures that scale poorly with sequence length. To address this, the authors propose Anchored Posterior Sampling (APS), which combines (i) quantized expectation, a differentiable surrogate that maps token distributions to expected embeddings, quantizes them back, and optimizes measurement consistency, and (ii) anchored remasking, which preferentially unmasks high-confidence, informative tokens early in the reverse process. The paper shows improvements on image inverse problems such as super-resolution, deblurring, inpainting, HDR, and nonlinear blur, and also includes extensions to stylization and diffusion language models for question answering.

**Compliance With Llm Reviewing Policy:**

Affirmed.

**Key Questions For Authors:**

- Can the authors provide stronger compute-matched comparisons to prior discrete samplers, especially given the heavy inner-loop optimization at each reverse step?

- Can the authors add a clearer ablation over optimization budget: number of reverse steps, number of inner optimization steps, perceptual-loss weight, and anchor-threshold schedule.

- For QA, can you evaluate on more standard benchmark with standard automatic evaluation?

**Limitations:**

yes

**Strengths And Weaknesses:**

Strengths:

- Method is fairly novel and technically interesting. The quantized-expectation idea is a clean way to get gradient-like guidance in discrete diffusion without relying on a fully continuous relaxation, and the anchored remasking component is a sensible adaptation of anchor-first decoding to posterior sampling.The empirical evaluation is broad.

Weaknesses:

- Lack of sufficiently compute-matched evaluation. The method is described as efficient at test time, but in practice the appendix uses 15 reverse steps with 100 inner optimization steps per reverse step,

- The ablations do not yet seem deep enough on the central optimization knobs. From the appendix snippet, APS has several tunable quantities: number of optimization steps, perceptual-loss coefficient, and the remasking threshold schedule. These choices are central to the method, but from the main paper it is not obvious how robust results are to them

- Some of the non-vision demonstrations are relatively weakly validated. The QA benchmark is only 20 questions, evaluated with Qwen models as reward model and judge, which makes it hard to know whether the improvements hold outside of this setting.

---

> ### Author Rebuttal · Authors · 2026-03-31
>
> Dear Reviewer mFey,
>
> Thank you for your constructive feedback and for recognizing that our work is **"fairly novel and technically interesting."** We appreciate your assessment that the **"premise is well-motivated"** and that Quantized Expectation is a **"clean way to get gradient-like guidance."** Below, we address your concerns regarding compute-matched evaluations and ablation studies.
>
> ---
>
> > **Q1. Lack of sufficiently compute-matched evaluation. The method is described as efficient at test time, but in practice the appendix uses 15 reverse steps with 100 inner optimization steps per reverse step.**
>
> **A1.** We would like to clarify the nature of this "inner-loop" optimization. While APS uses 100 optimization steps, these are **computationally lightweight** because they only involve backpropagation with respect to logit vectors through a small decoder.
>
> Importantly, unlike continuous diffusion baselines (e.g., DPS or PSLD), **APS does not require backpropagation through the denoising backbone**, which in our case is an 8B parameter model (MMaDA). This results in a significant wall-clock time advantage. As requested, we point to **Appendix C.4, Table 7 (page 29)**, where we provide a direct compute-matched comparison against G2D2. Furthermore, Table 8 shows that even with the inner loop, APS remains faster while achieving comparable/better reconstruction quality than competing methods.
>
> ---
>
> > **Q2. The ablations do not yet seem deep enough on the central optimization knobs... it is not obvious how robust results are to them.**
>
> **A2.** We understand the concern regarding the robustness of hyper-parameters.
>
> * **Optimization Steps:** Appendix C.3.4 (Table 6) shows that performance saturates quickly, and the method is not overly sensitive to the exact step count beyond a reasonable threshold.
> * **Perceptual Loss:** Table 5 (Appendix C.2) explores the coefficient $\lambda$. We found a wide range of values yields consistent results.
> * **Remasking Threshold:** We maintain the default schedule recommended by the generative prior (MMaDA) across all experiments, demonstrating that APS does not require task-specific tuning of the diffusion schedule itself.
>
> For clarity, we’ll move the robustness analysis from the appendix to the main paper.
>
> ---
>
> > **Q3. Some of the non-vision demonstrations are relatively weakly validated. The QA benchmark is only 20 questions...**
>
> **A3.** While the primary focus of this paper is on image domain, the proposed algorithm generalizes to text domain as well. We had initially included a question-answering task as a proof of concept.
>
> To provide more robust evidence of generalizability, we have expanded our evaluation to include two standard benchmarks with 100 examples each. We evaluate on RewardBench-2 (constraint adherence) and RM-Bench (human preference alignment). As shown below, APS consistently outperforms the baseline:
>
> | Dataset | Method | Reward |
> | :--- | :--- | :--- |
> | RewardBench-2 | **APS (Ours)** | **1.7937** |
> | RewardBench-2 | Baseline | 1.7145 |
> | RM-Bench | **APS (Ours)** | **1.8690** |
> | RM-Bench | Baseline | 1.8281 |
>
> *Note: Higher reward scores indicate better alignment and performance.* These results suggest that the benefits of APS transition effectively from vision to language domains.

---

> > ### Author Rebuttal · Reviewer_mFey · 2026-04-03
> >
> > Thanks for the rebuttal. I maintain my positive rating.

---

### Official Review · Reviewer_g3FB · 2026-03-14

**Soundness:** 4
**Presentation:** 4
**Significance:** 4
**Originality:** 4
**Overall Recommendation:** 5
**Confidence:** 4

**Summary:**

This paper studies posterior sampling for masked discrete diffusion models. The authors propose Anchored Posterior Sampling (APS), a training-free method built on two main ideas: quantized expectation, which introduces a differentiable surrogate for measurement-guided posterior sampling in discrete token space, and anchored remasking, which adaptively prioritizes informative tokens during decoding. Experiments cover a range of image inverse problems, as well as stylization and a small language-model guidance setting. Overall, the paper tackles an interesting and relatively underexplored problem in discrete diffusion.

**Compliance With Llm Reviewing Policy:**

Affirmed.

**Key Questions For Authors:**

The core problem addressed in this paper is how to propagate gradients over discrete indices. This is not a totally new conceptual problem, as there has also been prior work on effective gradient propagation over codebook indices in discrete tokenizers [1]. Could the authors discuss the differences between this paper and those lines of work?

[1] F. Shi et al. Scalable Image Tokenization with Index Backpropagation Quantization, ICCV 2025

**Limitations:**

yes

**Strengths And Weaknesses:**

Strengths:
1. The problem is well motivated, posterior sampling is well developed for continuous diffusion, but much less so for masked discrete diffusion.
2. It is supported by a rigorous mathematical derivation of the bound.
3. The efficiency angle is potentially appealing. The paper argues that APS remains competitive with continuous diffusion methods while being substantially cheaper at inference time.
4. The experimental section is fairly broad. In addition to standard inverse problems on FFHQ and ImageNet, the paper also includes stylization and a language-model setting, which helps demonstrate the potential scope of the approach.

Weaknesses:
1. The extension to question answering is interesting, but the benchmark appears to contain only 20 questions, which makes the evidence rather limited.
2. Posterior sampling is also being used less and less in continuous diffusion, especially because the additional inference cost overhead becomes unacceptable for large scale generation models. This hinders the practical value of posterior sampling.

---

> ### Author Rebuttal · Authors · 2026-03-31
>
> viewer g3FB,
>
> Thank you for your thoughtful comments. We appreciate your recognition that our paper studies an **interesting and relatively underexplored** domain in posterior sampling using discrete diffusion. We are encouraged by your assessment that our mathematical derivation of the bound is "rigorous" and that our method remains "competitive with continuous diffusion" while being more efficient. Below, we address your questions.
>
> ---
>
> > **Q1. The extension to question answering is interesting, but the benchmark appears to contain only 20 questions, which makes the evidence rather limited.**
>
> **A1.** While the primary focus of this paper is on image domain, the proposed algorithm generalizes to text domain as well. We included a question-answering task as a proof of concept. However, we agree that a larger sample size strengthens the empirical evidence.
>
> Following your suggestion, we have expanded our evaluation to include two additional benchmarks with 100 examples each. These benchmarks evaluate the model's ability to adhere to complex constraints (RewardBench 2) and its alignment with human preference scores (RM-Bench). As shown below, APS consistently outperforms the baseline across these larger-scale settings:
>
> | Dataset | Method | Reward |
> | :--- | :--- | :--- |
> | RewardBench 2 | **APS (Ours)** | **1.7937** |
> | RewardBench 2 | Baseline | 1.7145 |
> | RM-Bench | **APS (Ours)** | **1.8690** |
> | RM-Bench | Baseline | 1.8281 |
>
> In this table, higher Reward score indicates better performance.
>
> ---
>
> > **Q2. Posterior sampling is also being used less and less in continuous diffusion, especially because the additional inference cost overhead becomes unacceptable for large scale generation models. This hinders the practical value of posterior sampling.**
>
> **A2.** We thank the reviewer for this insightful observation regarding the community's shift. However, we believe discrete diffusion offers a unique advantage here. In continuous diffusion, score networks typically approximate the *mean* of the noise-conditional prior $\mathbb{E}_{p(X_0|x_t)}[X_0]$, which is often difficult to "tilt" or guide effectively without heavy computation. In contrast, discrete diffusion models explicitly parameterize the categorical distribution $p(X_0|x_t)$ in product form. As demonstrated in our paper, this explicit representation makes the distribution significantly more amenable to guidance and posterior sampling. We believe our work could spark interesting follow-up works along this direction.
>
> ---
>
> > **Q3. The core problem addressed in this paper is how to propagate gradients over discrete indices... Could the authors discuss the differences between this paper and those lines of work?**
>
> **A3.** We thank the reviewer for pointing out this relevant line of work. While both involve gradient propagation through discrete structures, the objectives differ fundamentally:
>
> 1.  **Task Difference:** Works like [1] focus on **representation learning** (building a better tokenizer/codebook), whereas our work focuses on **inference-time guidance**. APS is a training-free method that enables posterior sampling using *pre-trained* discrete backbones with existing tokenizers.
> 2.  **Methodological Contribution:** This paper derives a variational upper bound for training discrete diffusion posterior samplers from scratch (Theorem 3.1) and also by using a lightweight adapter (Theorem 3.2). The exact upper bounds are computationally expensive, which calls for a series of approximations such as the straight-through estimator (STE) that helps reduce the complexity from $\mathcal{O}(K^L)$ to $\mathcal{O}(KL)$.
>
> We note that STE previously appeared in (Bengio et al., 2013) and also in VQ-VAE (Oord et al., 2017). The proposed reference (Shi et al., 2025) is another form of tokenizing images. We agree that exploring how advanced tokenizers (like the one in Shi et al., 2025) interact with multi-modal priors like MMaDA is a promising direction. We will include this comparison and reference in our revised related work section.
>
> ### References
> [1] F. Shi et al. Scalable Image Tokenization with Index Backpropagation Quantization, ICCV 2025.
>
> [2] Bengio et al. "Estimating or propagating gradients through stochastic neurons for conditional computation." arXiv:1308.3432 (2013).
>
> [3] Van Den Oord et al. "Neural discrete representation learning." NeurIPS (2017).

---

> > ### Author Rebuttal · Reviewer_g3FB · 2026-04-03
> >
> > The author has addressed my concerns, so I will keep my original score.

---

### Decision · Program_Chairs · 2026-04-30

**Decision:**

Accept (regular)

**Comment:**

This paper introduces Anchored Posterior Sampling (APS), a test-time posterior sampling method for masked discrete diffusion models. APS combines quantized expectation, which enables gradient-like updates in discrete token space, with anchored remasking, prioritizing high-confidence tokens early during decoding. The method leverages a pretrained discrete diffusion backbone, updating only a lightweight adaptation network per sample, avoiding retraining the main model. The authors provide theoretical training and test-time bounds and demonstrate APS on image inverse problems (super-resolution, deblurring, inpainting, HDR, nonlinear blur), stylization, and language-model guidance.

Strengths include a novel focus on training-free posterior sampling, innovative quantized expectation and anchored remasking components, broad experimental validation across multiple domains, and efficient test-time adaptation relative to continuous diffusion methods. Reviewer concerns regarding small QA benchmarks, hyperparameter robustness, and table clarity were addressed in the rebuttal, including expanded evaluation and clarifications.

Limitations include limited comparisons to some continuous diffusion baselines, minor presentation issues (long formulas, table labels), and additional computation due to the inner optimization loop, though backpropagation through the large pretrained model is avoided.

Overall, APS is technically solid, methodologically novel, and empirically well-supported. Its broad applicability and lightweight test-time adaptation justify acceptance.